# Stimulus-dependent relationships between behavioral choice and sensory neural responses

Daniel Chicharro[1,2]*, Stefano Panzeri[1†], Ralf M Haefner[3†]*

[1]Neural Computation Laboratory, Center for Neuroscience and Cognitive Systems@UniTn, Istituto Italiano di Tecnologia, Rovereto, Italy; [2]Department of Neurobiology, Harvard Medical School, Boston, United States; [3]Brain and Cognitive Sciences, Center for Visual Science, University of Rochester, Rochester, United States

**Abstract** Understanding perceptual decision-making requires linking sensory neural responses to behavioral choices. In two-choice tasks, activity-choice covariations are commonly quantified with a single measure of choice probability (CP), without characterizing their changes across stimulus levels. We provide theoretical conditions for stimulus dependencies of activity-choice covariations. Assuming a general decision-threshold model, which comprises both feedforward and feedback processing and allows for a stimulus-modulated neural population covariance, we analytically predict a very general and previously unreported stimulus dependence of CPs. We develop new tools, including refined analyses of CPs and generalized linear models with stimulus-choice interactions, which accurately assess the stimulus- or choice-driven signals of each neuron, characterizing stimulus-dependent patterns of choice-related signals. With these tools, we analyze CPs of macaque MT neurons during a motion discrimination task. Our analysis provides preliminary empirical evidence for the promise of studying stimulus dependencies of choice-related signals, encouraging further assessment in wider data sets.

*For correspondence:
Daniel_Chicharro@hms.harvard.edu (DC);
ralf.haefner@rochester.edu (RMH)

[†]Co-senior authors

**Competing interests:** The authors declare that no competing interests exist.

## Introduction

How perceptual decisions depend on responses of sensory neurons is a fundamental question in systems neuroscience (*Parker and Newsome, 1998*; *Gold and Shadlen, 2001*; *Romo and Salinas, 2003*; *Gold and Shadlen, 2007*; *Siegel et al., 2015*; *van Vugt et al., 2018*; *O'Connell et al., 2018*; *Steinmetz et al., 2019*). The seminal work of *Britten et al., 1996* showed that responses from single cells in area MT of monkeys during a motion discrimination task covaried with behavioral choices. Similar activity-choice covariations have been found in many sensory areas during a variety of both discrimination and detection two-choice tasks (see *Nienborg et al., 2012*; *Cumming and Nienborg, 2016*, for a review). Identifying which cells encode choice, and how and when they encode it, is essential to understand how the brain generates behavior based on sensory information.

With two-choice tasks, Choice Probability (CP) has been the most prominent measure (*Britten et al., 1996*; *Parker and Newsome, 1998*; *Nienborg et al., 2012*) used to quantify activity-choice covariations. Although early studies (*Britten et al., 1996*; *Dodd et al., 2001*) explored potential dependencies of the CP on the stimulus content, no significant evidence was found of a CP stimulus dependency. Accordingly, it has become common to report for each neuron a *single* CP value to quantify the strength of activity-choice covariations. This scalar CP value has been typically calculated either only from trials with a single, non-informative stimulus level (e.g. *Dodd et al., 2001*; *Parker et al., 2002*; *Krug et al., 2004*; *Wimmer et al., 2015*; *Katz et al., 2016*; *Wasmuht et al., 2019*), or by pooling trials across stimulus levels (so-called grand CP [*Britten et al., 1996*]) under the

assumption that choice-related neural signals are separable from stimulus-driven responses (e.g. *Verhoef et al., 2015*; *Pitkow et al., 2015*; *Smolyanskaya et al., 2015*; *Bondy et al., 2018*). Alternatively, a single CP is sometimes obtained simply averaging CPs across stimulus levels (e.g. *Cai and Padoa-Schioppa, 2014*; *Latimer et al., 2015*; *Liu et al., 2016*). Even when activity-choice covariations are modeled jointly with other covariates of the neural responses using Generalized Linear Models (GLMs) (*Truccolo et al., 2005*; *Pillow et al., 2008*), the stimulus level and the choice value are also usually used as separate predictors of the responses (*Park et al., 2014*; *Runyan et al., 2017*; *Scott et al., 2017*; *Pinto et al., 2019*; *Minderer et al., 2019*).

This focus on characterizing a neuron by a single CP value is mirrored in the existing theoretical studies. Existing theoretical results rely on a standard feed-forward model of decision making in which a neural representation of the stimulus is converted by a threshold mechanism into a behavioral choice (*Shadlen et al., 1996*; *Cohen and Newsome, 2009b*; *Haefner et al., 2013*) assuming a single, zero-signal stimulus level, and hence ignoring stimulus dependencies of CPs. Furthermore, so far no analytical mechanistic model accounts for feedback contributions to activity-choice covariations known to be important empirically (*Nienborg and Cumming, 2009*; *Cumming and Nienborg, 2016*; *Bondy et al., 2018*).

The main contribution of this work is to extend CP analysis reporting a single CP value for each cell to a more complete characterization of within-cell patterns of choice-related activity across stimulus levels. First, we extended the analytical results of *Haefner et al., 2013* to the general case of informative stimuli and to include both feedforward and feedback sources of the covariation between the choice and each cell. Our results predict that CP stimulus dependencies can appear in a cell-specific way because of stimulus-dependencies of cross-neuronal correlations. We show that they can also appear for all neurons because of the transformation of the neural representation of the stimulus into a binary choice, if the decision-making process relies on a threshold mechanism (or threshold criterion) to convert a continuous decision variable into a binary choice. Second, we developed two new analytical methods (a refined CP analysis and a new generalized linear model with stimulus-choice interactions) with increased power to detect stimulus dependencies in activity-choice covariations. Our new CP analysis isolates within-cell stimulus dependencies of activity-choice covariations from across-cells heterogeneity in the magnitude of the CP values, which may hinder their detection (*Britten et al., 1996*). Third, we applied this analysis framework to the classic dataset of *Britten et al., 1996* containing recordings from neurons in visual cortical area MT and found evidence for our predicted population-level threshold-induced dependency but also additional interesting cell-specific dependencies. We found consistent results on the existence of stimulus-choice interactions in neural activity both with our refined CP analysis and using generalized linear models with interaction terms. Finally, we show that main properties of the additional dependencies found can be explained modeling the cross-neuronal correlation structure induced by gain fluctuations (*Goris et al., 2014*; *Ecker et al., 2014*; *Kayser et al., 2015*; *Schölvinck et al., 2015*), which have been shown to explain a substantial amount of response variability in MT visual cortex (*Goris et al., 2014*).

## Results

We will first present the analysis of a theoretical model of how informative stimuli modulate choice probabilities. We will then analyze MT visual cortex neuronal responses from *Britten et al., 1996*, applying new methods developed to quantify stimulus-dependent activity-choice covariations with CPs and GLMs. This analysis provides preliminary empirical evidence in support of using these new methods for studying stimulus dependencies of activity-choice covariations.

### A general account for choice-related neural signals in the presence of informative stimuli

In a two-choice psychophysical task, such as a stimulus discrimination or detection task, a neuron is said to contain a 'choice-related signal', or 'decision-related signal' when its activity carries information about the behavioral choice above and beyond the information that it carries about the stimulus (*Britten et al., 1996*; *Parker and Newsome, 1998*; *Nienborg et al., 2012*). The interpretation of choice-related signals in terms of decision-making mechanisms is however difficult. Much progress in our understanding of their meaning has relied on using models to derive mathematically the

relationship between the underlying decision-making mechanisms and different measures of activity-choice covariation (*Haefner et al., 2013*; *Pitkow et al., 2015*) usually used to quantify choice-related signals.

The most widely used measure of activity-choice covariation for tasks involving two choices is *choice probability*, $\mathrm{CP}$. The $\mathrm{CP}$ is defined as the probability that a random sample of neural activity from all trials with behavioral choice $D$ equal to 1 is larger than one sample randomly drawn from all trials with choice $D = -1$ (*Britten et al., 1996*; *Parker and Newsome, 1998*; *Nienborg et al., 2012*; *Haefner et al., 2013*):

$$\mathrm{CP} \equiv \int_{-\infty}^{\infty} \mathrm{d}r\, p(r|D=1) \int_{-\infty}^{r} \mathrm{d}r'\, p(r'|D=-1), \tag{1}$$

where $r$ is any measure of the neural activity, which we will here consider to be the neuron's per-trial spike count. Another prominent measure of choice-related signals is *choice correlation*, $\mathrm{CC}$ (*Pitkow et al., 2015*). This quantity is defined under the assumption that the binary choice $D$ is mediated by an intermediate continuous decision value, $d$. This value may represent the brain's estimate of the stimulus, or an internal belief about the correct choice. The definition of CC further assumes that the categorical choice $D$ is related to $d$ via a thresholding operation such that the choice depends on whether $d$ is smaller or larger than a threshold θ (*Gold and Shadlen, 2007*). Its expression is as follows:

$$\mathrm{CC} \equiv \mathrm{corr}(r, d) = \frac{\mathrm{cov}(r, d)}{\sqrt{\mathrm{var}\, r}\sqrt{\mathrm{var}\, d}}, \tag{2}$$

where $\mathrm{cov}(r, d)$ is the covariance of the neural responses with $d$, and $\mathrm{var}\, r$, $\mathrm{var}\, d$ their variance across trials. Perhaps, the simplest measure of activity-choice covariation, which has been used in empirical studies (*Mante et al., 2013*; *Ruff et al., 2018*), is what we called the *choice-triggered average*, $\mathrm{CTA}$, defined as the difference between a neuron's average spike count $r$ across trials with behavioral decision $D = 1$ minus the average spike count in trials with decision $D = -1$:

$$\mathrm{CTA} \equiv \langle r \rangle_{D=1} - \langle r \rangle_{D=-1}. \tag{3}$$

The CP and CTA quantify activity-choice covariations without assumptions about the underlying decision-making mechanisms. However, their interpretation has commonly (*Nienborg et al., 2012*) been informed in previous analytical and computational studies by assuming a specific feedforward decision-threshold model of choice-related signals (*Shadlen et al., 1996*; *Cohen and Newsome, 2009b*). *Haefner et al., 2013* used that model to derive an analytical expression for CP valid under two assumptions that are often violated in practice: first, the model assumes a causally feedforward structure in which sensory responses caused the decision, and second, it is assumed that both decisions are equally likely. However, the presence of informative stimuli leads to one choice being more likely than the other, hampering the application of the analytical results to Grand CPs and to detection tasks (*Bosking and Maunsell, 2011*; *Smolyanskaya et al., 2015*), which involve informative stimuli. Furthermore, decision-related signals have empirically been shown to reflect substantial feedback components (*Nienborg and Cumming, 2009*; *Nienborg et al., 2012*; *Macke and Nienborg, 2019*). We will next extend this previous model (*Haefner et al., 2013*) to obtain a general expression of the CP valid for informative stimuli and regardless of the feedforward or feedback origin of the dependencies between the neural responses and the decision variable.

We first consider a most generic model in which we simply assume that the response $r_i$ of the $i - th$ sensory neurons covaries with the behavioral decision $D$, but without making any assumption about the origin of that covariation (*Figure 1A*). We find that to a first approximation (exact solution provided in Methods), the CP of cell $i$ captures the difference between the distributions $p(r_i|D = 1)$ and $p(r_i|D = -1)$ resulting from a difference in their means, and hence is related to the CTA:

$$\mathrm{CP}_i \approx \frac{1}{2} + \frac{1}{2\sqrt{\pi}} \frac{\mathrm{CTA}_i}{\sqrt{\mathrm{var}\, r_i}}. \tag{4}$$

The $\mathrm{CTA}$ generically quantifies the linear dependencies between responses and choice, and this approximation of the CP does not depend on their feedforward or feedback origin (*Figure 1A*). We

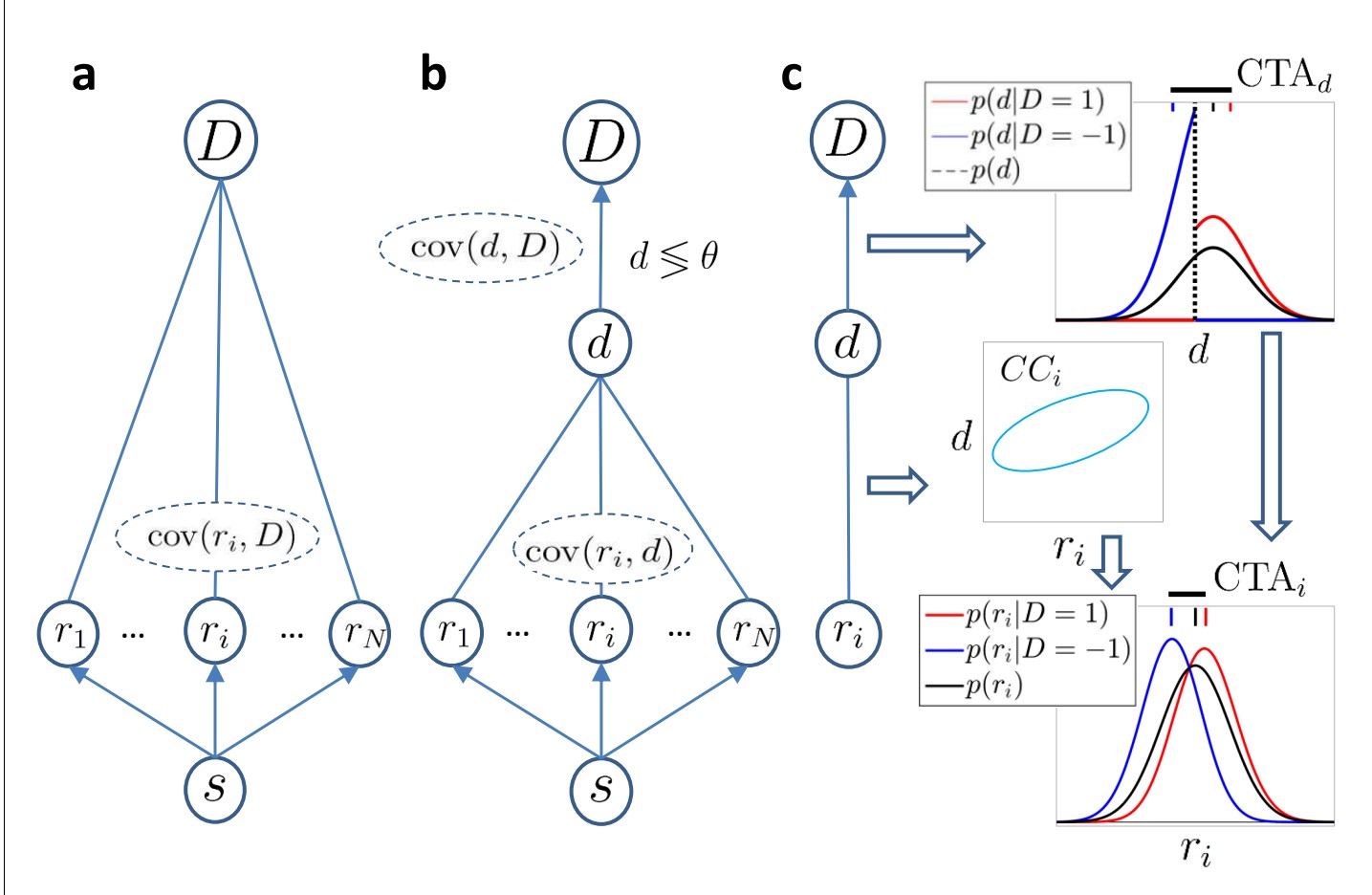

**Figure 1.** Models of choice probabilities. Arrows indicate causal influences. Undirected edges indicate relationships that may be due to feedforward, feedback, and/or common inputs. (a) A model agnostic to the causal origin of the choice–response covariation: the response of sensory neurons encoding a stimulus $s$ covaries with choice $D$. (b) Threshold model with a continuous decision variable $d$ mediating the relationship between responses and choice. The binary decision is made comparing $d$ to a threshold θ. (c) The threshold mechanism (vertical dashed black line) dichotomizes the $d$-space, resulting in a difference between the means of the conditional distributions associated with $D = \pm 1$ (red and blue vertical dashes on top of figure). This difference is quantified by $\mathrm{CTA}_d$ (horizontal thick black line) and implies a non-zero difference between the choice-triggered average responses ($\mathrm{CTA}_i$) in the presence of a correlation, $\mathrm{CC}_i$, between $d$ and $r_i$.

next add the assumption that the relationship between a neuron's response and the choice is mediated by the continuous variable $d$, as commonly assumed by previous studies and described above (*Figure 1B*). This splits any correlation between the neural response $r_i$ and choice $D$ into the product of the two respective correlations: $\mathrm{corr}(r_i, D) = \mathrm{corr}(r_i, d)\mathrm{corr}(d, D) = \mathrm{CC}_i\mathrm{corr}(d, D)$, where $\mathrm{CC}_i = \mathrm{corr}(r_i, d)$ is the choice correlation as defined in *Equation 2*. It follows (see Methods) that:

$$\mathrm{CTA}_i = \mathrm{CC}_i \frac{\sqrt{\mathrm{var}\, r_i}}{\sqrt{\mathrm{var}\, d}} \mathrm{CTA}_d, \tag{5}$$

where $\mathrm{CTA}_d$ is the average difference in $d$ between the two choices, in analogy to the $\mathrm{CTA}_i$ for neuron $i$. *Equation 5* describes how activity-choice covariations appear in the model (*Figure 1C*): the threshold mechanism dichotomizes the space of the decision variable, resulting in a different mean of $d$ for each choice, which is quantified in $\mathrm{CTA}_d$. If the activity of cell $i$ is correlated with the decision variable $d$ (non zero $\mathrm{CC}_i$), the $\mathrm{CTA}_d$ is then reflected in the $\mathrm{CTA}_i$ of the cell. In previous theoretical work (*Haefner et al., 2013*), the distribution over $d$ was assumed to be fixed and centered on the threshold value θ. Here, we remove that assumption and consider that $d$ may not be centered on the

threshold if the stimulus is informative, containing evidence in favor of one of the two choices, or if the choice is otherwise biased. In those cases, the normalized $\mathrm{CTA}_d$ in **Equation 5**, namely $\mathrm{CTA}_d/\sqrt{\mathrm{var}\,d}$, can be determined (see Materials and methods) in terms of the probability of choosing choice 1, $p_{\mathrm{CR}} \equiv p(D=1) = p(d>\theta)$, which we call the 'choice rate', $p_{\mathrm{CR}}$. Since the decision variable is determined as the combination of the responses of many cells, its distribution is well approximated by a Gaussian distribution, but now with a nonzero mean determined by the stimulus content. With this assumption, the normalized $\mathrm{CTA}_d$ for $p_{\mathrm{CR}} = 0.5$ is equal to $4/\sqrt{2\pi}$, and for each other $p_{\mathrm{CR}}$ value differs by a scaling factor

$$h(p_{\mathrm{CR}}) = \frac{\sqrt{2\pi}\,\phi(\Phi^{-1}(p_{\mathrm{CR}}))}{4p_{\mathrm{CR}}(1-p_{\mathrm{CR}})}, \qquad (6)$$

where $\phi(x)$ is the density function of a zero-mean, unit variance, Gaussian distribution, and $\Phi^{-1}$ is the corresponding inverse cumulative density function. By construction, $h(p_{\mathrm{CR}}) = 1$ for $p_{\mathrm{CR}} = 0.5$ where it has its minimum. Given the factor $h(p_{\mathrm{CR}})$, combining **Equations 4 and 5** we can relate CP and CC across different ratios $p_{\mathrm{CR}}$, corresponding to different stimulus levels, irrespectively of whether CP is caused by feedforward or feedback signals. In the linear approximation (see Methods for the exact formula and derivation with the decision-threshold model), this relationship reads:

$$\mathrm{CP}_i(p_{\mathrm{CR}}) \approx \frac{1}{2} + \frac{\sqrt{2}}{\pi} h(p_{\mathrm{CR}})\,\mathrm{CC}_i(p_{\mathrm{CR}}). \qquad (7)$$

For equal fractions of choices, $p_{\mathrm{CR}} = 0.5$, this CP expression corresponds to the linear approximation derived in **Haefner et al., 2013**. Note that extending the CP formula to $p_{\mathrm{CR}} \neq 0.5$ required us to also make explicit the dependency of the choice correlations on the choice rate, $\mathrm{CC}_i(p_{\mathrm{CR}})$. Unlike $h(p_{\mathrm{CR}})$ which is an effect of the decision-making threshold mechanism and shared by all neurons, $\mathrm{CC}_i(p_{\mathrm{CR}})$ is specific to and generally different for each neuron, reflecting its role in the perceptual decision-making process. A CC stimulus dependence may arise as a result of stimulus-dependent decision feedback (**Haefner et al., 2016**; **Bondy et al., 2018**; **Lange and Haefner, 2017**), or other sources of stimulus-dependent cross-neuronal correlations (**Ponce-Alvarez et al., 2013**; **Orbán et al., 2016**) such as shared gain fluctuations (**Goris et al., 2014**). In fact, we will show below that gain-induced stimulus-dependent cross-neuronal correlations account for observed features in our empirical data. Note that we do not distinguish between CC stimulus dependencies and a dependence of the CC on $p_{\mathrm{CR}}$. We do not make this distinction here because most generally a change in the stimulus level results in a change of $p_{\mathrm{CR}}$, and the two cannot be disentangled. However, the $p_{\mathrm{CR}}$ more generally depends on other factors such as the reward value, attention level, or arousal state, and in **Equation 7** the separate dependencies on the stimulus and $p_{\mathrm{CR}}$ can be explicitly indicated as $\mathrm{CC}_i(p_{\mathrm{CR}}, s)$ when the experimental paradigm allows to separate these two influences.

For simplicity, we presented above only the general relationship between the CP and CC in **Equation 7** derived as a linear approximation for weak activity-choice covariations, as this is the regime relevant for single sensory neurons. See Methods for the exact analytical solution from the threshold model (**Equation 16**) and Appendix 1 for its derivation. Despite the assumption of weak activity-choice covariations, this approximation is very close over the empirically relevant range of CC's (**Figure 2A–B**). Below we will focus on a concrete type of CC stimulus dependence, namely originated by gain fluctuations, but it is clear from **Equation 7** that any CC stimulus dependence will modify the $\mathrm{CP}(p_{\mathrm{CR}})$ shape induced purely by the threshold effect. A summary of the overall relation between the CP, $\mathrm{CTA}$, and CC is provided in **Figure 2D**.

The model provides a concrete prediction of a stereotyped dependence of CP on $p_{\mathrm{CR}}$ through $h(p_{\mathrm{CR}})$ when the choice-related signals are mediated by an intermediate decision variable $d$, which is testable using data. First, under the assumption that CC is constant and therefore $h(p_{\mathrm{CR}})$ is the only source of CP dependence on $p_{\mathrm{CR}}$, for a positive CC (CP>0.5), the $\mathrm{CP}(p_{\mathrm{CR}})$ should have a *minimum* at $p_{\mathrm{CR}} = 0.5$ and increase symmetrically as $p_{\mathrm{CR}}$ deviates from 0.5 as the result of a change in the stimulus in either direction (**Figure 2A**). When the CC is negative (CP<0.5), then $\mathrm{CP}(p_{\mathrm{CR}})$ should have a *maximum* at $p_{\mathrm{CR}} = 0.5$ and analogously decrease symmetrically as $p_{\mathrm{CR}}$ deviates from 0.5. Second, since the influence of $h(p_{\mathrm{CR}})$ is multiplicative, it creates higher absolute differences in the CP across different stimulus levels for cells with a stronger CP (either larger or smaller than 0.5). Third, the dependence on $h(p_{\mathrm{CR}})$ is weak for a wide range of $p_{\mathrm{CR}}$ values (**Figure 2A**), making it empirically

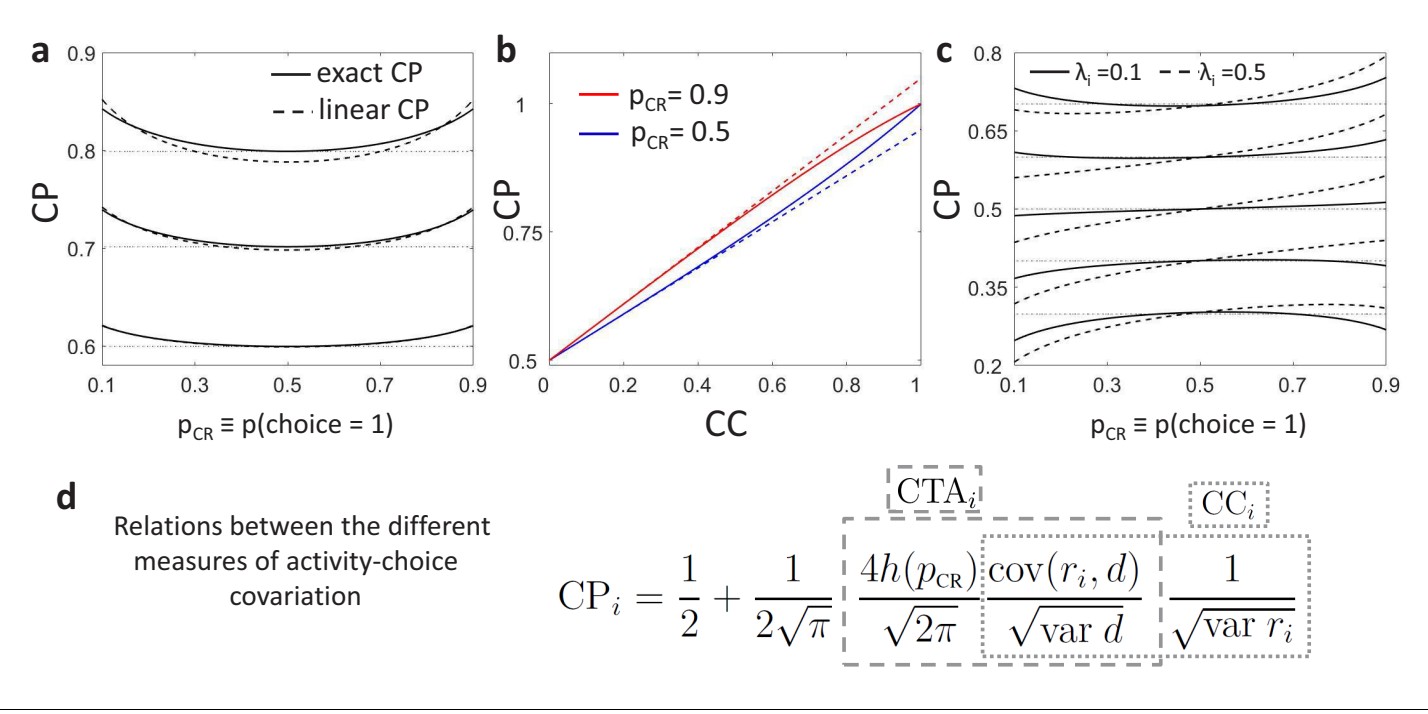

**Figure 2.** Predictions for stimulus dependencies from the threshold model. (a) CP dependence on $p_{CR}$ through the threshold-induced factor $h(p_{CR})$. Results are shown for three values of a stimulus-independent choice correlation, $CC_i$, isolating the shape of $h(p_{CR})$ from other stimulus dependencies. Solid curves represent the exact solution of the CP obtained from our model (see Methods, *Equation 16*) and dashed curves its linear approximation (*Equation 7*). (b) Comparison of the exact solution of the CP (solid) and its linear approximation (dashed), as a function of the magnitude of a stimulus-independent choice correlation. Results are shown for two values of $p_{CR}$, 0.5 and 0.9. (c) CP dependence on $p_{CR}$ when together with the factor $h(p_{CR})$ stimulus dependencies also appear through stimulus-dependent choice correlations induced by response gain fluctuations (*Equation 11*). Results are shown for five values of $CC(p_{CR} = 0.5)$ (dotted horizontal lines) and in each case for two values of $\lambda_i$, the fraction of the variance of a cell $i$ caused by the gain fluctuations (Methods). (d) Summary of the derived relationships as provided by *Equations 4-7*.

detectable only when including highly informative stimuli in the analysis to obtain $p_{CR}$ values very different from 0.5. However, for those $p_{CR}$ values, CP estimates are less reliable, because only for few trials the choice is expected to be inconsistent with the sensory information, meaning that one of the two distributions $p(r_i|D=1)$ or $p(r_i|D=-1)$ is poorly sampled. This means that to detect the $h(p_{CR})$ modulation for single cells, many trials would be needed for each value of $p_{CR}$ to obtain good estimates. Because $h(p_{CR})$ is common to all cells, averaging $CP(p_{CR})$ profiles across cells can also improve the estimation. This averaging may also help to isolate the $h(p_{CR})$ modulation, assuming that cell-specific $CP_i$ stimulus dependencies introduced through choice correlations $CC_i$ are heterogeneous across cells and average out. We refer to Appendix 1 for a detailed analysis of the statistical power for the detection of $h(p_{CR})$ as a function of the number of trials and cells used to estimate an average $CP(p_{CR})$ profile. We will present below (Section 'Stimulus dependence of choice-related signals in the responses of MT cell') evidence for the $h(p_{CR})$ modulation from a re-analysis of the data in *Britten et al., 1996*.

### The structure of CP stimulus dependencies induced by response gain fluctuations

We will now focus on a concrete source of stimulus-dependent correlations that leads to a non-constant $CC(p_{CR})$, namely the effect of gain fluctuations into the stimulus-response relationship (*Goris et al., 2014*; *Ecker et al., 2014*; *Kayser et al., 2015*; *Schölvinck et al., 2015*). *Goris et al., 2014* showed that 75% of the variability in the responses in monkeys MT cells when presented with drifting gratings could be explained by gain fluctuations. We derive the CP dependencies on $p_{CR}$ in a feedforward model of decision-making (*Shadlen et al., 1996*; *Haefner et al., 2013*) that also

models the effect of gain fluctuations in the responses. The feedfoward model considers a population of sensory responses, $\vec{r} = (r_1, ..., r_n)$, with tuning functions $\vec{f}(s) = (f_1(s), ..., f_n(s))$, responses $r_i = f_i(s) + \xi_i$, and a covariance structure $\Sigma$ of the neuron's intrinsic variability $\xi_i$. The responses are read out into the decision variable with a linear decoder

$$d = \vec{w}^\top \vec{r} \equiv \sum_{i=1}^{n} w_i r_i, \tag{8}$$

where $\vec{w}$ are the read-out weights. The categorical choice $D$ is made by comparing $d$ to a threshold θ. With this model, the general expression of *Equation 7* reduces to

$$\mathrm{CP}_i(p_{\mathrm{CR}}) \approx \frac{1}{2} + \frac{\sqrt{2}}{\pi} h(p_{\mathrm{CR}}) \frac{(\Sigma(s)\vec{w})_i}{\sqrt{\Sigma_{ii}(s)} \sqrt{\vec{w}^\top \Sigma(s)\vec{w}}}. \tag{9}$$

where $(\Sigma(s)\vec{w})_i = \mathrm{cov}(r_i, d)$ and $\mathrm{var}\, d = \vec{w}^\top \Sigma(s)\vec{w}$. This expression corresponds to the one derived by *Haefner et al., 2013*, except for $h(p_{\mathrm{CR}})$ and for the fact that we now explicitly indicate the dependence of the correlation structure $\Sigma(s)$ on the stimulus. The expression relates the CP magnitude to single-unit properties such as the neurometric sensitivity, as well as to population properties, such as the decoder pooling size and the magnitude of the cross-neuronal correlations, which determine CC (*Shadlen et al., 1996*; *Haefner et al., 2013*). In particular, if the decoding weights are optimally tuned to the structure of the covariability $\Sigma(p_{\mathrm{CR}} = 0.5)$ at the decision boundary, this results in a proportionality between $\mathrm{CP}_i(p_{\mathrm{CR}} = 0.5)$ and the neurometric sensitivity of the cells: $\mathrm{CP}_i(p_{\mathrm{CR}} = 0.5) \propto f_i'/\sigma_{r_i}$ (*Haefner et al., 2013*), as has been experimentally observed (*Britten et al., 1996*; *Parker and Newsome, 1998*). While this feedfoward model is generic, we concretely study CC stimulus dependencies induced by the effect of global gain response fluctuations in cross-neuronal correlations. Following *Goris et al., 2014* we modeled the responses of cell $i$ in trial $k$ as $f_{ik}(s) = g_k f_i(s)$, where $g_k$ is a gain modulation factor shared by the population. We assume that the readout weights $\vec{w}$ are stimulus-independent. As a consequence, the covariance of population responses $\Sigma$ has a component due to the gain fluctuations:

$$\Sigma(s) = \bar{\Sigma} + \sigma_G^2 \vec{f}(s)\vec{f}^\top(s), \tag{10}$$

where $\sigma_G^2$ is the variance of the gain $g$ and $\bar{\Sigma}$ is the covariance not associated with the gain, which for simplicity we assume to be stimulus independent. The component of the cross-neuronal covariance matrix $\Sigma$ induced by gain fluctuations is proportional to the tuning curves ($\propto \vec{f}(s)\vec{f}^T(s)$). A deviation $\Delta s \equiv s - s_0$ of the stimulus from the uninformative stimulus $s_0$ produces a change $\Delta \vec{f} = \vec{f}'(s_0)\Delta s$ in the population firing rates, which affects the variability of the responses, the variability of the decoder, and their covariance, which all vary with $\Delta s$. Because the variance of the decoder $\mathrm{var}\, d = \vec{w}^\top \Sigma(s)\vec{w}$ and the covariance $\mathrm{cov}(r_i, d) = (\Sigma(s)\vec{w})_i$ both depend on the concrete form of the read-out weights, the effect of gain-induced stimulus dependencies on the CP is specific for each decoder. Under the assumption of an optimal linear decoder at the decision boundary $s_0$ ($\vec{w} \propto \Sigma^{-1}\vec{f}'(s_0)$), we obtain an approximation of the CC dependence on the stimulus deviation $\Delta s$ from $s_0$ (see Methods for details):

$$\mathrm{CC}_i(p_{\mathrm{CR}}) = \mathrm{CC}_i(p_{\mathrm{CR}} = 0.5) + \sigma_G \lambda_i \left[1 - \mathrm{CC}_i^2(p_{\mathrm{CR}} = 0.5)\right] \frac{\Delta s}{\sqrt{\mathrm{var}\, d}}, \tag{11}$$

where the slope is determined by the coefficient $\beta_{p_{\mathrm{CR}}} = \sigma_G \lambda_i \left[1 - \mathrm{CC}_i^2(p_{\mathrm{CR}} = 0.5)\right]$, with $\lambda_i$ being the fraction of the variance of cell $i$ caused by the gain fluctuations (Methods). The choice rate $p_{\mathrm{CR}}$ is determined by the stimulus $\Delta s$ as characterized by the psychometric function. For this form of the slope coefficient $\beta_{p_{\mathrm{CR}}}$ obtained with an optimal decoder all the factors contributing to it are positive (*Figure 2C*). In Appendix 4 we further analytically describe how gain fluctuations introduce CP stimulus dependencies not only for an optimal decoder, but also for any unbiased decoders. Conversely to the factor $h(p_{\mathrm{CR}})$, the pattern of $\mathrm{CP}(p_{\mathrm{CR}})$ profiles produced by the gain fluctuations is cell-specific, with a stronger asymmetric component for cells with higher $\lambda_i$ (*Figure 2C*). Furthermore, while the sign of the multiplicative modulation $h(p_{\mathrm{CR}})$ changes when CC>0 or CC<0, the gain-induced contribution in *Equation 11* is additive. As seen in *Figure 2C*, for cells with a weak activity-choice

covariation for uninformative stimuli ($\mathrm{CP}(p_{\mathrm{CR}} = 0.5)$ close to 0.5), this implies that the CP of a neuron can actually change from below 0.5 to above 0.5 across the stimulus range presented in the experiment.

## Stimulus dependencies of choice-related signals in the responses of MT cells

In the light of our findings above, we re-analyzed the classic *Britten et al., 1996* data containing responses of neurons in area MT in a coarse motion direction discrimination task (see Methods for a description of the data set). Our objective is to identify any patterns of CP dependence on the choice rate/stimulus level. First, we describe our results testing for the threshold-induced CP stimulus dependence, $h(p_{\mathrm{CR}})$, and then more generally we characterize the $\mathrm{CP}(p_{\mathrm{CR}})$ patterns found in the data using clustering analysis. Finally, as an alternative to CP analysis, we show how to extend Generalized Linear Models (GLMs) of neural activity to include stimulus-choice interaction terms that incorporate the stimulus dependencies of activity-choice covariations derived with our theoretical approach and found above in the MT data.

### Testing the presence of a threshold-induced CP stimulus dependence in experimental data

We start describing how to analyze within-cell $\mathrm{CP}(p_{\mathrm{CR}})$ profiles to test the existence of the threshold-induced modulation. The theoretically derived properties of $h(p_{\mathrm{CR}})$ suggest several empirical signatures that will be reflected in the within-cell $\mathrm{CP}(p_{\mathrm{CR}})$ profiles. First, because $h(p_{\mathrm{CR}})$ introduces a multiplicative modulation of the choice correlation, for informative stimuli it leads to an increase of the CP for cells with positive choice correlation ($\mathrm{CP} > 0.5$) and to a decrease for cells with negative choice correlation ($\mathrm{CP} < 0.5$). Second, because $h(p_{\mathrm{CR}})$ is multiplicative, the absolute magnitude of the modulation will be higher for cells with stronger choice correlation, that is CPs most different from 0.5. Third, the effect of $h(p_{\mathrm{CR}})$ is strongest when one choice dominates and hence most noticeable for highly informative stimuli.

These properties of $h(p_{\mathrm{CR}})$ indicate that, to detect this modulation, it is necessary to examine within-cell $\mathrm{CP}(p_{\mathrm{CR}})$ profiles isolated from across-cells heterogeneity in the magnitude of the CP values. Ideally, we would like to calculate a $\mathrm{CP}(p_{\mathrm{CR}})$ profile for each cell and analyze the shape of these single-cell profiles. However, given the available number of trials, estimates of $\mathrm{CP}(p_{\mathrm{CR}})$ profiles for single cells are expected to be noisy. The estimation error of the CP is higher when $p_{\mathrm{CR}}$ is close to 0 or 1, the same $p_{\mathrm{CR}}$ values for which the $h(p_{\mathrm{CR}})$ modulation would be most noticeable. The standard error of $\hat{\mathrm{CP}}$ can be approximated as $\mathrm{SEM}(\hat{\mathrm{CP}}) \approx 1/\sqrt{12 K p_{\mathrm{CR}}(1 - p_{\mathrm{CR}})}$ (*Bamber, 1975*; *Hanley and McNeil, 1982*, see Methods), where $K$ is the number of trials. In the Britten et al. data set the number of trials varies for different stimulus levels, and most frequently $K = 30$ for highly informative stimuli. In that case, for $p_{\mathrm{CR}} = 0.9$, only three trials for choice $D = -1$ are expected, and $\mathrm{SEM}(\hat{\mathrm{CP}}) \approx 0.18$. As can be seen from *Figure 2A*, this error surpasses the order of magnitude of the CP modulations expected from $h(p_{\mathrm{CR}})$. This means that we need to combine CP estimates of adjacent $p_{\mathrm{CR}}$ values, and/or combine estimated $\mathrm{CP}(p_{\mathrm{CR}})$ profiles across neurons, to reduce the standard error (See Appendix 1 for a detailed analysis of the statistical power for the detection of $h(p_{\mathrm{CR}})$).

When averaging CPs across neurons, two considerations are important. First, cells that for $p_{\mathrm{CR}} = 0.5$ have a CP higher or lower than 0.5 should be separated, given that the sign of the CC leads to an inversion of the profile resulting from $h(p_{\mathrm{CR}})$ (*Equation 7*). If not separated, the $h(p_{\mathrm{CR}})$-dependence would average out, or the average $\mathrm{CP}(p_{\mathrm{CR}})$ profile would reflect the proportion of cells with CPs higher or lower than 0.5 in the data set. Second, the average should correspond to an average -across cells- of within-cell $\mathrm{CP}(p_{\mathrm{CR}})$ profiles, and hence it should only include cells for which a full $\mathrm{CP}(p_{\mathrm{CR}})$ profile can be calculated. This is important because for each cell $i$ the $h(p_{\mathrm{CR}})$ modulation is relative to the value of $\mathrm{CP}_i(p_{\mathrm{CR}} = 0.5)$. If a different subset of cells was included in the average of the CP at each $p_{\mathrm{CR}}$ value, the resulting shape across $p_{\mathrm{CR}}$ values of the averaged CPs would not be an average of within-cell $\mathrm{CP}(p_{\mathrm{CR}})$ profiles. Conversely, in that case, the resulting shape would reflect the heterogeneity in the magnitude of the CP values across the subsets of cells averaged at each $p_{\mathrm{CR}}$ value. In the single-cell recordings from Britten et al., the range of stimulus levels used varies across neurons, and for a substantial part of the cells a full $\mathrm{CP}(p_{\mathrm{CR}})$ profile cannot be constructed.

Following the second consideration, those cells were excluded from the analysis to avoid that they only contributed to the average at certain $p_{CR}$ values.

We derived the following refined procedure to analyze $\text{CP}(p_{CR})$ profiles. As a first step, we constructed a $\text{CP}(p_{CR})$ profile for each cell. First, for each cell and each stimulus coherence level we calculated a CP estimate if at least four trials were available for each decision. For the experimental data set, CPs are always estimated from its definition (*Equation 1*), and we will only use the theoretical expression of $h(p_{CR})$ to fit the modulation of the experimentally estimated $\text{CP}(p_{CR})$ profiles. Second, as a first way to improve the CP estimates, we binned $p_{CR}$ values into five bins and assigned stimulus coherence levels to the bins according to the psychometric function that maps stimulus levels to $p_{CR}$, with the central bin containing the trials from the zero-signal stimulus. A single CP value per bin for each cell was then obtained as a weighted average of the CPs from stimulus levels assigned to each bin. The weights were calculated as inversely proportional to the standard error of the estimates, giving more weight to the most reliable CPs (see Methods). The results that we present hereafter are all robust to the selection of the minimum number of trials and the binning intervals. Unless otherwise stated, in all following analyses we included all the cells ($N = 107$) for which we had data to compute CPs in all five bins, thus allowing us to estimate a full within-cell $\text{CP}(p_{CR})$ profile. As a second step, we averaged the within-cell $\text{CP}(p_{CR})$ profiles across cells, taking into account the two considerations above. As before, averages were weighted by inverse estimation errors.

*Figure 3A* shows the averaged $\text{CP}(p_{CR})$ profiles. To assess the statistical significance of the CP dependence on $p_{CR}$, we developed a surrogates method to test whether a pattern consistent with the predicted CP-increase for informative stimuli could appear under the null hypothesis that the CP has a constant value independent of $p_{CR}$ (see Methods). For the cells with average CP higher than 0.5, we found that the modulation of the CP was significant ($p = 0.0006$), with higher CPs obtained

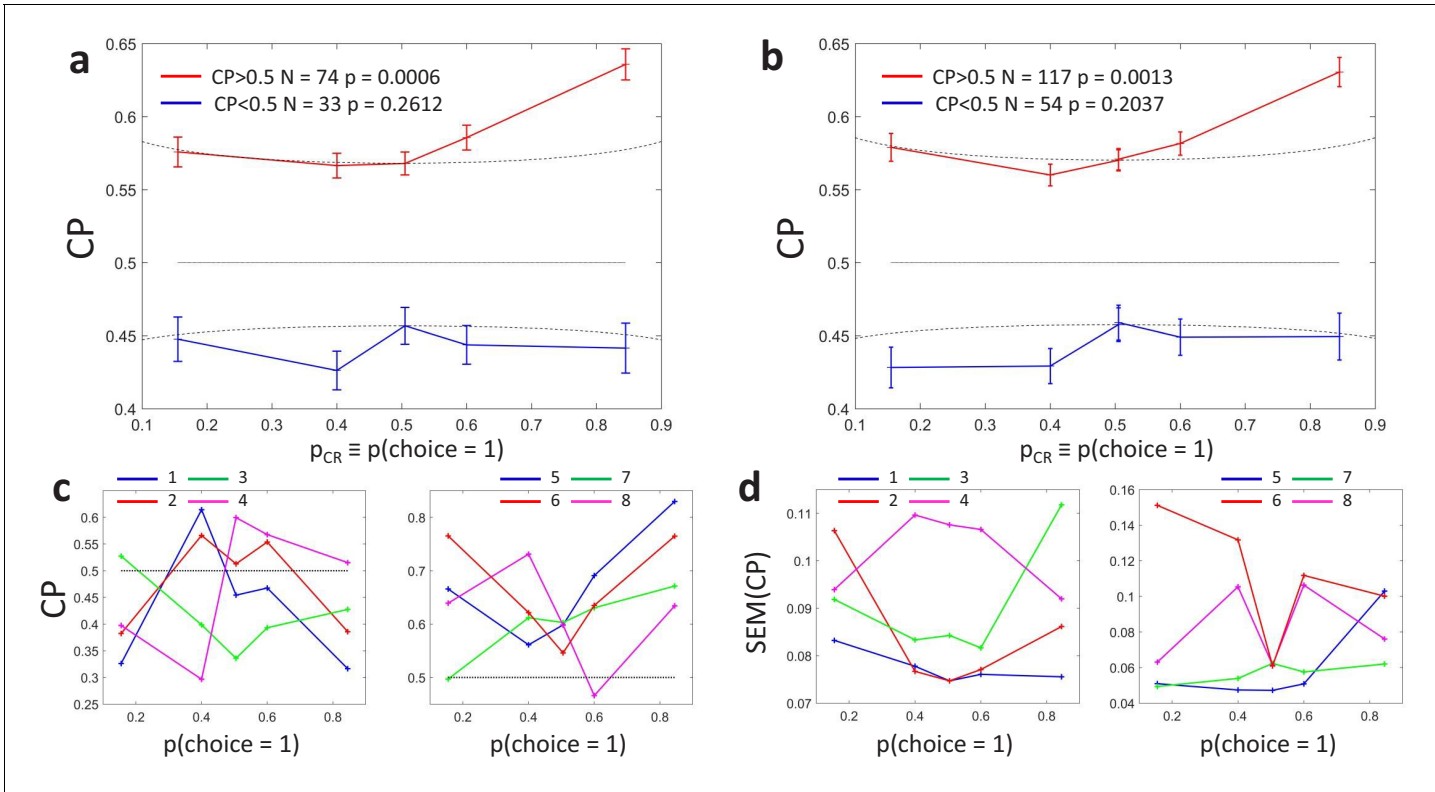

**Figure 3.** Choice probability as a function of the choice rate for MT cells during a motion direction discrimination task (*Britten et al., 1996*). (a) Average CP as a function of $p_{CR} \equiv p(D = 1)$. The average across $N = 107$ cells was calculated separately for cells with average CP higher or lower than 0.5. Dotted curves reflect the relationship predicted by the factor $h(p_{CR})$ (*Equation 6*). Significance of the stimulus dependencies was evaluated against the null hypothesis of a constant CP value using surrogate data (see Methods). (b) Same analysis but with a less strict inclusion criterion (see main text). (c) $\text{CP}(p_{CR})$ profile for four example cells with average CP lower and higher than 0.5, respectively. (d) Standard errors of the estimated CP for the example cells as a function of $p_{CR}$.

for $p_{CR}$ close to 0 or one in agreement with the model. For cells with average CP lower than 0.5, the modulation was not significant ($p = 0.26$). While the actual absence of a modulation would imply that the choice-related signals in these neurons are not mediated by a continuous intermediate decision-variable but may be, for example, due to categorical feedback, we point out the lower power of this statistical test due to fewer neurons being in the CP<0.5 group and the expected effect size being lower, too. First, there were 74 cells with CP higher than 0.5 but only 33 with CP lower than 0.5, meaning that the estimation error is larger for the average $CP(p_{CR})$ profile of the cells with CP<0.5. Second, as the modulation predicted by $h(p_{CR})$ is multiplicative, its impact is expected to be smaller when the magnitude of $CP - 0.5$ is smaller. *Figure 3A* shows that CP values are on average closer to 0.5 for the cells with CP<0.5, in agreement with Figure 5 of *Britten et al., 1996*. This means that fewer cells classified in the group with CP<0.5 have choice-related responses. Therefore, the fact that we cannot validate the prediction of an inverted symmetric $h(p_{CR})$ modulation for the cells with CP<0.5 with respect to the cells with CP>0.5 is not strong evidence against the existence of a threshold-induced CP stimulus dependence. We further confirmed the robustness of the results in a wider set of cells. For this purpose, we repeated the analysis forming subsets separately including cells with a computable CP for the three bins with $p_{CR}$ lower or equal 0.5, and the three with $p_{CR}$ higher or equal than 0.5. Also in this case the observed $CP(p_{CR})$ pattern was significant ($p = 0.0013$) for cells with average CP higher than 0.5 (*Figure 3B*, $N = 171$), and non-significant for cells with CP lower than 0.5 (p=0.20).

Interestingly, the identified significant $CP(p_{CR})$ dependence for the cells with CP>0.5 goes beyond the symmetric threshold-induced shape predicted by $h(p_{CR})$, both in magnitude and shape (*Figure 2A*), since the increase is bigger for $p_{CR}$ values close to 1 than to 0. This implies that the choice correlation for each neuron, $CC_i(p_{CR})$, must systematically change with $p_{CR}$ as well, contributing to the overall CP stimulus dependency observed. In particular, the observed average $CP(p_{CR})$ profile indicates that the CP increase appears to be higher for $p_{CR}$>0.5. The finding of this asymmetry is consistent with results reported in *Britten et al., 1996*, who found a significant but modest effect of coherence direction on the CP (see their Figure 3). By experimental design, the direction of the dots corresponding to choice $D = 1$ was tuned for each cell separately to coincide with their most responsive direction. This means that this asymmetry indicates that CPs tend to increase more when the stimulus provides evidence for the direction eliciting a higher response. However, *Britten et al., 1996* found no significant relation between the global magnitude of the firing rate and the CP (see their Figure 3), and we confirmed this lack of relation specifically for the subset of $N = 107$ cells (no significant correlation coefficient between average rate and average CP values, $p = 0.33$). This eliminates the possibility that higher CPs for high $p_{CR}$>0.5 values are due only to higher responses, and suggests a richer underlying structure of $CP(p_{CR})$ patterns, which we will investigate next using cluster analysis to identify the predominant patterns shared by the within-cell $CP(p_{CR})$ profiles.

## Characterizing the experimental patterns of CP stimulus dependencies with cluster analysis

We carried out unsupervised $k$-means clustering (*Bishop, 2006*) to examine the patterns of $CP(p_{CR})$ without a priori assumptions about a modulation $h(p_{CR})$ associated with the threshold effect. Clustering was performed on $CP(p_{CR}) - 0.5$, with each cell represented as a vector in a five-dimensional space, where five is the number of $p_{CR}$ bins used to summarize the data as described above. To consider both the shape and sign of the modulation, distances between neurons were calculated with the cosine distance between their $CP(p_{CR})$ profiles (one minus the cosine of the angle between the two vectors). Clustering was performed for a range of specified numbers of clusters. Specifying the existence of two clusters, we naturally recovered the distinction between cells with CP higher or lower than 0.5 (*Figure 4A*). The statistical significance of any $p_{CR}$-modulation was again assessed constructing surrogate $CP(p_{CR})$ profiles and repeating the clustering analysis on those surrogates. As before, a significant dependence of the CP on $p_{CR}$ was found only for the cluster associated with CP higher than 0.5 ($p = 0.0007$ for CP>0.5 and $p = 0.21$ for CP<0.5).

As mentioned above, the divergence from $h(p_{CR})$ of the average $CP(p_{CR})$ profile for cells with CP>0.5 suggests that cell-specific modulations are introduced through $CC_i(p_{CR})$. While the variability of individual $CP_i(p_{CR})$ profiles (*Figure 3C*) is expected to reflect substantially the high estimation

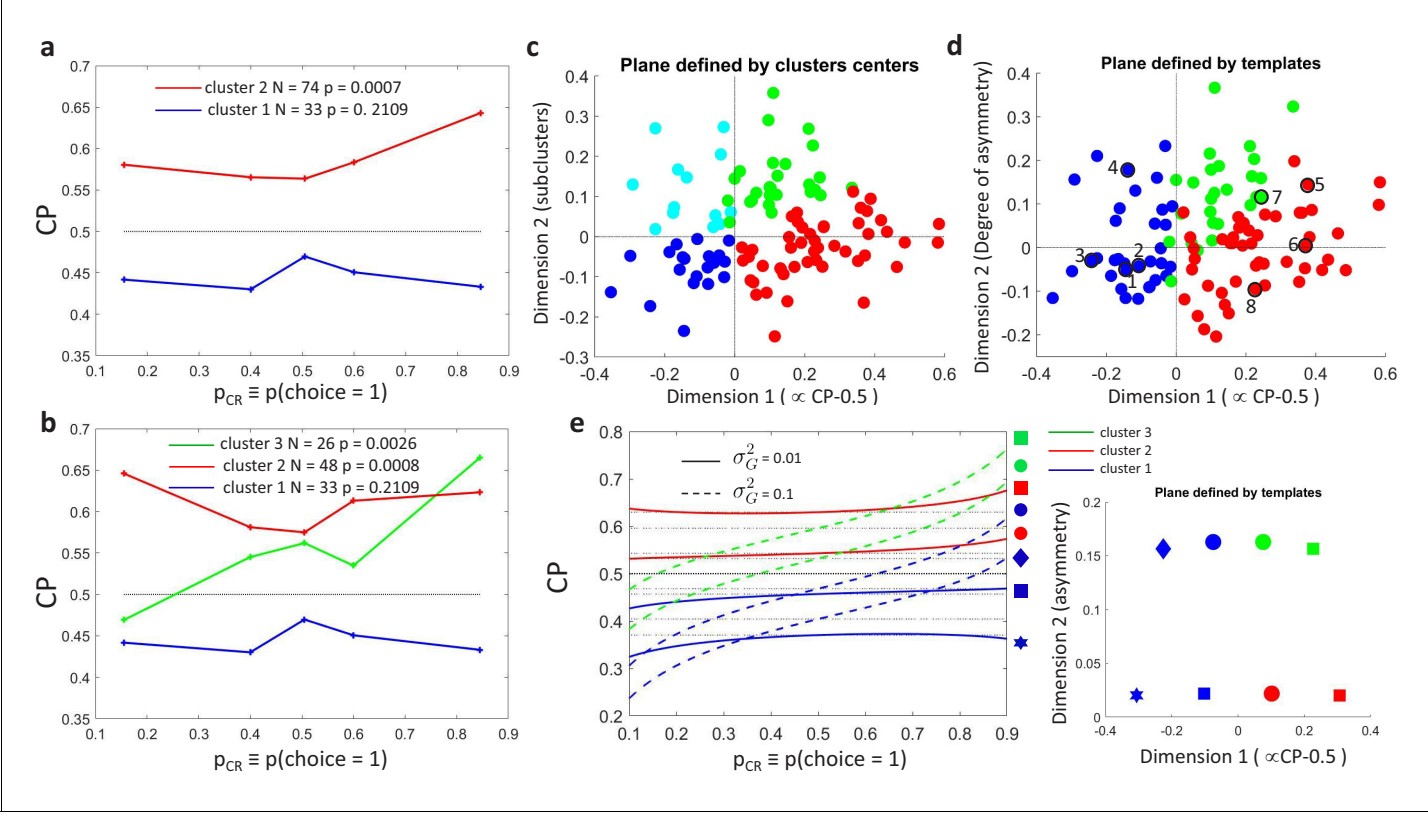

**Figure 4.** Clustering analysis of choice probability as a function of $p_{CR}$. (**a–b**) CP as a function of $p_{CR}$ for clusters of the MT cells determined by $k$-means clustering. Each $CP(p_{CR})$ profile corresponds to the center of a cluster. Significance of the modulation was quantified as in *Figure 3*. (**a**) Two clusters ($N_c = 2$) for all cells. (**b**) Further subclustering of cells with average CP>0.5 into two subclusters. (**c**) Representation of the $CP(p_{CR})$ profiles in a two-dimensional space spanned by the cluster means. The horizontal axis is defined by clusters 1 and 2 and closely aligned with $CP - 0.5$. Vertical axis is defined as perpendicular to horizontal axis in the plane defined by the subcluster means. Colors correspond to the clusters of panel b, with blue and cyan further indicating subclusters of cells with average CP<0.5 (see *Appendix 3—figure 1A*). (**d**) Space defined by projection onto two templates: a constant relationship (x-axis) representing the magnitude of $CP - 0.5$, and a monotonic relationship with slope 1 (y-axis) representing CP asymmetry. Colors correspond to the clusters of panel b and numbers indicate example cells shown in *Figure 3C*. (**e**) Modeling the influence of neuronal gain modulation on $CP(p_{CR})$ profiles. $CP(p_{CR})$ profiles for different combinations of strength of the gain fluctuations, $\sigma_G^2$, and the choice correlation that would be obtained for the uninformative stimulus $s_0$ with no gain fluctuations, $CC_{i0}(s_0)$. We display $CP(p_{CR})$ for four values of $CC_{i0}(s_0)$ (curves vertically separated) and two values of $\sigma_G^2$ (solid vs dashed). Each curve corresponds to a point in the two-dimensional space defined by the symmetric and asymmetric templates introduced in panel b. See Methods for model details.

errors of $\hat{CP}$ for the single cells (*Figure 3D*), the presence of subclusters can identify $CP(p_{CR})$ patterns common across cells.

We proceed to examine subclusters within the CP>0.5 cluster with a significant $CP(p_{CR})$ profile, excluding from our analysis cells within the CP<0.5 cluster (analogous results were found when increasing the number of clusters in a nonhierarchical way, without a priori excluding these cells, see *Appendix 3—figure 1A*). Average $CP(p_{CR})$ profiles obtained when inferring two subclusters of cells with CP>0.5 are shown in *Figure 4B*. For both subclusters the $CP(p_{CR})$ dependence is significant ($p = 0.0008$ for cluster two and $p = 0.0026$ for cluster 3, respectively, in *Figure 4B*). The larger cluster has a more symmetric shape of dependence on $p_{CR}$, with an increase of CP in both directions when the stimulus is informative, consistent with the prediction of a threshold-induced CP stimulus dependence $h(p_{CR})$. For the smaller cluster the dependence is asymmetric, with a CP increase when the stimulus direction is consistent with the preferred direction of the cells and a decrease in the opposite direction. We verified that no significant difference exists between the firing rates of the cells in the two subclusters (Wilcoxon rank-sum test, $p = 0.23$). The monotonic shape of the second subcluster mirrors the dependency produced by response gain fluctuations as predicted by the gain model described above. This suggests that the neurons in this subcluster differ from the neurons in the

other subcluster by a substantially larger gain-induced variability, a testable prediction for future experiments and further discussed below.

Introducing a second cluster allows for representing each neuron's $\mathrm{CP}(p_{\mathrm{CR}})$-dependency in the two-dimensional space (*Figure 4C*) spanned by the mean profiles for each of the three clusters. The horizontal axis corresponds to the separation between the two initial clusters, and is closely aligned to the departure of the average CP from 0.5. The vertical axis is defined by the vectors corresponding to the centers of the two subclusters and hence is determined separately for the cells with average CP higher and lower than 0.5 (see Methods for details, and *Appendix 3—figure 1A*). The vertical axis is associated with the degree to which the $\mathrm{CP}(p_{\mathrm{CR}})$ dependence is symmetric or asymmetric with respect to $p_{\mathrm{CR}} = 0.5$. Cells for which the CP increases consistently with its preferred direction of motion coherence lie on the upper half-plane. To further support this interpretation of the axis, we repeated the clustering procedure replacing the nonparametric $k$-means procedure with a parametric procedure that defines the subclusters with a symmetric and an asymmetric template, respectively. The data is distributed approximately equally in both spaces (*Figure 4C–D*).

Similar results were also obtained when increasing the number of clusters non-hierarchically. Introducing a third cluster for all cells leaves almost unaltered the cluster of cells with CP lower than 0.5 (*Appendix 3—figure 1B*). The cluster of cells with CP higher than 0.5 splits into two subclusters analogous to the ones found from cells with CP higher than 0.5 alone. The distinction between cells with more symmetric and asymmetric $\mathrm{CP}(p_{\mathrm{CR}})$ dependencies is robust to the selection of a larger number of clusters, that is, clusters with this type of dependencies remain large when allowing for the discrimination of more patterns (*Appendix 3—figure 1C*). However, we do not mean to claim that the variety of $\mathrm{CP}(p_{\mathrm{CR}})$ profiles across cells can be reduced to three separable clusters. As reflected in the distributions in *Figure 4C–D*, the clusters are not neatly separable. Indeed, a richer variety of profiles would be expected if the properties of $\mathrm{CP}(p_{\mathrm{CR}})$ profiles across cells were associated with their tuning properties and the structure of feedback projections, as we further argue in the Discussion. The predominance of a symmetric and asymmetric pattern would only reflect which are the predominant $\mathrm{CP}(p_{\mathrm{CR}})$ shapes shared across cells.

This clustering analysis confirms the presence of shared patterns of CP stimulus-dependence across cells, whose shape is compatible with the analytical predictions from the threshold- and gain-related dependencies. The symmetric component of CP stimulus dependence is congruent with $h(p_{\mathrm{CR}})$ (*Equation 6*), albeit with a larger magnitude than predicted (*Figures 2A* and *3A*, and additional analysis of the statistical power in Appendix 1). This stronger modulation suggests an additional symmetric contribution of the choice correlation $\mathrm{CC}(p_{\mathrm{CR}})$ and/or a dynamic feedback reinforcing the stronger modulation for highly informative stimuli. However, while the cluster analysis separates the predominant $\mathrm{CP}(p_{\mathrm{CR}})$ patterns, the Britten et al. data lacks the statistical power to further distinguish between $h(p_{\mathrm{CR}})$ and symmetric $\mathrm{CC}(p_{\mathrm{CR}})$ contributions with a similar shape.

## Gain-induced CP stimulus dependencies in the MT responses

Three key features of the $\mathrm{CP}(p_{\mathrm{CR}})$ dependencies observed for the MT cells are qualitatively explained by introducing shared gain fluctuations in the decision threshold model described above (*Figure 4E*) – the first two manifesting itself on the population (cluster) level and the third one on an individual neuron level. First, a shared gain variability predicts the existence of the asymmetric CP stimulus dependence seen in cluster 3 (*Equation 11* and *Figure 2C*). Second, the average CP of the asymmetric cluster 3 is lower than the average CP of the symmetric cluster 2 (compare red and green profiles in *Figure 4B+E*). And third, if gain variability is indeed a driving factor for the observed asymmetry in cluster 3, then within this cluster, neurons with a higher amount of gain variability should also have a steeper $\mathrm{CP}(p_{\mathrm{CR}})$ profile, a prediction we could confirm as described in the next paragraph.

In order to test this prediction, for each neuron in cluster 3, we first computed the degree of asymmetry of its $\mathrm{CP}(p_{\mathrm{CR}})$ profile from the data directly, by simply fitting a quadratic function to $\mathrm{CP}(p_{\mathrm{CR}})$ (Methods). Next, and independently of this, we used the method of *Goris et al., 2014* to estimate the amount of gain variability for each neuron. Knowing each neuron's gain variability allowed us to predict each neuron's degree of asymmetry (slope of $\mathrm{CP}(p_{\mathrm{CR}})$ as determined by $\beta_{p_{\mathrm{CR}}}$, using *Equation 11*). We indeed found a significant correlation between the predicted and the observed slopes ($r = 0.58$, $p = 0.0018$) supporting the conclusion that shared gain variability underlies

the observed asymmetric shape of $\mathrm{CP}(p_{\mathrm{CR}})$ for the neurons in cluster 3. For cluster 2, in which the symmetric pattern is predominant, no analogous correlation was found ($r = 0.15$, $p = 0.35$). It is important to note that the asymmetry predicted by the gain variability overestimates the actually observed one by an order of magnitude (average observed slope of $0.002 \pm 0.0003$ compared to an average predicted slope of $0.034 \pm 0.008$). However, this is not surprising given our simplifying assumption of a single global gain factor across the whole population whereas in practice the gain fluctuations are likely inhomogeneous across the population. Furthermore, the actual read-out used by the brain may deviate from the optimal one, further reducing the expected match between predictions and observations. A more precise modeling of CP–stimulus dependencies would require measurements of the cross-neuronal correlation structure that is not available from the single unit recordings of *Britten et al., 1996* but will be for future population recordings.

## Modeling stimulus-dependent choice-related signals with GLMs

The implications of a stimulus-dependent relationship between the behavioral choice and sensory neural responses are not restricted to measuring them as CPs, for which activity-choice covariations are quantified without incorporating other explanatory factors of neural responses. To further substantiate the existence of this stimulus-dependent relationship in MT data, and to understand how our model predictions could help to refine other analytical approaches, we examined how representing that relationship can improve statistical models of neural responses. In particular, we study how the stimulus-dependent choice-related signals that we discovered may inform the refinement of Generalized Linear Models (GLMs) of neural responses (*Truccolo et al., 2005*; *Pillow et al., 2008*). In the last few years, GLMs have been used for modelling choice dependencies together with the dependence on other explanatory variables, such as the external stimulus, response memory, or interactions across neurons (*Park et al., 2014*; *Runyan et al., 2017*). Typically, in a GLM of firing rates each explanatory variable contributes with a multiplicative factor that modulates the mean of a Poisson process. In their classical implementation, the choice modulates the firing rate as a binary gain factor, with a different gain for each of the two choices (*Park et al., 2014*; *Runyan et al., 2017*; *Pinto et al., 2019*). The multiplicative nature of this factor already introduces some covariation between the impact of the choice on the rate and the one of the other explanatory variables. However, using a single regression coefficient to model the effect of the choice on the neural responses may be insufficient if choice-related signals are stimulus dependent, as suggested by our theoretical and experimental analysis.

We developed a GLM (see Methods) that can model stimulus-dependencies of choice signals (or, in other words, stimulus-choice interactions) by including multiple choice-related predictors that allow for a different strength of dependence of the firing rate on the choice for different subsets of stimulus levels (via the choice rate, $p_{\mathrm{CR}}$). We fitted this model, which we call the stimulus-dependent-choice GLM, to MT data and we compared its cross-validated performance against two traditional GLMs. In the first type, called the stimulus-only GLM, the rate in each trial is predicted only based on the external stimulus level. In a second type, that we called stimulus-independent-choice GLM and that corresponds to the traditional way to include choice signals in a GLM (*Park et al., 2014*; *Runyan et al., 2017*; *Scott et al., 2017*; *Pinto et al., 2019*; *Minderer et al., 2019*), additionally the effect of choice is included, but using only a single, stimulus-independent choice predictor.

To compare the models, we separated the trials recorded from each MT cell (*Britten et al., 1996*) into training and testing sets, and calculated the average cross-validated likelihood for each type of model on the held-out testing set. To quantify the increase in predictability when adding the choice as a predictor we defined the relative increase in likelihood (RIL) as the relative increase of further adding the choice as a predictor relative to the increase of previously adding the stimulus as a predictor. RIL measures the relative influence of the choice and the sensory input in the neural responses. *Figure 5A* compares the cross-validated RIL values obtained on MT neural data when fitting either the stimulus-independent-choice or the stimulus-dependent-choice GLMs. We found that RIL values were mostly higher when allowing for multiple choice parameters, both in terms of average RIL values (*Figure 5C*) and in terms of the proportion of cells in each cluster for which the RIL was higher than a certain threshold, here selected to be at 10% (*Figure 5B*).

GLMs that include stimulus-choice interaction terms can be used not only to better describe the firing rate of neural responses, but also to individuate more precisely the neurons or areas by their choice signals. To illustrate this point, we show how adding the interaction terms may change the

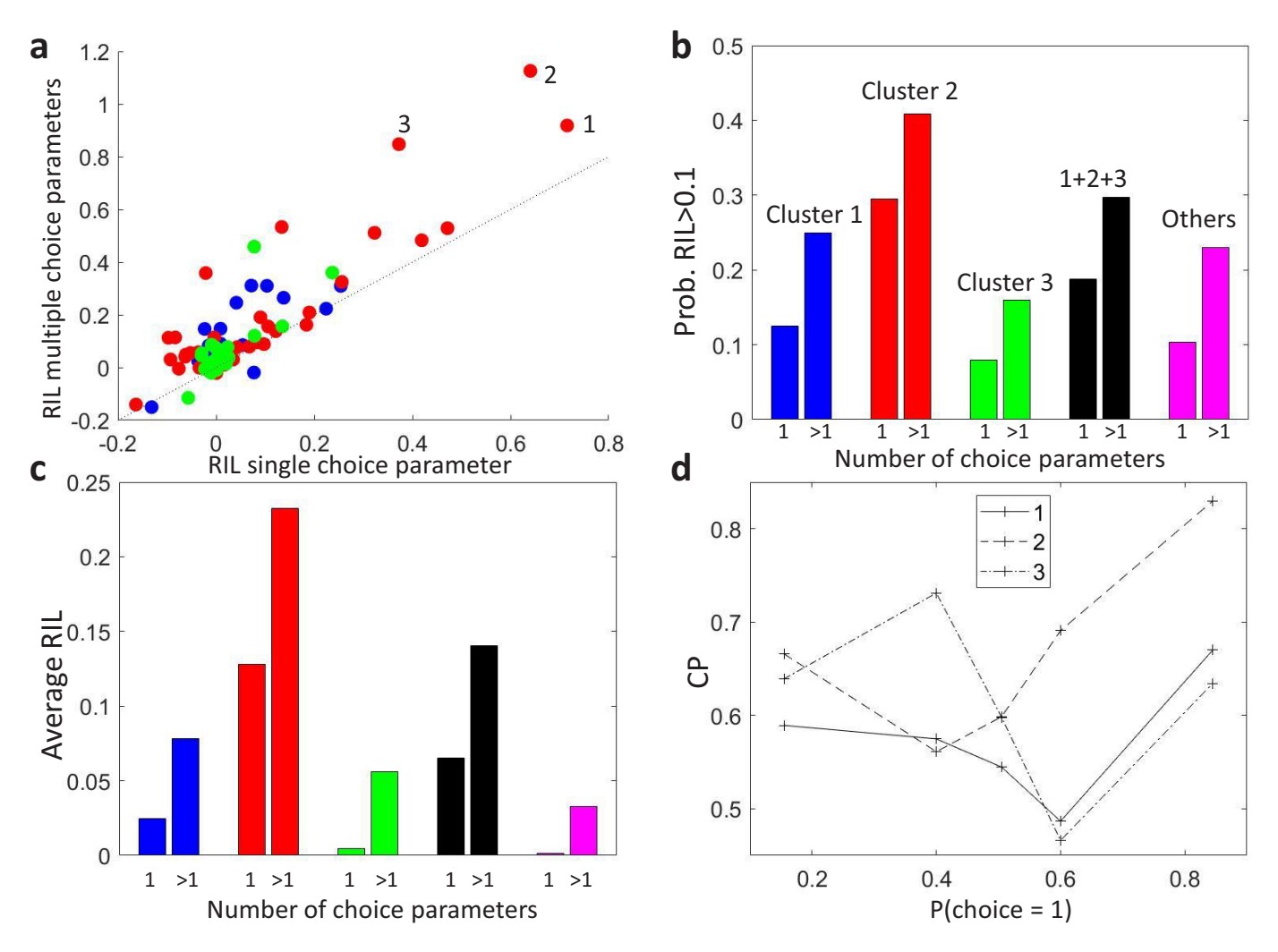

**Figure 5.** Modeling stimulus-dependent choice-related signals with GLMs. (a) Scatter plot of the cross-validated relative increase in likelihood (RIL), with respect to a stimulus-only model, of the stimulus-dependent-choice GLMs (multiple choice parameters) versus the stimulus-independent-choice GLMs (a single choice parameter). (b) Proportion of cells with RIL>0.1 for the two types of models, grouped by the clusters as in *Figure 4B*. Cells not included in the set of 107 cells for which a CP value could be estimated for each bin of $p_{CR}$ are labeled as 'Others'. (c) Average RIL values, grouped as in b. (d) $\mathrm{CP}(p_{CR})$ profiles of the three cells with the highest RIL in the stimulus-dependent-choice GLMs, as numbered in panel a.

relative comparison of cells by their RIL values. Consider the three neurons with highest RIL for the stimulus-dependent-choice GLM (*Figure 5A*, and with corresponding $\mathrm{CP}(p_{CR})$ profiles shown in *Figure 5D*). The ranking of cells 1 and 2 by RIL flips with respect to the stimulus-independent-choice GLM because of the higher $\mathrm{CP}(p_{CR})$ modulation of cell 2. Similarly, while the RIL with multiple choice parameters for cells 1 and 3 are close, the RIL of cell 3 is substantially lower with a single choice parameter, indicating that its pattern of stimulus dependence is less well captured by a single parameter. The degree to which a model with interaction terms improves the predictability will depend on the shape of the $\mathrm{CP}(p_{CR})$ patterns, which themselves are expected to vary across areas or across cells with different tuning properties. For example, we see in *Figure 5C* that for the cluster with an asymmetric $\mathrm{CP}(p_{CR})$ profile (cluster 3), the average RIL with only one choice parameter suggests that this type of cells are not choice driven. The reason is that for the cells in this cluster the *sign* of the choice influence on the rate can be stimulus dependent, which is impossible to model by a single choice parameter. Furthermore, the profile of the GLM choice parameters across stimulus levels provides a characterization of stimulus-dependent choice-related signals analogous to the $\mathrm{CP}(p_{CR})$ profile, in this case within the GLM framework, hence allowing efficient inference including

principled regularization and the ability to account for a range of factors beyond choices and stimuli. Overall, we expect that accounting for stimulus-choice interactions in GLMs will allow for a more accurate assessment of the relative importance of stimulus and choice on neural responses.

## Discussion

Our work makes several contributions to the understanding of how choice and stimulus signals in neural activity are coupled. The first is that we derived a general analytical model of perceptual decision-making predicting how the relationship between sensory responses and choice should depend on stimulus strength, regardless of whether this relationship is due to feedforward or feedback choice-related signals. The key model assumption is that the link between sensory responses and choices is mediated by a continuous decision variable and a thresholding mechanism. Second, we designed new, more powerful methods to measure within-cell dependencies of choice probabilities (CPs) on stimulus strength. Third, we studied CP stimulus dependencies in the classic dataset by *Britten et al., 1996*. Interestingly, we found a rich and previously unknown structure in how CPs in MT neurons depend on stimulus strength. In addition to a symmetric dependence predicted by the thresholding operation, we found an asymmetric dependence which we could explain by incorporating previously proposed gain fluctuations (*Goris et al., 2014*) in our model, thereby introducing a stimulus-dependent component in the cross-neuronal covariance. Finally, we showed that generalized linear models (GLMs) that account for stimulus-choice interactions better explain sensory responses in MT and allow for a more accurate characterization of how stimulus-driven and how choice-driven a cell's response is.

### Advances on analytical solutions of choice probabilities

Previous work has demonstrated that solving analytically models of perceptual decision-making can lead to important new insights on the interpretation of the relationship between neural activity and choice in terms of decision-making computations (*Bogacz et al., 2006*; *Gold and Shadlen, 2007*; *Haefner et al., 2013*). In particular, previous analytical work on CPs has shown how experimentally measured CPs relate to the read-out weights by which sensory neurons contribute to the internal stimulus decoder in a feedforward model, assuming both choices are equally likely (*Haefner et al., 2013*; *Pitkow et al., 2015*). Here, we provided a general analytical solution of CPs in a more general model, with informative stimuli resulting in an unbalanced choice rate, and valid both for feedforward and feedback choice signals. We derived the analytical dependency of CP on the probability of one of the choices ($p_{CR} \equiv p(\text{choice} = 1)$), which mediates the dependence of the CP on the stimulus strength. Our model is therefore directly applicable to both discrimination and detection tasks, for any stimulus strength that elicits both choices. As we demonstrated, these advances in the analytical solution of the decision-threshold model allowed for detecting and interpreting stimulus dependencies of choice-related signals in neural activity.

### Characterization of patterns of choice probability stimulus-dependencies from sensory neurons

Characterizing within-cell stimulus dependencies of activity-choice covariations at the population level requires isolating these dependencies from across-cells heterogeneity in the magnitude of the CP values. Our analytical analysis suggests possible reasons why previous attempts failed to find stimulus dependencies of CPs in real neural data. First, the magnitude of the CP dependence on $p_{CR}$ is proportional to the magnitude of choice-related signals (i.e. on how different CPs are from 0.5). This implies that neuron-specific dependencies need to be characterized for each cell individually, relative to the CP obtained with the uninformative stimulus. Only neurons for which a full individual CP profile can be estimated should be averaged to determine stimulus dependencies at the population level, or otherwise the overall average CP profile of stimulus dependence will be dominated by variability associated with the different subsets of neurons contributing to the CP estimate at each stimulus level. Second, the threshold-induced predicted direction of CP dependence on $p_{CR}$ is different for neurons with CP larger or smaller than 0.5, that is, neurons more responsive to opposite choices. This opposite modulation can cancel out the magnitude of the overall threshold-induced dependence of the CP on stimulus strength when averaging over all neurons, as done in previous

analyses (*Britten et al., 1996*). Informed by these insights we characterized the within-cell dependencies of choice-related signals on stimulus strength. The application of our refined methods to the classic neural data from MT neurons during a perceptual decision-making task of *Britten et al., 1996* allowed us to find stimulus dependencies of CPs, while previous analyses had not detected a significant effect.

Our understanding of how CP-stimulus dependencies may arise within the decision-making process, and the methods we used to measure these dependencies in existing data, will allow future studies to perform more fine-grained analyses and interpret more appropriately choice-related signals. Traditional analyses computed a single CP value for each neuron by either concentrating on zero-signal trials (e.g. *Dodd et al., 2001*; *Parker et al., 2002*; *Krug et al., 2004*; *Wimmer et al., 2015*; *Katz et al., 2016*; *Wasmuht et al., 2019*) or calculating grand CPs (*Britten et al., 1996*) across stimulus levels (e.g. *Cai and Padoa-Schioppa, 2014*; *Verhoef et al., 2015*; *Latimer et al., 2015*; *Pitkow et al., 2015*; *Smolyanskaya et al., 2015*; *Liu et al., 2016*; *Bondy et al., 2018*). Grand CPs are calculated directly as a weighted average of the CPs estimated for each stimulus level, or by pooling the responses from trials of all stimulus levels, after subtracting an estimate of the stimulus-related component (*Kang and Maunsell, 2012*). Our theoretical CP analysis shows that the latter procedure also corresponds to a specific type of weighted average (Appendix 2). Using the so computed individual CP values for each cell, areas or populations were then often ranked in terms of their averaged CP values per neurons. Areas with higher CP values are then identified as areas key for decision-making (e.g. *Nienborg and Cumming, 2006*; *Cai and Padoa-Schioppa, 2014*; *Pitkow et al., 2015*; *Yu et al., 2015*).

However, if CPs depend on $p_{\mathrm{CR}}$, it is clear that a single grand CP value cannot summarize this dependence. The use of average single CPs may thus introduce confounds in their comparison and miss important cell-specific information. For example, $\mathrm{CP}(p_{\mathrm{CR}})$ patterns with different sign for different $p_{\mathrm{CR}}$ values will result in lower average CP values. Similarly, the comparison of the grand CP of a cell across tasks may mostly reflect changes in the sampling in each task of stimulus levels, leading to a change in how much the CP(s) associated with each stimulus level contributes to the grand CP. As a result, the change in the grand CP may be interpreted as indicating the existence of task-dependent choice-related signals, even if the CP(s) profile is invariant. In the same way, if the structure of $\mathrm{CP}(p_{\mathrm{CR}})$ patterns covaries with the tuning properties, the comparison of the grand CP across cells with different tuning properties may mostly depend on the sampling of stimulus levels. This limitation is not specific to average CP values, and applies to other measures that consider choice-related and stimulus-driven components of the response as separable, such as partial correlations (e.g. *Zaidel et al., 2017*). Our work instead indicates that the shape of the $\mathrm{CP}(p_{\mathrm{CR}})$ patterns cannot be summarized in the average, and this shape may be informative about the role of the activity-choice covariations, when comparing across cells with different tuning properties, cells from different areas, or across tasks (e.g. *Romo and Salinas, 2003*; *Nienborg and Cumming, 2006*; *Nienborg et al., 2012*; *Krug et al., 2016*; *Sanayei et al., 2018*; *Shushruth et al., 2018*; *Jasper et al., 2019*; *Steinmetz et al., 2019*). Our new methods allow quantifying these CP patterns to better characterize the covariations between neural activity and choice across neurons and populations.

A key novelty introduced in our study is the development of a model-inspired methodological procedure for identifying genuine within-cell $\mathrm{CP}(p_{\mathrm{CR}})$ profiles, that would otherwise be masked by across-cells heterogeneity in the magnitude of the CP values. As representative examples of how our procedure may find previously unnoticed patterns of CP dependencies, we discuss the previous analyses in *Britten et al., 1996* and in *Dodd et al., 2001*. *Britten et al., 1996* analyzed the dependence of the CP on the stimulus strength at the population level (see their Figure 3). In particular, for each stimulus level they averaged the CP of all cells for which an estimate of the CP was calculated, without separating cells with CP higher or lower than 0.5. Furthermore, in their data set, the stimulus levels vary across cells, and hence in their analysis different subsets of cells contribute to the CP average at each stimulus level. *Dodd et al., 2001* presented a scatter plot of the CPs for all cells and stimulus levels (see their Figure 6). Although this analysis did not average cells with $\mathrm{CP}{>}0.5$ and $\mathrm{CP}{<}0.5$, in the scatter plot the cell-identity of each dot is not represented. This means that it is not possible to trace the within-cell $\mathrm{CP}(s)$ profiles. Like in the case of Britten et al., also in *Dodd et al., 2001* the sampled stimulus levels varied across cells, further confounding the within-cell $\mathrm{CP}(s)$ profiles with heterogeneity of CP magnitudes across cells. As shown by our analysis of the

data of *Britten et al., 1996*, our analytical tools can add extra discoveries from these data, by removing some potential confounds that may have obscured the presence of within-cell CP patterns. It is important to note however that our model-based results do not imply in any way that these previous papers reached to inaccurate conclusions, as these analyses were done for purposes other than discovering the within-cell patterns predicted by our models. In particular, most of the analysis of *Dodd et al., 2001* used only CPs calculated from trials with non-informative stimuli, and their main results did not rely on the evaluation of CP stimulus dependencies. Similarly, while *Britten et al., 1996* used z-scoring to calculate grand CPs combining all stimulus levels, their analysis did not involve the comparison of grand CPs across areas or types of cells with different tuning properties. As discussed above, it is for this kind of comparisons, when the patterns of $\mathrm{CP}(p_{\mathrm{CR}})$ profiles may themselves vary across the groups of cells compared, that reducing $\mathrm{CP}(p_{\mathrm{CR}})$ profiles to a single CP value may confound the comparison.

## Generalized linear models with stimulus-choice interactions

Our work has also implications for improving generalized linear models (GLMs) of neural activity, which are very widely used to describe neural responses in the presence of many explanatory variables that could predict the neuron's firing rate, such as the external stimulus, motor variables, autocorrelations or refractory periods, and the interaction with other neurons (*Truccolo et al., 2005*). While usually the stimulus and the choice are treated as separate explanatory variables (e.g. *Park et al., 2014*; *Runyan et al., 2017*; *Scott et al., 2017*; *Pinto et al., 2019*; *Minderer et al., 2019*), we used GLMs including explicit interactions between choice and stimulus to show that, consistently with the finding of non-constant $\mathrm{CP}(p_{\mathrm{CR}})$ patterns, these models improved the goodness of fit for the responses of MT cells. Importantly, making the choice term depend on the choice rate, $p_{\mathrm{CR}}$, affected the quantification of how stimulus-driven or choice-driven different cells are, quantified as the increased in predictive power when further adding the choice as a predictor after the stimulus. This suggests a more fine-grained way to compare the degree of a neuron's association with the behavioral choice or the stimulus, for example across neuron types or brain areas (*Runyan et al., 2017*; *Pinto et al., 2019*; *Minderer et al., 2019*). Our GLMs with multiple choice parameters associated with subsets of stimulus levels also allow characterizing the patterns in the vector of choice parameters analogously to our analysis of $\mathrm{CP}(p_{\mathrm{CR}})$-patterns. Furthermore, our approach can be extended straightforwardly to GLMs that model the influence of the choice across the time-course of the trials (*Park et al., 2014*), by making the stimulus-choice interaction terms time-dependent. GLMs with time-dependent stimulus-choice interaction terms can also be useful for experimental settings with multiple sensory cues presented at different times (e.g. *Romo and Salinas, 2003*; *Sanayei et al., 2018*) or a continuous time-dependent stimulus (*Nienborg and Cumming, 2009*), to account for a difference in the interaction of stimuli with the choice depending on the time they are presented. Similarly, the interaction terms may also help to model the influence of choice history in the processing of sensory evidence in subsequent trials (*Tsunada et al., 2019*; *Urai et al., 2019*), in which case the interaction terms would be between the stimulus and the choice from the previous trial.

## Patterns of stimulus-choice interactions as a signature of mechanisms of perceptual decision-making

Theoretical and experimental evidence suggests that the patterns of stimulus dependence of choice-related signals may be informative about the mechanisms of perceptual decision-making. Activity-choice covariations have been characterized in terms of the structure of cross-neuronal correlations and of feedforward and feedback weights (*Shadlen et al., 1996*; *Cohen and Newsome, 2009b*; *Nienborg and Cumming, 2010*; *Haefner et al., 2013*; *Cumming and Nienborg, 2016*). Stimulus dependencies may be inherited from the dependence of cross-neuronal correlations on the stimulus (*Kohn and Smith, 2005*; *Ponce-Alvarez et al., 2013*), or from decision-related feedback signals (*Bondy et al., 2018*). Experimental (*Nienborg and Cumming, 2009*; *Cohen and Maunsell, 2009a*; *Bondy et al., 2018*), and theoretical (*Lee and Mumford, 2003*; *Maunsell and Treue, 2006*; *Wimmer et al., 2015*; *Haefner et al., 2016*; *Ecker et al., 2016*) work indicates that top-down modulations of sensory responses play an important role in the perceptual decision-making process. In particular, feedback signals are expected to show cell-specific stimulus dependencies associated

with the tuning properties (*Lange and Haefner, 2017*). Different coding theories attribute different roles to the feedback signals, for example, conveying predictive errors (*Rao and Ballard, 1999*) or prior information for probabilistic inference (*Lee and Mumford, 2003*; *Fiser et al., 2010*; *Haefner et al., 2016*; *Tajima et al., 2016*; *Bányai and Orbán, 2019*, *Bányai et al., 2019*; *Lange and Haefner, 2020*). Accordingly, characterizing the stimulus dependencies of activity-choice covariations in connection with the tuning properties of cells is expected to provide insights into the role of feedback signals and may help to discriminate between alternative proposals. Such an analysis would require simultaneous recordings of populations of neurons tiling the space of receptive fields, and the joint characterization of the cross-neuronal correlations and tuning properties. Although this is beyond the scope of this work, we have shown that the analysis methods we proposed are capable of identifying a nontrivial structure in the stimulus-dependencies of choice-related signals. A better understanding of their differences across brain areas, across cells with different tuning properties, or for different types of sensory stimuli, promises further insights into the mechanisms of perceptual decision-making.

While we here analyzed single-cell recordings, our conclusions hold for any type of recordings used to study activity-choice covariations. This spans the range from single units (*Britten et al., 1996*), multiunit activity (*Sanayei et al., 2018*), and measurements resulting from different imaging techniques at different spatial scales like intrinsic imaging or fMRI (*Choe et al., 2014*; *Thielscher and Pessoa, 2007*; *Runyan et al., 2017*; *Michelson et al., 2017*). Given the increasing availability of population recordings, larger number of trials due to chronic recordings, and the advent of stimulation techniques to help to discriminate the origin of the choice-related signals (*Cicmil et al., 2015*; *Tsunada et al., 2016*; *Yang et al., 2016*; *Lakshminarasimhan et al., 2018*; *Fetsch et al., 2018*; *Yu and Gu, 2018*), we expect our tools to help gain new insights into the mechanisms of perceptual decision-making.

## Materials and methods

We here describe the derivations of the CP analytical solutions, our new methods to analyze stimulus dependencies in choice-related responses, and we describe the data set from *Britten et al., 1996* in which we test the existence of stimulus dependencies.

### An exact CP solution for the threshold model

We first derive our analytical CP expression valid in the presence of informative stimuli, decision-related feedback, and top-down sources of activity-choice covariation, such as prior bias, trial-to-trial memory, or internal state fluctuations. We follow *Haefner et al., 2013* and assume a threshold model of decision making, in which the choice $D$ is triggered by comparing a decision variable $d$ with a threshold $\theta$, so that if $d > \theta$ choice $D = 1$ is made, and $D = -1$ otherwise. The identification of the binary choices as $D = \pm 1$ is arbitrary and an analogous expression would hold with another mapping of the categorical variable. The choice probability (*Britten et al., 1996*) of cell $i$ is defined as

$$\mathrm{CP}_i = p(r_{i|D=1} > r_{i|D=-1}) = \int_{-\infty}^{\infty} \mathrm{d}r_i\, p(r_i|D=1) \int_{-\infty}^{r_i} \mathrm{d}r_i'\, p(r_i'|D=-1) \tag{12}$$

and measures the separation between the two choice-specific response distributions $p(r_i|D=-1)$ and $p(r_i|D=1)$. It quantifies the probability of responses to choice $D=1$ to be higher than responses to $D=-1$. If there is no dependence between the choice and the responses this probability is $\mathrm{CP} = 0.5$. To obtain an exact solution of the CP, we assume that the distribution $p(r_i, d)$ of the responses $r_i$ of cell $i$ and the decision variable $d$ can be well approximated by a bivariate Gaussian. Under this assumption, following *Haefner et al., 2013* (see their Supplementary Material) the probability of the responses for choice $D = 1$ follows the distribution

$$p(z_i|D=1) = \frac{1}{p_{\mathrm{CR}}} \phi(z_i; 0, 1) \Phi\left(\frac{\langle d \rangle + \frac{\sigma_{r_i,d}}{\sigma_{r_i}} z_i - \theta}{\sigma_{d|r_i}}; 0, 1\right), \tag{13}$$

where a more parsimonious expression is obtained using the z-score $z_i = (r_i - \langle r_i \rangle)/\sigma_{r_i}$. This distribution is a skew-normal (*Azzalini, 1985*), where $\phi(\cdot; 0, 1)$ is the standard normal distribution with zero

mean and unit variance, and $\Phi(\cdot;0,1)$ is its cumulative function. Furthermore, $\sigma_{r_i,d}$ is the covariance of $r_i$ and $d$, $\sigma_{d|r_i}$ is the conditional standard deviation of $d$ given $r_i$, and the probability of $D=1$ is

$$p_{\mathrm{CR}} \equiv p(d>\theta) = \Phi\left(\frac{\langle d\rangle - \theta}{\sigma_d}\right), \tag{14}$$

which determines the rate of each choice over trials. The choice $D=-1$ could equally be taken as the choice of reference, resulting in an analogous formulation. Intuitively, $p_{\mathrm{CR}}$ increases when the mean of the decision variable $\langle d\rangle$ is higher than the threshold $\theta$, and decreases when its standard deviation $\sigma_d$ increases. The form of the distribution of *Equation 13* can be synthesized in terms of $p_{\mathrm{CR}}$ and the correlation coefficient $\rho_{r_i d}$, which was named by *Pitkow et al., 2015* choice correlation (CC$_i$). In particular, defining $\alpha \equiv \rho_{r_i d}/\sqrt{1-\rho_{r_i d}^2}$ and $c \equiv \Phi^{-1}(p_{\mathrm{CR}})/\sqrt{1-\rho_{r_i d}^2}$

$$p(z_i|D=1) = \frac{1}{p_{\mathrm{CR}}}\phi(z_i;0,1)\Phi(\alpha z_i + c; 0,1). \tag{15}$$

The CP is completely determined by $p(z_i|D=-1)$ and $p(z_i|D=1)$, and these distributions depend only on $p_{\mathrm{CR}}$ and the correlation coefficient $\rho_{r_i d}$. Plugging the distribution of *Equation 15* into the definition of the CP (*Equation 12*) an analytical solution is obtained:

$$\mathrm{CP}_i = \frac{1}{2} + \frac{\mathrm{T}\left(\Phi^{-1}(p_{\mathrm{CR}}), \frac{\rho_{r_i d}}{\sqrt{2-\rho_{r_i d}^2}}\right)}{p_{\mathrm{CR}}(1-p_{\mathrm{CR}})}, \tag{16}$$

where $\mathrm{T}$ is the Owen's T function (*Owen, 1956*). In Appendix 1, we provide further details of how this expression is derived. For an uninformative stimulus ($p_{\mathrm{CR}}=0.5$), the function $\mathrm{T}$ reduces to the arctangent and the exact result obtained in *Haefner et al., 2013* is recovered. The dependence on $\rho_{r_i d}$ can be understood because under the Gaussian assumption the linear correlation captures all the dependence between the responses and the decision variable $d$. The dependence on $p_{\mathrm{CR}}$ reflects the influence of the threshold mechanism, which maps the dependence of $r_i$ with $d$ into a dependence with choice $D$ by partitioning the space of $d$ in two regions.

While *Equation 16* provides an exact solution of the CP, in the Results section we present and mostly focus on a linear approximation to understand how the stimulus content modulates the choice probability. This approximation is derived (Appendix 1) in the limit of a small $\rho_{r_i d}$, which leads to CPs close to 0.5 as usually measured in sensory areas (*Nienborg et al., 2012*). However, as we show in the Results and further justify in the Appendix this approximation is robust for a wide range of $\rho_{r_i d}$ values. The linear approximation relates the choice probability to the Choice Triggered Average (CTA) (*Haefner, 2015*; *Chicharro et al., 2017*), defined as the difference of the mean responses for each choice (*Equation 3*). Given the binary nature of choice $D$, the CTA is directly proportional to the covariance of the responses and the choice: $\mathrm{CTA}_i = \mathrm{cov}(r_i, D)/[2p(D=1)p(D=-1)]$. [Note: This relation holds for the covariance between any variable $x$ and a binary variable $D$, and independently of the convention adopted for the values of $D$: the factor 2 has to be replaced by $a-b$ in general for $D=a,b$ instead of $D=1,-1$.] This relation between $\mathrm{CTA}_i$ and $\mathrm{cov}(r_i, D)$, given the factorization $\mathrm{corr}(r_i, D) = \mathrm{CC}_i\mathrm{corr}(d, D)$ resulting from the mediating decision variable $d$ in the threshold model, allows expressing the $\mathrm{CTA}_i$ as in *Equation 5*, connecting $\mathrm{CTA}_i$ to the choice-triggered average of $d$, $\mathrm{CTA}_d$. This connection indicates that in the threshold model $\mathrm{CTA}_i$ is expected to be stimulus dependent, since an informative stimulus $s$ shifts the mean of $d$, thus altering the dichotomization of $d$ produced by the threshold $\theta$. The exact form of $\mathrm{CTA}_d$ depends on the distribution $p(d)$. However, since $d$ is determined by a whole population of neurons, its distribution is expected to be well approximated by a Gaussian distribution, even if the distribution of neural responses for any single neuron is not Gaussian. With this Gaussian approximation, the normalized $\mathrm{CTA}_d$ in *Equation 5*, namely $\mathrm{CTA}_d/\sqrt{\mathrm{var}\,d}$, is specified in terms of the probability of choosing choice 1, $p_{\mathrm{CR}} \equiv p(D=1) = p(d>\theta)$, by the factor $h(p_{\mathrm{CR}})$ (*Equation 6*). In more detail, the CTA is

$$\mathrm{CTA}_i \equiv \langle r_i \rangle_{D=1} - \langle r_i \rangle_{D=-1} = \frac{4h(p_{\mathrm{CR}})}{\sqrt{2\pi}} \rho_{r_i d} \sigma_{r_i} = \frac{4h(p_{\mathrm{CR}})}{\sqrt{2\pi}\sigma_d} \sigma_{r_i,d}. \tag{17}$$

## Neuronal data

To study stimulus dependencies in the relationship between the responses of sensory neurons and the behavioral choice, we analyzed the data from *Britten et al., 1996* publicly available in the Neural Signal Archive (http://www.neuralsignal.org). In particular, we analyzed data from file nsa2004.1, which contains single unit responses of macaque MT cells during a random dot discrimination task. This file contains 213 cells from three monkeys. We also used file nsa2004.2, which contains paired single units recordings from 38 sites from one monkey. In the experimental design, for the single unit recordings the direction tuning curve of each neuron was used to assign a preferred-null axis of stimulus motion, such that opposite directions along the axis yield a maximal difference in responsiveness (*Bair et al., 2001*). For paired recordings, the direction of stimulus motion was selected based on the direction tuning curve of the two neurons and the criterion used to assign it varied depending on the similarity between the tuning curves. For cells with similar tuning, a compromise between the preferred directions of the two neurons was made. For cells with different tuning, the axis were chosen to match the preference of the most responsive cell. To minimize the influence in our analysis of the direction of motion selection, we only analyzed the most responsive cell from each site. Accordingly, our initial data set consisted in a total of 251 cells. The same qualitative results were obtained when limiting the analysis to data from nsa2004.1 alone. Further criteria regarding the number of trials per each stimulus level were used to select the cells. As discussed below, if not indicated otherwise, we present the results from 107 cells that fulfilled all the criteria required.

## Analysis of stimulus-dependent choice probabilities

Our analysis of choice probabilities stimulus dependencies is based on examining the patterns in the $\mathrm{CP}(p_{\mathrm{CR}})$ profile as a function of the probability $p_{\mathrm{CR}} \equiv p(D=1)$. We here describe how these profiles are constructed, the surrogates-based method used to assess the significance of stimulus dependencies, and the clustering analysis used to identify different stimulus dependence patterns. Matlab functions are available at https://github.com/DanielChicharro/CP_DP (*Chicharro, 2021*; copy archived at swh:1:rev:5850c573860eb04317e7dc550f96b1f47ca91c6a) to calculate weighted average CPs, to obtain CP profiles, and to generate surrogates consistent with the null hypothesis of a constant CP.

### Profiles of CP as a function of the choice rate

We constructed $\mathrm{CP}(p_{\mathrm{CR}})$ profiles instead of $\mathrm{CP}(s)$ profiles based on the prediction from the theoretical threshold model of the modulatory factor $h(p_{\mathrm{CR}})$. We estimated the $p_{\mathrm{CR}}$ value associated with each random dots coherence level using the psychophysical function for each monkey separately. For each coherence level, we calculated a CP value if at least 15 trials were available in total, and at least four for each choice. In the original analysis of *Britten et al., 1996* stimulus dependencies $\mathrm{CP}(s)$ were examined averaging across cells the CP at each coherence level. This analysis did not separate the within-cell stimulus dependencies $\mathrm{CP}(s)$ from variability due to changes in choice probabilities across cells. In particular, in the data set the stimulus levels presented vary across cells, which means that for each coherence level the average CP does not only reflect any potential stimulus dependence of the CP but also which subset of cells contribute to the average at that level. Therefore, we binned the range of $p_{\mathrm{CR}}$ in a way that for each cell at least one stimulus level mapped to each bin of $p_{\mathrm{CR}}$. We here present the results using five bins defined as $[0-0.3, 0.3-(0.5-\varepsilon), (0.5-\varepsilon)-(0.5+\varepsilon), (0.5+\varepsilon)-0.7, 0.7-1]$, where $\varepsilon$ was selected such that only trials with the uninformative (zero coherence) stimulus were comprised in the central bin. Results are robust to the exact definition of the bins. We selected larger bins for highly informative stimulus levels for two reasons. First, the stimulus levels used in the experimental design do not uniformly cover the range of $p_{\mathrm{CR}}$, there are more stimulus levels corresponding to $p_{\mathrm{CR}}$ values close to $p_{\mathrm{CR}} = 0.5$. Second, the CP estimates are worse for highly informative stimuli. In particular, the standard error of the CP estimates depends on the magnitude of the CP itself (*Bamber, 1975*; *Hanley and McNeil, 1982*) but for small $|\mathrm{CP} - 0.5|$ can be approximated as

$$\text{SEM}(\hat{\text{CP}}) \approx 1/\sqrt{12 K p_{\text{CR}}(1 - p_{\text{CR}})}, \tag{18}$$

where $K$ is the number of trials. The product $p_{\text{CR}}(1 - p_{\text{CR}})$ is maximal at $p_{\text{CR}} = 0.5$ and decreases quadratically when $p_{\text{CR}}$ approximates 0 or 1. Furthermore, in the data set the number of trials $K$ is higher for stimuli with low information, while most frequently $K = 30$ for highly informative stimuli. We used these estimates of the $\hat{\text{CP}}$ error to combine the CPs of $M_k$ different stimulus levels assigned to the same bin $k$ of $p_{\text{CR}}$. The average $\text{CP}(p_{\text{CR},k})$ for bin $k$ was calculated as $\text{CP}(p_{\text{CR},k}) = \sum_j^{M_k} w_j \text{CP}(s_{jk})$ with normalized weights proportional to $\sqrt{K_j p_{\text{CR},j}(1 - p_{\text{CR},j})}$. A full profile $\text{CP}(p_{\text{CR}})$ could be constructed for 107 cells, while for the rest a CP value could not be calculated for at least one of the bins because of the criteria on the number of trials. Together with the profile $\text{CP}(p_{\text{CR}})$, we also obtained an estimate of its error as a weighted average of the errors, which corresponds to

$$\text{SEM}(\hat{\text{CP}}(p_{\text{CR},k})) = 1/(\sqrt{12 M_k} \langle w_{\text{U}} \rangle), \tag{19}$$

where $\langle w_{\text{U}} \rangle$ is the average of the unnormalized weights $w_{\text{U},j} \equiv \sqrt{K_j p_{\text{CR},j}(1 - p_{\text{CR},j})}$. Following this procedure, we can iteratively calculate weighted averages of the CPs across different sets. In particular, we used this same type of average to obtain averaged $\text{CP}(p_{\text{CR}})$ profiles across cells. Importantly, in contrast to the analysis of *Britten et al., 1996*, we previously separated the cells into two groups, with a positive or negative average $\text{CP} - 0.5$ value, given that the effect of $h(p_{\text{CR}})$ predicts an inverse modulation by $p_{\text{CR}}$.

## Surrogates to test the significance of CP stimulus dependencies

Given a certain average profile $\text{CP}(p_{\text{CR}})$, we want to assess whether the pattern observed is compatible with the null hypothesis of a constant CP value for all $p_{\text{CR}}$ values. In particular, because the error of the CP estimates is sensitive to the number of trials $K$ and to $p_{\text{CR}}$ (*Equation 18*), we want to discard that any structure observed is only a consequence of changes of $K$ and $p_{\text{CR}}$ across the bins used to calculate the $\text{CP}(p_{\text{CR}})$ profiles. For this purpose, we developed a procedure to build surrogate data sets compatible with the hypothesis of a flat $\text{CP}(p_{\text{CR}})$ and that preserves at each stimulus level the number of trials for each choice equal to the number in the original data. The surrogates are built shuffling the trials across stimulus levels to destroy any stimulus dependence of the CP. However, because the responsiveness of the cell changes across levels according to its direction tuning curve, responses need to be normalized before the shuffling. *Kang and Maunsell, 2012* showed that, to avoid underestimating the CPs, this normalization should take into account that mean responses at each level are determined by the conditional mean response for each choice and also by the choice rate. Under the assumption of a constant CP, they proposed an alternative z-scoring, which estimates the mean and standard deviation correcting for the different contribution of trials corresponding to the two choices (see Appendix 2 for details of their method).

We applied the z-scoring of *Kang and Maunsell, 2012* to pool the responses within an interval of stimulus levels with low information, preserving only the separation of trials corresponding to each choice. We selected the interval from $-1.6\%$ to $1.6\%$ of coherence values, which comprises a third of the informative coherence levels used in the experiments. Because these stimuli have low information they lead to $p_{\text{CR}}$ values close to $p_{\text{CR}} = 0.5$ and hence we can approximate the CP as constant within this interval. The fact that the factor $h(p_{\text{CR}})$ is almost constant around $p_{\text{CR}} = 0.5$ (see *Figure 2A*) further supports this approach. We used this pool of neural responses to sample responses for all stimulus levels in the surrogate data set. For each stimulus level of the surrogate data, the number of trials for each choice was preserved as in the original data. In these surrogates, apart from random fluctuations, any structure in the $\text{CP}(p_{\text{CR}})$ profiles can only be produced by the changes in $K$ and $p_{\text{CR}}$ across bins. To test the existence of significant stimulus dependencies in the original $\text{CP}(p_{\text{CR}})$ profiles we calculated the differences $\Delta \text{CP}_k = \text{CP}(p_{\text{CR},k+1}) - \text{CP}(p_{\text{CR},k})$ for the bins $k = 1, ..., 4$. To test for an asymmetric pattern with respect to $p_{\text{CR}} = 0.5$ the average of $\Delta \text{CP}_k$ across bins was calculated. To test for a symmetric pattern the sign of the difference was flipped for the bins corresponding to $p_{\text{CR}} < 0.5$ before averaging. When testing for a pattern consistent with the modulation predicted by the threshold model, the shape was inverted for cells with average CP lower than 0.5. The same procedure was applied to each surrogate $\text{CP}(p_{\text{CR}})$ profile. We generated

8000 surrogates and estimate the p-value as the number of surrogates for which the average $\Delta \mathrm{CP}$ was higher than for the original data.

## Clustering analysis

We used clustering analysis to examine the patterns in the $\mathrm{CP}(p_{\mathrm{CR}})$ profiles beyond the stereotyped shape $h(p_{\mathrm{CR}})$ predicted from the threshold model. We first used nonparametric $k$-means clustering for an exploratory analysis of which patterns are more common among the 107 cells for which a complete $\mathrm{CP}(p_{\mathrm{CR}})$ profile could be constructed. The clustering was implemented calculating cosine distances between vectors defined as $\mathrm{CP}(p_{\mathrm{CR}}) - 0.5$. The selection of this distance is consistent with the prediction of the threshold model that a different pattern is expected for cells with a CP higher or lower than 0.5. We examined the patterns associated with the clusters as a function of the number of clusters to identify robust patterns of dependence (see *Appendix 3—figure 1B,C*). We then focused on a symmetric and an asymmetric pattern of $\mathrm{CP}(p_{\mathrm{CR}})$ with respect to $p_{\mathrm{CR}} = 0.5$, for cells with average CP higher than 0.5. To better interpret these two clusters, we complemented the analysis with a parametric clustering approach in which a symmetric and asymmetric template were a priori selected to cluster the $\mathrm{CP}(p_{\mathrm{CR}})$ profiles. To assess the significance of the $\mathrm{CP}(p_{\mathrm{CR}})$ patterns we repeated the same clustering procedure for surrogate data generated as described above. We refer to Appendix 3 for a more detailed description of the construction, visualization, and significance assessment of the $\mathrm{CP}(p_{\mathrm{CR}})$ patterns.

## The effect of response gain fluctuations on choice probabilities

To model the effect on the CP of response gain fluctuations we adopted a classic feedforward encoding/decoding model (*Shadlen et al., 1996*; *Haefner et al., 2013*), with a linear decoder $d = \vec{w}^{\top}\vec{r}$ (*Equation 8*), for which the CP depends on cross-neuronal correlations and the read-out weights $\vec{w}$ following *Equation 9*. This expression can be derived from *Equation 7* directly calculating the choice correlation from its definition (*Equation 2*). The expressions $\mathrm{cov}(r_i, d) = (\Sigma(s)\vec{w})_i$ and $\sigma_d^2 = \vec{w}^{\top}\Sigma(s)\vec{w}$ are obtained as derived in the Supplementary Material S1 of *Haefner et al., 2013*. For this model, if the read-out weights are optimized to the form of covariability for the uninformative stimulus $s_0$ at the decision boundary, the CPs are proportional to the neurometric sensitivity of the cells (*Haefner et al., 2013*; *Pitkow et al., 2015*), a relationship for which there is some experimental support (e.g. *Britten et al., 1996*; *Parker et al., 2002*, reviewed in *Nienborg et al., 2012*). In more detail, modeling the responses as $r_i = f_i(s) + \xi_i$, with tuning functions $\vec{f}(s) = (f_1(s), ..., f_n(s))$ and a covariance structure $\Sigma$ of the neuron's intrinsic variability $\xi_i$, the optimal read-out weights have the form

$$\vec{w} = \frac{\Sigma^{-1}(s_0)\vec{f}'(s_0)}{\vec{f}'^{T}(s_0)\Sigma^{-1}(s_0)\vec{f}'(s_0)}, \tag{20}$$

where $\vec{f}'(s_0)$ and $\Sigma(s_0)$ are the derivative of the tuning curves and the responses covariance matrix, respectively, for $s = s_0$. With these optimal weights, the covariability of the population responses is unbundled, with $\Sigma^{-1}(s_0)$ canceling the effect of $\Sigma(s_0)$ in $\mathrm{cov}(r_i, d) = (\Sigma(s)\vec{w})_i$, and for each cell the CC is proportional to its own neurometric sensitivity $f_i'/\sigma_{r_i}$, namely

$$\mathrm{CC}_i(s_0) = \frac{f_i'(s_0)\sigma_d(s_0)}{\sigma_{r_i}(s_0)}. \tag{21}$$

While this expression is valid for the uninformative stimulus $s_0$, we examined how this CP expression is perturbed for other informative stimuli $s$ in the presence of gain fluctuations that make the covariance structure $\Sigma(s)$ stimulus-dependent, altering the structure for which the read-out weights are optimized. *Goris et al., 2014* estimated that in MT gain fluctuations accounted for more than 75% of the variance in the responses to sinusoidal gratings, and we found that in the data set of *Britten et al., 1996* gain fluctuations also explain a large fraction of trial-to-trial variability of the neurons (62 ± 25% across neurons). Trial-to-trial excitability fluctuations are modeled as a a gain modulatory factor $g_k$, such that the tuning function for cell $i$ in trial $k$, is $f_{ik}(s) = g_k f_i(s)$. In general, the magnitude of the gain may vary across cells, as well as the degree to which the gain co-fluctuates across cells. We here modeled a global gain fluctuation affecting the response of the whole

population. Given the gain variability, the covariance structure can be partitioned as in *Equation 10*, as the sum of a component $\bar{\Sigma}$ unrelated to the gain fluctuations – which for simplicity we consider to be stimulus-independent – and the gain-induced covariance $\sigma_G^2 \vec{f}(s)\vec{f}^T(s)$. In a first order approximation, a change $\Delta s = s - s_0$ in the stimulus leads to a change in the covariance structure such that

$$\Sigma(s) \approx \Sigma(s_0) + \sigma_G^2 [\vec{f}(s_0)\vec{f}'^T(s_0) + \vec{f}'(s_0)\vec{f}^T(s_0)]\Delta s, \tag{22}$$

where $\Sigma(s_0) = \bar{\Sigma} + \sigma_G^2 \vec{f}(s_0)\vec{f}^T(s_0)$ is the covariance structure for which the weights are optimized. Combining this covariance structure with the form of the optimal read-out weights (*Equation 20*), we derive the changes in $\mathrm{cov}(r_i, d)$, $\sigma_{r_i}^2$, and $\sigma_d^2$ with $\Delta s$, and given *Equation 9* determine the perturbation of the CP, leading to the following CP expressions

$$\mathrm{CC}_i(p_{\mathrm{CR}} = 0.5) = \sqrt{1 - \lambda_i^2}\,\mathrm{CC}_{i0}(p_{\mathrm{CR}} = 0.5) \tag{23a}$$

$$\mathrm{CC}_i(p_{\mathrm{CR}}) \approx \mathrm{CC}_i(p_{\mathrm{CR}} = 0.5)\left[1 - \beta_{\sigma_r}\Delta s(p_{\mathrm{CR}})\right] + \frac{\sigma_G^2 f_i(s_0)\Delta s(p_{\mathrm{CR}})}{\sqrt{\vec{w}^\top \Sigma(s_0)\vec{w}}\sqrt{\Sigma_{ii}(s_0)}}, \tag{23b}$$

where $\sigma_d^2(s_0) = \vec{w}^\top\Sigma(s_0)\vec{w}$, $\sigma_{r_i}^2(s_0) = \Sigma_{ii}(s_0)$, $\beta_{\sigma_r} \equiv \sigma_G^2 f_i(s_0)f_i'(s_0)/\sigma_{r_i}^2(s_0)$, and $\lambda_i^2 \equiv \sigma_{r_ig}^2(s_0)/\sigma_{r_i}^2(s_0)$, as introduced in *Equation 11*, with $s_0$ resulting in $p_{\mathrm{CR}} = 0.5$ for an unbiased decoder. *Equation 23a* relates the choice correlation for $p_{\mathrm{CR}} = 0.5$ with the choice correlation $\mathrm{CC}_{i0}(p_{\mathrm{CR}} = 0.5)$ that cell $i$ would have if there were no gain fluctuations ($\sigma_G^2 = 0$). The coefficient $\beta_{\mathrm{CC}} \equiv \sqrt{1 - \lambda_i^2}$ indicates a decrease in the CC in the presence of gain fluctuations, because of the increase in the response variability produced by the fluctuations, namely $\sigma_{r_ig}^2 = \sigma_G^2 f_i^2$. *Equation 23b* describes the $\mathrm{CC}(p_{\mathrm{CR}})$ profile induced by the gain fluctuations. The second summand corresponds to the increase in the choice correlation due to a new component of $\mathrm{cov}(r_i, d)$ proportional to $\Delta s$, given that the whole population response determining $d$ is jointly modulated by the gain. In the first summand, the factor $\left[1 - \beta_{\sigma_r}\Delta s(p_{\mathrm{CR}})\right]$ indicates an attenuation of $\mathrm{CC}_i(p_{\mathrm{CR}} = 0.5)$ analogous to $\sqrt{1 - \lambda_i^2}$ in *Equation 23a*, associated with an increase of the variance in the responses $r_i$ due to $\Delta s$. Rearranging the terms in *Equation 23b*, and taking into account the form of the CC for $p_{\mathrm{CR}} = 0.5$ (*Equation 21*), the expression in *Equation 11* is obtained, which indicates that the overall effect of the gain fluctuations is an increase of the choice correlation for the stimuli to which the cell is more responsive. A more general form of this expression is derived in Appendix 4, valid for any unbiased decoder.

Apart from producing an asymmetric $\mathrm{CP}(p_{\mathrm{CR}})$ pattern, the gain fluctuations also create a negative covariation between the CP at $p_{\mathrm{CR}} = 0.5$ and the degree of asymmetry of the $\mathrm{CP}(p_{\mathrm{CR}})$ pattern. This covariation appears because the cells with a higher portion of their variability driven by the gain fluctuations (higher $\lambda_i$) have a higher attenuation of $\mathrm{CC}_i(p_{\mathrm{CR}} = 0.5)$, given *Equation 23a*, while both a higher $\lambda_i$ and smaller $\mathrm{CC}_i(p_{\mathrm{CR}} = 0.5)$ lead to an increase in the slope $\beta_{p_{\mathrm{CR}}} \equiv \sigma_G\lambda_i(1 - \mathrm{CC}_i^2(p_{\mathrm{CR}} = 0.5))$ of the dependence on $\Delta s(p_{\mathrm{CR}})$ in *Equation 23b*. Furthermore, a smaller $\mathrm{CC}_i(p_{\mathrm{CR}} = 0.5)$ also leads to a smaller effect of the multiplicative symmetric modulation $h(p_{\mathrm{CR}})$, further contributing to the negative covariation between the magnitude of the CP and predominance of the symmetric or asymmetric pattern.

To illustrate the properties common to the model and to the $\mathrm{CP}(p_{\mathrm{CR}})$ patterns from the MT data, *Figure 4E* shows CPs from *Equation 23* as a function of $p_{\mathrm{CR}}$ for examples combining four values of $\mathrm{CC}_{i0}(p_{\mathrm{CR}} = 0.5)$ and two values of $\sigma_G^2$, while the other parameters of the cell responses are kept constant. In particular, to determine $\lambda_i^2$ only in terms of the strength of the gain we fixed the rate to $f_i(p_{\mathrm{CR}} = 0.5) = 10\,\mathrm{spike/s}$ and considered the variance not associated with the gain to be equal to that rate, so that $\lambda_i^2 = 1/(1 + 0.1/\sigma_G^2)$. Accordingly, the values of $\sigma_G^2$ in *Figure 4E*, namely $\sigma_G^2 = 0.1$ and $\sigma_G^2 = 0.01$, correspond to $\lambda_i^2 = 0.5$ and $\lambda_i^2 = 0.09$, respectively. Further analysis of the model is provided in Appendix 4, where we also discuss the form of the $\mathrm{CP}(p_{\mathrm{CR}})$ pattern produced by gain fluctuations when the decoder is composed by two pools of opposite choice preference (*Shadlen et al., 1996*).

To experimentally estimate the coefficients $\beta_{p_{\mathrm{CR}}}^{(exp)}$ we fitted a quadratic regression of the CPs on the stimulus levels. To theoretically estimate the coefficients $\beta_{p_{\mathrm{CR}}}^{(th)}$, we used the negative binomial

model of *Goris et al., 2014* to estimate $\sigma_G^2$ for each cell and used the form $\beta_{p_{\mathrm{CR}}} \equiv \sigma_G \lambda_i (1 - \mathrm{CC}_i^2(p_{\mathrm{CR}} = 0.5))$ predicted by the gain model (*Equation 11*) to estimate $\beta_{p_{\mathrm{CR}}}^{(th)}$.

## Generalized linear models modeling the interaction between stimulus and choice predictors

We implemented a new GLM, called stimulus-dependent-choice GLM, that includes regression coefficients quantifying the effect on the firing rate of interactions between stimulus and choice. This model of the firing rate of each neuron was compared to two simpler and traditional models: a stimulus-only GLM, which includes only stimulus predictors of the neuron's firing rate, and a stimulus-independent-choice GLM, which includes together with the stimulus predictor a single, stimulus-independent choice predictor.

In more detail, all three GLMs were Poisson models in which the mean firing rate $\mu(r_i)$ of cell $i$ was generally expressed by the following equation:

$$\log(\mu(r_i)) = \Sigma_{j=0}^{4} a_j s^j + \Sigma_{j=1}^{N_c} I_{P_j}(p_{\mathrm{CR}}) b_j D. \tag{24}$$

The terms $\Sigma_{j=0}^{4} a_j s^j$ are present in all three types of GLM, and model the stimulus influence with a fourth order polynomial function of the stimulus level. These are the only terms of the stimulus-only GLM.

The choice dependence is modeled by $\Sigma_{j=1}^{N_c} I_{P_j}(p_{\mathrm{CR}}) b_j D$, with the parameter $N_c$ ($N_c \in \{1, 2, 3\}$) setting the number of possible different levels of stimulus-dependent choice (we restricted the fitting to up to three different choice levels for simplicity, and because we found empirically this to work well for the MT data analyzed here). $I_{P_j}(p_{\mathrm{CR}})$ is an indicator function which equals one if a $p_{\mathrm{CR}}$ value belongs to the subset $P_j$ of values selected to be associated with the choice parameter $b_j$, and is zero otherwise. For the stimulus-independent-choice model, we set $N_c = 1$ so that the choice affects the predicted responses equally for all stimulus levels. For the stimulus-dependent-choice GLM, we set $N_c > 1$. For this stimulus-dependent-choice GLM, we determined the subsets of stimulus levels associated with each of those parameters using the $\mathrm{CP}(p_{\mathrm{CR}})$ profiles for a first characterization of the stimulus dependencies. Like for the CP analysis, for each cell we determined which coherence values could be included in the analysis given a criterion requiring a minimum number of trials for each choice (at least 4). The existence of non-monotonic $\mathrm{CP}(p_{\mathrm{CR}})$ profiles, such as the symmetric pattern around $p_{\mathrm{CR}} = 0.5$, indicated that it would be sub-optimal to tile the domain of $p_{\mathrm{CR}}$ with $N_p$ bins and assign a different choice-parameter level to each bin. Accordingly, we first estimated the $\mathrm{CP}(p_{\mathrm{CR}})$ profile of each cell and then used $k$-means clustering with an Euclidean distance to cluster the components of $\mathrm{CP}(p_{\mathrm{CR}})$, corresponding to the bins of $p_{\mathrm{CR}}$, into $N_c$ subsets. A different GLM choice-parameter $b_j$ was then assigned to each choice-parameter level $j = 1, ... N_c$.

We compared the predictive power of the three types of models using cross-validation. To avoid that the choice-parameters fitted were affected by the ratio of trials with each choice, we matched the number of trials of each choice used to fit the model at each choice-parameter level. We first merged in two pools, one for each choice separately, the trials of all stimulus levels assigned to the same choice-parameter level. We then determined the number of trials from each pool to be included in the fitting set as an 80% of the trials available in the smallest pool, hence matching the number of trials selected from each choice. The remaining trials were left for the testing set. This procedure was repeated for each choice-parameter level and a GLM model was fitted on the fitting set obtained combining the selected trials for all levels. This random separation between fitting and testing data sets was repeated 50 times and the average predictive power was calculated. Performance was then quantified comparing the increase in the likelihood of the data in the testing set with respect to the likelihood of the null model which assumes a constant firing rate ($L_0$). To determine if incorporating the choice as a predictor improved the prediction, we examined the relative increase in likelihood (RIL) defined as the ratio of the likelihood increase $L(choice, stimulus) - L(stimulus)$ and the increase $L(stimulus) - L_0$. For the stimulus-dependent-choice models, we selected the most predictive model from $N_c = 2, 3$. To evaluate the improvement when considering stimulus-dependent choice influences, we compared the RIL obtained for the stimulus-dependent-choice and stimulus-independent-choice models.

## Code availability

The codes for the analysis of Choice Probability stimulus-dependencies and GLMs with stimulus-choice interaction terms are available at https://github.com/DanielChicharro/CP_DP (copy archived at swh:1:rev:5850c573860eb04317e7dc550f96b1f47ca91c6a).

## Acknowledgements

This work was supported by the BRAIN Initiative (grants No. R01 NS108410 and No. U19 NS107464 to SP, U19 NS118246 to RMH), the Fondation Bertarelli, and the CRCNS program (grant R01 EY028811 to RMH). We thank KH Britten, WT Newsome, MN Shadlen, S Celebrini, and JA Movshon for making their data available in the Neural Signal Archive (http://www.neuralsignal.org/), and W Bair for maintaining this database.

## Additional information

### Funding

| Funder | Grant reference number | Author |
| --- | --- | --- |
| National Institute of Neurological Disorders and Stroke | R01 NS108410 | Stefano Panzeri |
| National Institute of Neurological Disorders and Stroke | U19 NS107464 | Stefano Panzeri |
| National Eye Institute | R01 EY028811 | Ralf M Haefner |
| Fondation Bertarelli | | Daniel Chicharro |
| National Institute of Neurological Disorders and Stroke | U19 NS118246 | Ralf M Haefner |

The funders had no role in study design, data collection and interpretation, or the decision to submit the work for publication.

### Author contributions

Daniel Chicharro, Conceptualization, Data curation, Formal analysis, Investigation, Visualization, Methodology, Writing - original draft; Stefano Panzeri, Conceptualization, Supervision, Funding acquisition, Writing - review and editing; Ralf M Haefner, Conceptualization, Formal analysis, Supervision, Funding acquisition, Writing - review and editing

### Author ORCIDs

Daniel Chicharro https://orcid.org/0000-0002-4038-258X
Stefano Panzeri https://orcid.org/0000-0003-1700-8909
Ralf M Haefner https://orcid.org/0000-0002-5031-0379

### Decision letter and Author response

Decision letter https://doi.org/10.7554/eLife.54858.sa1
Author response https://doi.org/10.7554/eLife.54858.sa2

## Additional files

### Supplementary files

• Transparent reporting form

### Data availability

No data was collected as part of this study.

The following previously published dataset was used:

| Author(s) | Year | Dataset title | Dataset URL | Database and Identifier |
|---|---|---|---|---|
| Britten KH, Newsome WT, Shadlen MN, Celebrini S, Movshon JA | 1996 | A relationship between behavioral choice and the visual responses of neurons in macaque MT | http://www.neuralsignal.org/index_data.html | The Neural Signal Archive, Macaque |

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

# Appendix 1

## The analytical threshold model of choice probability

We here provide details of how the CP analytical expression of *Equation 16* is obtained from the definition of Choice Probability (*Equation 12*) when the probability of the responses for each choice has the form of *Equation 15*, derived from the threshold model. We subsequently characterize the statistical power for the detection of the threshold-induced modulation $h(p_{\mathrm{CR}})$, as a function of the magnitude of the CP, the number of trials, and the number of cells used to estimate an average $\mathrm{CP}(p_{\mathrm{CR}})$ profile.

## Derivation of the CP analytical expression

Plugging the distribution of *Equation 15* into the definition of the CP we get

$$\mathrm{CP}_i = \frac{1}{2p_{\mathrm{CR}}(1-p_{\mathrm{CR}})}\left[1 - \alpha \int_{-\infty}^{\infty} \mathrm{d}x\phi(\alpha x + c)\Phi^2(x) - \left(\int_{-\infty}^{\infty} \mathrm{d}x\phi(x)\Phi(\alpha x + c)\right)^2\right]. \tag{A1}$$

This expression is derived analogously to Equation S1.2 in *Haefner et al., 2013*, and generalizes the case examined there, which corresponds to $c = 0$. To solve this expression, we use some results involving integrals of normal distributions:

$$\int_{-\infty}^{\infty} \mathrm{d}x\, x^q \phi(x)\Phi^n(\alpha x + c) = (1-q)\Phi\left(\frac{c}{b}\right) + q\frac{\alpha}{b}\phi\left(\frac{c}{b}\right)$$
$$+ 2(1-n)\mathrm{T}\left(\frac{c}{b}, \frac{1}{\sqrt{1+2\alpha^2}}\right), \tag{A2}$$

where $b = \sqrt{1+\alpha^2}$ and $\mathrm{T}$ is the Owen's T function (*Owen, 1956*). The equality above is valid for the cases $q = 0, n = 1, 2$, and $q = 1, n = 1$, which we used to derive the expressions of the CP and $\mathrm{CTA}$. Using the equality for $q = 0, n = 1, 2$ into *Equation (A1)* we obtain the CP expression of *Equation 16*. On the other hand, the case with $q = 1, n = 1$ allows deriving the expression of the CTA (*Equation 17*) calculating $\langle r_i \rangle_{D=1}$ and $\langle r_i \rangle_{D=-1}$ using the form of *Equation 15* for $P(r_i | D = 1)$.

The CP linear approximation of *Equation 4* is generically valid when the activity-choice covariations are well captured by the linear dependence between the responses and the choice. It can be generically derived with a first order approximation such that $d$ or $D$ only affect the mean of the distribution of the responses $r_i$. We here only present a restricted derivation, specifically from the exact CP solution resulting from the threshold model. It can be checked that the same approximation follows for example from the exact solution of the CP obtained when taking the conditional distributions $p(r_i | D)$ to be Gaussians (*Dayan and Abbot, 2001*; *Carnevale et al., 2013*) and not skew normals (*Equation 15*) like for the threshold model. Expanding *Equation 16* in terms of $\rho_{r_i d}$ we get a polynomial approximation

$$\mathrm{CP}_i = \frac{1}{2} + \frac{\sqrt{2}h(p_{\mathrm{CR}})}{\pi}\rho_{r_i d} + \sum_{k=1}^{\infty} \frac{\mathrm{d}^{2k}\left(\frac{\exp\left[-\frac{(\Phi^{-1}(p_{\mathrm{CR}}))^2}{2}\frac{2}{2-\rho_{r_i d}^2}\right]}{2\pi p_{\mathrm{CR}}(1-p_{\mathrm{CR}})\sqrt{2-\rho_{r_i d}^2}}\right)}{\mathrm{d}\rho_{r_i d}^{2k}}\Big|_{\rho_{r_i d}=0}\frac{\rho_{r_i d}^{2k+1}}{(2k+1)!}. \tag{A3}$$

This expansion contains only odd order terms, producing a symmetry of $\mathrm{CP} - 0.5$ with respect to the sign of $\rho_{r_i d}$. This explains why *Haefner et al., 2013* found that the linear approximation was accurate for a wide range of $\rho_{r_i d}$ values, since the choice correlation needs to be high so that the contribution of $\rho_{r_i d}^3$ is relevant. Up to order 3.

$$\mathrm{CP}_i \approx \frac{1}{2} + \frac{\sqrt{2}h(p_{\mathrm{CR}})}{\pi}\left[\rho_{r_i d} + \frac{1 - (\Phi^{-1}(p_{\mathrm{CR}}))^2}{12}\rho_{r_i d}^3\right]. \tag{A4}$$

Since $\Phi^{-1}(0.5) = 0$, for $p_{\mathrm{CR}}$ values for which $|1 - (\Phi^{-1}(p_{\mathrm{CR}}))^2| < 1$ the third order term makes a

smaller contribution than for the uninformative case. This is true for $\Phi^{-1}(p_{\mathrm{CR}}) \in (-\sqrt{2}, \sqrt{2})$, which leads to $p_{\mathrm{CR}} \in (0.08, 0.92)$. This means that the linear approximation is expected to be an even better approximation in this range than for $p_{\mathrm{CR}} = 0.5$. Furthermore, for $(\Phi^{-1}(p_{\mathrm{CR}}))^2 < 1$ the third order contribution is positive, so that for the $p_{\mathrm{CR}}$ values fulfilling this constraint, $p_{\mathrm{CR}} \in (0.16, 0.84)$, the linear approximation is expected to underestimate the CP. The range of $p_{\mathrm{CR}}$ in which the linear approximation underestimates or overestimates the CP can be seen in *Figure 2* of the main article.

## Statistical power for the detection of threshold-induced CP stimulus dependencies

In *Figure 2*, we showed the shape and magnitude of the threshold-induced CP modulation for different values of $\mathrm{CP}(p_{\mathrm{CR}} = 0.5)$. The magnitude of this modulation is small and is most noticeable for the extreme $p_{\mathrm{CR}}$ values, for which the estimation of the CPs is also the poorest. As discussed in the Results, we calculated weighted average CPs both across stimulus levels and across cells to reduce the standard error of the resulting averaged within-cell $\mathrm{CP}(p_{\mathrm{CR}})$ profiles. We here characterize the statistical power for the detection of this CP modulation as a function of the magnitude of $\mathrm{CP}(p_{\mathrm{CR}} = 0.5)$, the number of cells, and the number of trials per stimulus level. For this purpose, we generated responses following the probability distribution analytically derived in the model (*Equation 15*). We selected five $\mathrm{CP}(p_{\mathrm{CR}} = 0.5)$ levels, namely $\{0.55, 0.6, 0.65, 0.7, 0.75\}$, and for each we simulated cell responses corresponding to different $p_{\mathrm{CR}}$ values. We simulated responses for a collection of 5000 cells, selecting their $\mathrm{CP}(p_{\mathrm{CR}} = 0.5)$ values from a uniform distribution centered at each $\mathrm{CP}(p_{\mathrm{CR}} = 0.5)$ level, and a range of width 0.1. We repeated the same procedure using different numbers of trials. For each selection of the number of trials, we generated three times that number of trials for the $p_{\mathrm{CR}}$ bin corresponding to an uninformative stimulus, to allow the shuffling of trials used in the construction of surrogate data, as for the experimental data. We estimated averaged $\mathrm{CP}(p_{\mathrm{CR}})$ profiles for different numbers $N$ of cells. We repeated this estimation 500 times, independently randomly sampling from the 5000 cells the $N$ cells used for the average. For each of these repetitions, we generated the surrogate data required to implement the surrogates test described in Methods.

*Appendix 1—figure 1* shows the p-values obtained when testing for a symmetric increase of the CP with extreme $p_{\mathrm{CR}}$ values. As expected, p-values decrease with the number of cells, the number of trials, and the magnitude of the CP. This characterization of the p-values indicates the utility of our method to calculate averaged within-cell $\mathrm{CP}(p_{\mathrm{CR}})$ profiles, combining CP estimates across neighboring stimulus levels -analogously to increasing the number of trials- as well as averaging CP estimates across cells. We can compare these predicted p-values with the p-values obtained when analyzing the experimental data. Concretely, for cluster 2 in *Figure 4B*, which contains $N = 48$ cells, the average $\mathrm{CP}(p_{\mathrm{CR}} = 0.5)$ value is $\langle \mathrm{CP}(p_{\mathrm{CR}} = 0.5) \rangle = 0.57$ and, for each of the two $p_{\mathrm{CR}}$ bins most distant from $p_{\mathrm{CR}} = 0.5$, the average number of trials available is $\langle K \rangle = 147$, comprising all stimulus levels assigned to those bins. These values of $\langle \mathrm{CP}(p_{\mathrm{CR}} = 0.5) \rangle$, $N$, and $\langle K \rangle$ resulted in a p-value $p = 0.0008$ (*Figure 4B*), which is substantially smaller than what predicted in the simulations of *Appendix 1—figure 1A*. Accordingly, while the model correctly predicts the existence of a symmetric CP modulation with higher CP values for extreme $p_{\mathrm{CR}}$ values, it underestimates its actual strength. As discussed in Results, this stronger symmetric modulation may be due to other symmetric contributions from $\mathrm{CC}(p_{\mathrm{CR}})$ in addition to $h(p_{\mathrm{CR}})$, or to a dynamic amplification of the threshold-induced modulation, for example due to reinforcing decision-related feedback signals. Independently of the origin if this higher effect size, the method developed to better characterize within-cell $\mathrm{CP}(p_{\mathrm{CR}})$ dependencies allowed us to detect the actual presence of CP stimulus dependencies in the *Britten et al., 1996* data, and promises to be a useful tool to characterize the properties of these dependencies.

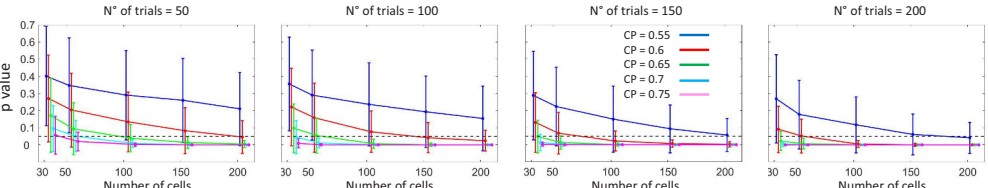

**Appendix 1—figure 1.** Statistical power for the detection of threshold-induced CP stimulus dependencies. For cell responses generated from the threshold model (see text for details), the figure characterizes the p-values obtained in the surrogates test used to assess the presence of a symmetric modulation of the CP, with increased CP values for unbalanced choice rates. Each panel presents the p-values as a function of the number of cells included to calculate an averaged $\mathrm{CP}(p_{\mathrm{CR}})$ profile, and of the $\mathrm{CP}(p_{\mathrm{CR}} = 0.5)$ value. Curves corresponding to different CP values are shifted for a better visualization of the error bars (standard deviation of the p-value across 500 simulations). The horizontal dashed line indicates the threshold of significance $p = 0.05$.

# Appendix 2

## The relation between the weighted average CP and the grand CP of z-scored responses

We here describe the connection between a weighted average CP and the corrected z-scoring procedure *Kang and Maunsell, 2012* used to calculate a grand CP pooling the responses across stimulus levels. We will use $z_c$ to refer to the responses with the corrected z-scoring, as opposed to the standard z-scoring $z$. For each stimulus level $s$, the corrected z-scoring is calculated as $z_c = (r - \tilde{\mu}_{r|s})/\tilde{\sigma}_{r|s}$ with

$$\tilde{\mu}_{r|s} \equiv \frac{\mu_{r|D=1,s} + \mu_{r|D=-1,s}}{2}, \qquad \tilde{\sigma}_{r|s} \equiv \sqrt{\frac{\sigma^2_{r|D=1,s} + \sigma^2_{r|D=-1,s}}{2} + \frac{\Delta\mu^2_{r|D,s}}{4}}, \tag{A5}$$

where $\mu_{r|D=\pm1,s}$ are the mean responses to each choice for the stimulus $s$, $\sigma^2_{r|D=\pm1,s}$ are the variance of the responses to each choice for the stimulus $s$, and $\Delta\mu_{r|D,s} = \mu_{r|D=1,s} - \mu_{r|D=-1,s}$ is the CTA for the fixed stimulus level $s$.

The relation between the CP and the CTA of *Equation 4* holds due to the definition of the measures and does not depend on the nature of the response variable, and hence also holds when the CP is not calculated for the raw responses but for their corrected z-scores. Furthermore, the relation holds when the CP is calculated for a fixed stimulus level, or when the responses are pooled across stimulus levels to calculate the grand CP. For the latter case

$$\text{grand CP}_{z_c} \approx \frac{1}{2} + \frac{1}{2\sqrt{\pi}} \frac{\Delta\mu_{z_c|D}}{\sigma_{z_c}}, \tag{A6}$$

where we drop the cell index $i$ for simplicity. We here use the notation $\Delta\mu_{z_c|D} = \text{CTA}_{z_c}$ and $\sigma_{z_c} = \text{var}\, z_c$, in comparison to *Equation 4*. Furthermore, in this section we need to explicitly differentiate which measures are calculated for a fixed stimulus, as indicated by the conditioning on $s$ in the subindex, and which measures are calculated from the distribution of the pooled responses, in which case there is no stimulus subindex, for example, for $\Delta\mu_{z_c|D}$. The grand $\text{CP}_{z_c}$ and $\Delta\mu_{z_c|D}$ are calculated from the distributions $p(z_c|D=\pm1)$ obtained after pooling, while for a fixed stimulus $\text{CP}_{z_c}(s)$ and $\Delta\mu_{z_c|D,s}$ are calculated from $p(z|D=\pm1,s)$. The choice rate $p_{\text{CR}}$ is defined as in the main text as $p_{\text{CR}} \equiv p(D=1|s)$, although previously the dependence on the fixed stimulus was implicit.

We use the definition of $z_c$ according to *Equation (A5)* to calculate $\Delta\mu_{z_c|D}$. Given the definition of $\tilde{\mu}_{r|s}$ and $\tilde{\sigma}_{r|s}$, we obtain that the conditional means $\mu_{z_c|D=1,s}$ and $\mu_{z_c|D=-1,s}$ are equal to $\mu_{z_c|D=1,s} = \Delta\mu_{r|D,s}/(2\tilde{\sigma}_{r|s})$ and $\mu_{z_c|D=-1,s} = -\Delta\mu_{r|D,s}/(2\tilde{\sigma}_{r|s})$, respectively. We calculate $\Delta\mu_{z_c|D}$ as follows:

$$\begin{aligned}
\Delta\mu_{z_c|D} &= \mu_{z_c|D=1} - \mu_{z_c|D=-1} = \int \mathrm{d}s\, p(s|D=1)\mu_{z_c|D=1,s} - \int \mathrm{d}s\, p(s|D=-1)\mu_{z_c|D=-1,s} \\
&= \int \mathrm{d}s\, \frac{1}{2}[p(s|D=1) + p(s|D=-1)]\frac{\Delta\mu_{r|D,s}}{\tilde{\sigma}_{r|s}}.
\end{aligned} \tag{A7}$$

The first equality corresponds to the definition of $\Delta\mu_{z_c|D}$. In the second equality, we have estimated the means $\mu_{z_c|D=\pm1}$ in terms of the conditional means $\mu_{z_c|D=\pm1,s}$ using the general property that the mean of a variable is equal to the average of its conditional means. The third equality results from inserting the actual values of $\mu_{z_c|D=\pm1,s}$. Given *Equation (A7)*, the choice-triggered average obtained after pooling the normalized responses $z_c$ across stimulus levels corresponds to a weighted average of $\Delta\mu_{r|D,s}/\tilde{\sigma}_{r|s}$ across stimulus levels. Indeed, the factor $[p(s|D=1) + p(s|D=-1)]/2$ is properly normalized and plays the role of a weight $w_{z_c}(s)$, since $\int \mathrm{d}s\, p(s|D=\pm1) = 1$ and hence $\int \mathrm{d}s\, w_{z_c}(s) = 1$. Moreover, $\tilde{\sigma}_{r|s}$ only introduces a second-order correction with respect to the standard normalization with $\sigma_{r|s}$. In particular, given the skew-normal distributions (*Equation 15*) resulting from the threshold model, both $\Delta\mu^2_{r|D,s}$ and $\sigma^2_{r|D=\pm1,s}$ depend quadratically on the strength of the activity-choice covariations, as determined by the choice correlation (*Arnold and Beaver, 2000*; *Azzalini, 2005*). Neglecting this second-order correction we have that $\Delta\mu_{r|D,s}/\tilde{\sigma}_{r|s} \cong \Delta\mu_{r|D,s}/\sigma_{r|s}$ and $\sigma_{z_c} \cong \sigma_z = 1$. Furthermore, taking into account the general relation between the CP and CTA

(*Equation 4*), $\Delta\mu_{r|D,s}/\sigma_{r|s}$ approximates $2\sqrt{\pi}(\text{CP}_r(s)-1/2)$, and analogously, as indicated in *Equation (A6)*, $\Delta\mu_{z_c|D}/\sigma_{z_c}$ approximates $2\sqrt{\pi}(\text{CP}_{z_c}-1/2)$. Altogether, *Equation (A7)* can be expressed for the CP as

$$\text{CP}_{z_c} \approx \int ds\, \frac{1}{2}[p(s|D=1)+p(s|D=-1)]\text{CP}_r(s). \tag{A8}$$

As mentioned above, the weights $w_{z_c}(s) \equiv [p(s|D=1)+p(s|D=-1)]/2$ are properly normalized to $\int w_{z_c}(s)ds = 1$, and hence $\text{CP}_{z_c}$ is approximated as a weighted average of the CPs of the responses at each stimulus level, $\text{CP}_r(s)$. This shows that in fact the use of corrected z-scores to pool responses across stimulus levels to calculate a grand CP is in the linear approximation equivalent to calculating a weighted average of the CPs at each stimulus level, with a specific selection of the average weights, namely $w_{z_c}(s)$. An analogous derivation with the uncorrected z-scoring shows that in that case the pooling across stimulus levels is associated with an improper use of unnormalized weights, which analytically confirms the arguments and simulations in *Kang and Maunsell, 2012* indicating that a grand CP calculated with the standard z-scoring provides a biased estimation of an underlying stimulus independent CP (see a detailed derivation in section S2 of *Chicharro et al., 2019*). The weights $w_{z_c}(s)$ differ from the ones inversely proportional to the standard error of the CP estimates (*Equation 18*). In particular, if the stimulus set is designed such that across all stimulus levels the rate of the choices is balanced, i. e. if $p(D=1)=p(D=-1)$, then these weights simplify to $w_{z_c}(s)=p(s)$, that is, the CPs are averaged according to the relative number of trials available for each stimulus level. When there are no CP stimulus dependencies, the weights related to the estimate error are preferable since the grand CP will provide a better estimate of the underlying constant CP. In the presence of CP stimulus dependencies, any grand CP calculated as a weighted average across stimulus levels may introduce some confoundings in the comparison of grand CPs across cell types, areas, or across tasks. For example, if the distribution of the presented stimuli $p(s)$ is not uniform, the weights $w_{z_c}(s)=p(s)$, will assign a higher weight to the $\text{CP}(s)$ of certain stimulus levels, and difference in the grand CP across cells may reflect for which stimulus levels the cells compared have a higher $\text{CP}(s)$. Accordingly, characterizing the $\text{CP}(s)$ patterns can also help to understand if differences in grand CPs reflect functionally meaningful differences or are produced by the grand CP weighted average estimation.

## Appendix 3

### Clustering analysis

We here provide further details about the alternative procedures used to cluster the $\mathrm{CP}(p_{\mathrm{CR}})$ profiles, about the visualization of the clusters, and about how to assess the significance of the $\mathrm{CP}(p_{\mathrm{CR}})$ patterns. As a first step, we implemented a nonparametric $k$-means clustering analysis to cluster the $\mathrm{CP}(p_{\mathrm{CR}})$ profiles of the 107 cells for which a full profile could be constructed. We started using $C = 2$ clusters (*Figure 4A*) and found that this nonparametric approach, when using the cosine distance, recovered qualitatively the same patterns obtained when separating a priori the cells into cells with an average CP higher or lower than 0.5 (*Figure 3A*). From the patterns of the two clusters only the one of cells with an average CP higher than 0.5 was found significant (see below for details on the significance analysis). Given this difference in significance, we subsequently increased the number of clusters in two alternative ways. In a first approach, we a priori separated the cells with an average CP higher or lower than 0.5 and continued the clustering analysis separately for these two groups. *Appendix 3—figure 1A* shows the obtained subclusters with $C = 2$ for the two groups separately. As a second approach, we increased the number of clusters to $C = 3$ without any previous separation in two groups. The resulting clusters (*Appendix 3—figure 1B*) indicated that the separation of the two subclusters for the cells with average CP higher than 0.5 naturally appears without enforcing the separation. Increasing the number of clusters without any a priori separation provided evidence that the two main patterns for cells with average CP higher than 0.5 are robust and still contain a substantial portion of the cells even for $C = 6$ (*Appendix 3—figure 1C*). We therefore focused our posterior analysis in characterizing the features of these symmetric and asymmetric patterns.

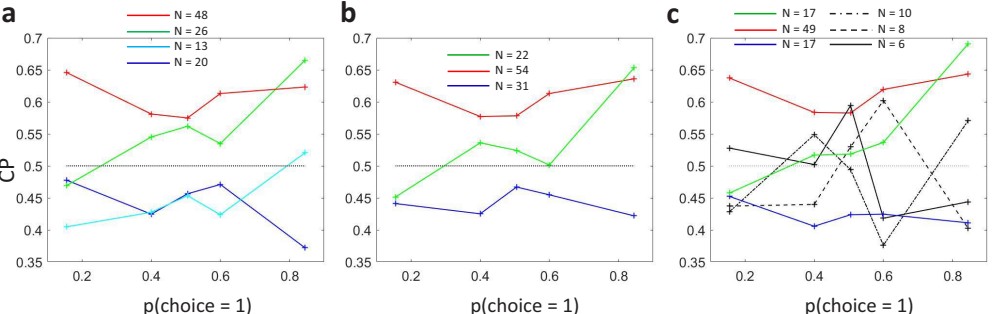

**Appendix 3—figure 1.** Subclustering of $\mathrm{CP}(p_{\mathrm{CR}})$ dependencies. (**a**) Analogous to *Figure 4B* but showing also the average profiles for the two subclusters obtained from cells with average $\mathrm{CP}<0.5$. (**b**) Nonparametric $k$-means clustering with three clusters determined from all cells. (**c**) Nonparametric $k$-means clustering with six clusters determined from all cells. The clusters more similar to the ones of *Figure 4B* are correspondingly coloured.

To evaluate the significance of the $\mathrm{CP}(p_{\mathrm{CR}})$ patterns found with the clustering analysis, we repeated the same clustering procedure for the surrogate data generated as described in Methods. For each surrogate, each of the $C$ clusters found was associated with the most similar original pattern of the ones being tested. For example, in *Figure 4B*, when testing the significance of the symmetric and asymmetric patterns for cells with average CP higher than 0.5, each of the two surrogate cluster patterns was assigned to the most similar pattern among the symmetric and asymmetric one. Subsequently, the average of $\Delta\mathrm{CP}_k$ across bins was calculated for the original and surrogate profiles as explained in the Methods. The p-value corresponding to each original pattern was calculated from the number of surrogate patterns associated with it for which the average $\Delta\mathrm{CP}_k$ was higher.

To visualize the clusters in *Figure 4* and *Appendix 3—figure 1A*, we constructed orthonormal axes using either the vectors corresponding to the center of the clusters or the selected templates, for nonparametric and parametric clustering, respectively. In the case of nonparametric clustering, the x-axis corresponds to the separation between the two initial clusters, and is closely aligned to the departure of the average CP from 0.5. The y-axis was built as a projection orthogonal to the x-axis of the vector connecting the center of the two subclusters. When templates were used, the

x-axis corresponds to the template with a constant CP and the y-axis was built as an orthogonal projection of the template with an asymmetric profile (a vector with a positive unit slope).

# Appendix 4

## The effect of gain fluctuations on the CP

We here derive a general CP expression accounting for the effect of response gain fluctuations valid for a feedforward encoding/decoding decision-making model with any unbiased weights, and subsequently focus on the decoder based on two pools of cells with opposite choice preference, as previously studied in *Shadlen et al., 1996*. As described in Methods, we consider a decoder $d = \vec{w}^\top \vec{r}$ (*Equation 8*), estimating the decision variable $d$ from the responses $r_i = f_i(s) + \xi_i$, with tuning functions $\vec{f}(s) = (f_1(s), ..., f_n(s))$ and a covariance structure $\Sigma(s)$ of the neuron's intrinsic variability $\xi_i$. The general expression of the CP valid for any stimulus level and agnostic about the source of the activity source covariations is

$$\mathrm{CP}_i(s) \approx \frac{1}{2} + \frac{\sqrt{2}h(p_{\mathrm{CR}})}{\pi}\mathrm{CC}_i(s), \tag{A9}$$

and, as described in *Equation 9*, the CC is determined by the covariance matrix $\Sigma(s)$ and the read-out weights $\vec{w}$ as

$$\mathrm{CC}_i(s) = \frac{(\Sigma(s)\vec{w})_i}{\sqrt{\vec{w}^\top\Sigma(s)\vec{w}}\sqrt{\sigma_{r_i}^2(s)}} \tag{A10}$$

where $\mathrm{cov}(r_i, d)(s) = (\Sigma(s)\vec{w})_i$ and $\sigma_d^2(s) = \vec{w}^\top\Sigma(s)\vec{w}$. Given that the covariance matrix has a structure $\Sigma(s) = \bar{\Sigma} + \sigma_G^2\vec{f}(s)\vec{f}^\top(s)$ (*Equation 10*), with the component $\sigma_G^2\vec{f}(s)\vec{f}^\top(s)$ induced by the gain fluctuations, a change $\Delta s = s - s_0$ in the stimulus level departing from the uninformative stimulus $s_0$ alters the covariance structure as indicated in *Equation 22*. This leads to the following perturbation of the CC

$$\mathrm{CC}_i(s) \approx \mathrm{CC}_i(s_0)\left[1 - \frac{1}{2}\left(\frac{\Delta\sigma_d^2(s)}{\sigma_d^2(s_0)} + \frac{\Delta\sigma_{r_i}^2(s)}{\sigma_{r_i}^2(s_0)}\right)\Delta s\right] + \frac{\Delta\mathrm{cov}(r_i, d)(s)\Delta s}{\sigma_{r_i}(s_0)\sigma_d(s_0)}, \tag{A11}$$

which is a generalization of *Equation 23b*, where $\Delta\sigma_{r_i}^2(s)$, $\Delta\sigma_d^2(s)$, and $\Delta\mathrm{cov}(r_i, d)(s)$ are the changes produced in the variances and covariance due to $\Delta s$. From *Equations (A10) and 22*, the change in the variability of the responses is $\Delta\sigma_{r_i}^2(s) = 2\sigma_G^2 f_i(s_0)f_i'(s_0)$, the change in the variability of the decision variable is $\Delta\sigma_d^2(s) = 2\sigma_G^2\vec{w}^\top\vec{f}(s_0)\vec{w}^\top\vec{f}'(s_0)$, and the covariance varies in $\Delta\mathrm{cov}(r_i, d)(s) = \sigma_G^2 f_k(s_0)\vec{w}^\top\vec{f}'(s_0) + \sigma_G^2 f_k'(s_0)\vec{w}^\top\vec{f}(s_0)$. Furthermore, for any unbiased decoder $\vec{w}^\top\vec{f}'(s_0) = 1$, so that $d = \vec{w}^\top\vec{r}$ properly estimates $d(s) - d(s_0) = \vec{w}^\top f'(\vec{s_0})\Delta s = \Delta s$ (*Moreno-Bote et al., 2014*). Taking this into account, *Equation (A11)* can be expressed as

$$\mathrm{CC}_i(s) \approx \mathrm{CC}_i(s_0)[1 - \sigma_G(\lambda_d\eta_d + \lambda_i\eta_i)\Delta s] + \sigma_G(\lambda_d\eta_i + \lambda_i\eta_d)\Delta s. \tag{A12}$$

Here $\lambda_i \equiv \sigma_{r_{ig}}(s_0)/\sigma_{r_i}(s_0)$ quantifies the portion of the responses variance of cell $i$ caused by the gain fluctuations, as defined in *Equation 11*, and $\lambda_d \equiv \vec{w}^\top\vec{f}(s_0)/\sigma_d(s_0)$ quantifies the portion of the variance of the decision variable caused by the gain fluctuations. We define $\eta_i \equiv f_i'(s_0)/\sigma_{r_i}(s_0)$, which quantifies the neurometric sensitivity of the cell, and $\eta_d \equiv 1/\sigma_d(s_0)$, which quantifies the behavioral sensitivity. Note that *Pitkow et al., 2015* defined the so-called neural threshold $\theta_i$ and behavioral threshold $\theta_d$ such that $\eta_i = 1/\theta_i$ and $\eta_d = 1/\theta_d$, but in our case the measures of sensitivity are more suited to describe the dependence of the CC. In particular, the $\mathrm{CC}_i(s_0)$ of the uninformative stimulus has an attenuation factor that depends on the relative increase in variability in the responses and in the decision variable, each determined by the product $\lambda_k\eta_k$ of the sensitivity to the change in the stimulus and the relative magnitude of the variance produced by the gain fluctuations. On the other hand, the additional contribution to $\mathrm{CC}_i(s)$ depends on the cross-products $\lambda_k\eta_{k'}$ of the relative magnitude of the gain-related variance for the cell and the sensitivity of the decision variable, or vice-versa. Rearranging the terms this equation can also be expressed as

$$\mathrm{CC}_i(s) \approx \mathrm{CC}_i(s_0) + \sigma_G\left[\lambda_i\eta_d\left(1 - \mathrm{CC}_i(s_0)\frac{\eta_i}{\eta_d}\right) + \lambda_d\eta_i\left(1 - \mathrm{CC}_i(s_0)\frac{\eta_d}{\eta_i}\right)\right]\Delta s. \tag{A13}$$

From this expression, *Equation 23b* in Methods is recovered for the case of the optimal decoder, since the CC with the optimal decoder at $s_0$ is equal to the ratio of the neural and the behavioral threshold (*Pitkow et al., 2015*), that is, $\mathrm{CC}(s_0) = \eta_i/\eta_d$, as indicated in *Equation 21*. For this optimal decoder the second summand of *Equation (A13)* is canceled out, while in the first $\mathrm{CC}_i(s_0)\eta_i/\eta_d$ equals $\mathrm{CC}_i^2(s_0)$, ensuring that the slope of dependence on $\Delta s$ is positive. More generally, $\lambda_d = \vec{w}^\top \vec{f}(s_0)$ is zero when the decoder is uncorrelated to the magnitude of the tuning curves. In this case the performance of the decoder is not affected by global gain fluctuations when presenting a non-informative stimulus (*Moreno-Bote et al., 2014*; *Ecker et al., 2016*) and the gain does not contribute to the variability of $d$ or its covariance with the cell responses. This additional assumption was used to determine *Equation 23a*, such that only the variance of the cell changes with respect to the case of no gain fluctuations.

We now additionally examine the decoder formed by two pools of cells with opposite choice preference, because of the role it has played in previous understanding of activity-choice covariations (*Shadlen et al., 1996*; *Cohen and Newsome, 2009b*). We consider the particular configuration examined in *Haefner et al., 2013*, such that the decoder is formed by $n$ neurons divided in two pools of $n/2$ neurons, all with the same variance $\sigma_{r_i}^2(s_0)$, and with the same intra-pool covariance $\mathrm{cov}_{\parallel}(r_i, r_j)$ for all pairs of cells within the same pool and the same inter-pool $\mathrm{cov}_{\perp}(r_i, r_j)$ for all pairs across pools. The read-out weights all have the same magnitude, with opposite sign for the cells of the two pools. For this configuration, *Haefner et al., 2013* derived (see their Suppl. Material S5) that $\mathrm{CC}_i(s_0) \approx \pm\sqrt{1/n + \Delta\varrho/2}$, where $\Delta\varrho \equiv (1 - 2/n)\rho_{\parallel} - \rho_{\perp}$ is the difference between the intra- and inter-pool correlations, in the limit of a large pool, and the sign of the choice correlation is the opposite across pools. They also showed that $\sigma_d(s_0) = n\sigma_{r_i}(s_0)|\mathrm{CC}_i(s_0)|/c$, with $c$ being a normalization factor of the weights to ensure that the decoder is unbiased ($\vec{w}^\top \vec{f}'(s_0) = 1$). Accordingly, for this decoder $\eta_i/\eta_d = nf_i'(s_0)|\mathrm{CC}_i(s_0)|/c$. Assuming that for a pair of neuron/antineuron in the two pools their firing rates have derivatives of same magnitude and opposite sign, $c = n\langle|f'(s_0)|\rangle$, being $\langle|f'(s_0)|\rangle$ the average of the magnitude of the derivatives. This leads to $\eta_i/\eta_d = f_i'(s_0)|\mathrm{CC}_i(s_0)|/\langle|f'(s_0)|\rangle$. Further following the idea that the two pools contain neurons and antineurons with the same response properties but opposite choice preference (*Cohen and Newsome, 2009b*), $\lambda_d = \vec{w}^\top\vec{f}(s_0) = 0$, since for the uninformative stimulus each neuron with tuning curve $f_k(s_0)$ is paired by a cell in the other pool with the same firing rate and an equal weight but of opposite sign. Accordingly, for this decoder *Equation (A13)* takes the form

$$\mathrm{CC}_i(s) \approx \mathrm{CC}_i(s_0) + \sigma_G \lambda_i \eta_d \left[ 1 - \mathrm{CC}_i^2(s_0) \frac{f_i'(s_0)}{\langle|f'(s_0)|\rangle} \right] \Delta s. \tag{A14}$$

Given the structure of the covariance matrix, this decoder is in fact optimal if furthermore all cells had the same derivative $f_i'(s_0)$, in which case *Equation (A14)* equals *Equation 11* and for all cells the $\mathrm{CP}(s)$ pattern has a positive slope in dependence on $\Delta s$. More generally, with this two-pools decoder and covariance matrix, the $\mathrm{CP}(s)$ pattern can have a negative slope for those cells with larger derivatives. Indeed, only for the optimal decoder the general model of *Equation (A13)* guarantees a positive slope.

Finally, in *Appendix 4—figure 1* we further characterize the dependencies predicted by the model when using an optimal decoder (*Equations 11 and 23*). The covariation of the coefficients $\beta_{p_{\mathrm{CR}}} \equiv \sigma_G \lambda_i (1 - \mathrm{CC}_i^2(p_{\mathrm{CR}} = 0.5))$ from *Equation 23b* and $\beta_{\mathrm{CC}} \equiv \sqrt{1 - \lambda_i^2}$ from *Equation 23a* that modulate the strength of the $\mathrm{CP}(p_{\mathrm{CR}})$ dependence and the magnitude of $\mathrm{CP}_i(p_{\mathrm{CR}} = 0.5)$, respectively, is shown in *Appendix 4—figure 1A*. We determined $\lambda_i^2$ only in terms of the strength of the gain as for *Figure 4E*, namely as $\lambda_i^2 = 1/(1 + 0.1/\sigma_G^2)$. The range $\sigma_G^2 = [0 - 0.5]$ corresponds to $\lambda_i^2 = [0 - 0.83]$. *Appendix 4—figure 1B-C* further illustrate how combinations of different $\mathrm{CC}_{i0}(p_{\mathrm{CR}} = 0.5)$ and $\sigma_G^2$ populate the 2-D space of $\mathrm{CP}(p_{\mathrm{CR}})$ profiles as in *Figure 4D,E*. $\mathrm{CP}(p_{\mathrm{CR}})$ profiles were simulated randomly sampling the average CP values from the ones observed for the MT cells. For *Appendix 4—figure 1B,D*, the fluctuation gains were uniformly sampled from the interval $\sigma_G^2 = [0 - 0.1]$, corresponding to $\lambda_i^2 = [0 - 0.5]$. For *Appendix 4—figure 1C,E*, the 2-D space was not evenly sampled, simulating a further dependence between $\mathrm{CC}_{i0}(p_{\mathrm{CR}} = 0.5)$ and $\sigma_G^2$ values which

determines the exact balance between the symmetric and asymmetric dependencies observed in the average profiles associated with each cluster.

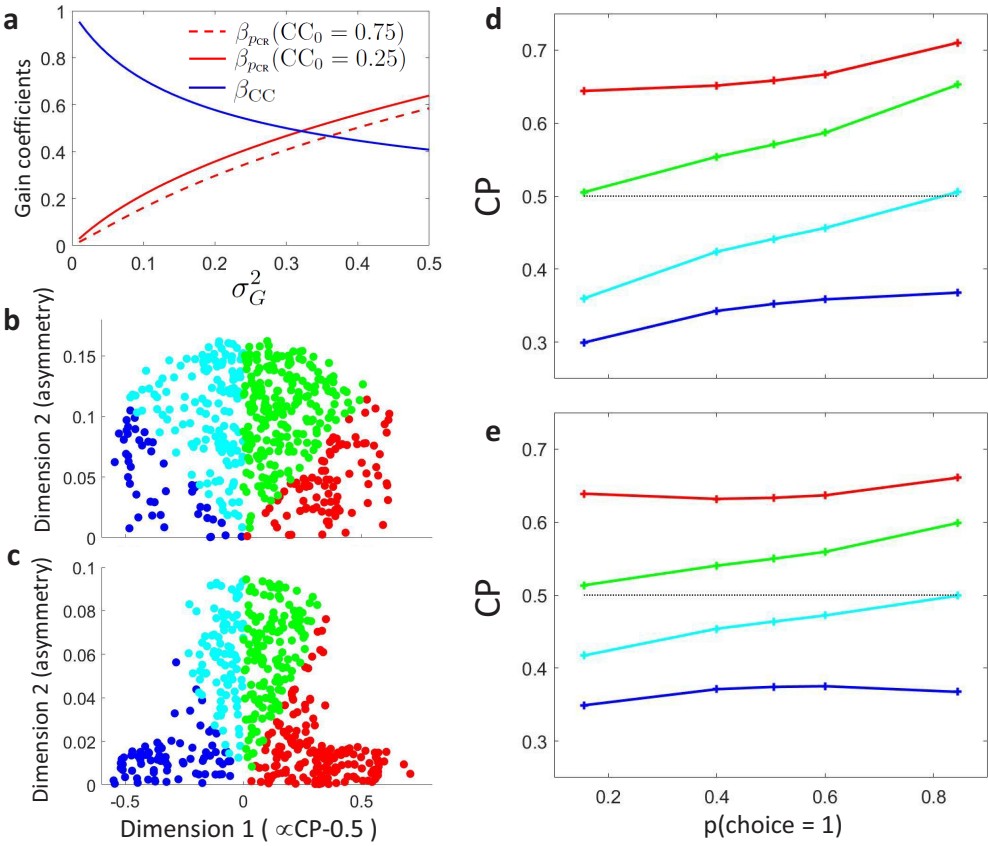

**Appendix 4—figure 1.** Modeling the influence of neuronal gain modulation on $\mathrm{CP}(p_{\mathrm{CR}})$ profiles. (**a**) Dependence of gain coefficients $\beta_{CC}$ and $\beta_{p_{\mathrm{CR}}}$ (**Equations 23**) on the strength of the gain fluctuations, $\sigma_G^2$ determines their effect on the choice correlation $\mathrm{CC}_i(s_0)$ for the uninformative stimulus $s_0$. $\beta_{p_{\mathrm{CR}}}$ determines the degree of asymmetry of the choice correlation dependence on the $p_{\mathrm{CR}}$. (**b**) $\mathrm{CP}(p_{\mathrm{CR}})$ profiles, represented in the same 2-D space as in **Figure 4D,E**, generated with a uniform sampling of $\mathrm{CC}_{i0}(s_0)$ consistent with the observed average CPs of the MT cells, and with a uniform sampling of the gain ($\sigma_G^2 \sim \mathcal{U}(0, 0.1)$). (**c**) Analogous to b, but with a nonuniform distribution in the 2-D space, reflecting structure in the covariation of $\mathrm{CC}_{i0}(s_0)$ and $\sigma_G^2$. (**d–e**) $\mathrm{CP}(p_{\mathrm{CR}})$ profiles corresponding to the clusters centers obtained when sampling the space according to panels b and c, respectively.

