## [Decision Letter]

**Acceptance summary:**

Perceptual decisions rely on neural activity of sensory neurons. This work presents new model-based analytical tools to understand the relationship between sensory stimulus, sensory neural activity and perceptual choice. Dependencies between neural and behavioural observables in studies of decision making can be quite complex and non-intuitive, as different observables depend in different ways on unobserved hidden states (e.g., decision variable, decision criterion). This paper derives these dependencies for standard assumptions (Gaussian variability) and predicts how dependencies will change under more realistic assumptions (gain modulation). This is a thoughtful reworking of previously published data, making the suggestion that there is a functional relationship between ongoing neural activity and behavioral decisions. This new analysis is however still limited by the available data. Ultimately, this paper suggests new avenues that should be explored by neuroscientists when modelling perceptual decisions.

**Decision letter after peer review:**

Thank you very much for submitting your article "Stimulus dependent relationships between behavioral choice and sensory neural responses" for consideration by *eLife*. Your article has been reviewed by 3 peer reviewers, and the evaluation has been overseen by a Reviewing Editor and Joshua Gold as the Senior Editor. The following individuals involved in review of your submission have agreed to reveal their identity: Jochen Braun (Reviewer #1); Andrew J Parker (Reviewer #3).

The reviewers have discussed the reviews with one another and the Reviewing Editor has drafted this decision to help you prepare a revised submission.

As the editors have judged that your manuscript is of interest, but as described below that additional experiments/analysis are required before it is published, we would like to draw your attention to changes in our revision policy that we have made in response to COVID-19 (https://elifesciences.org/articles/57162). First, because many researchers have temporarily lost access to the labs, we will give authors as much time as they need to submit revised manuscripts. We are also offering, if you choose, to post the manuscript to bioRxiv (if it is not already there) along with this decision letter and a formal designation that the manuscript is "in revision at *eLife*". Please let us know if you would like to pursue this option. (If your work is more suitable for medRxiv, you will need to post the preprint yourself, as the mechanisms for us to do so are still in development.)

Summary:

The manuscript "Stimulus dependent relationships between behavioral choice and sensory neural responses" by Haefner and colleagues addresses interpretations of choice probability, which is the capacity of sensory neurons to predict the decisions made by experimental subjects in sensory decision tasks. Variability in the responses of individual neurons is predictive of the subject's choice. This paper further develops previous theoretical findings concerning this "choice probability". At the centre of the new modelling approach is the important relationship between these variable responses and the strength of the stimulus-related component of neuronal firing and how this can be formalised in an algorithm.

Essential revisions:

This manuscript further develops previous theoretical findings concerning "choice probability", the relationship of the trial-to-trial variation of firing rates of a (sensory) neuron and the behavioral choice of a subject in perceptual decision making experiments. CPs and their interpretation have a long history, including the debate about their origin (do CPs reflect a causal effect of sensory neural variability on the decision or do they arise from top-down feedback signals?) Here, the authors identify an additional factor by showing that CP depends on the "choice rate" (the fraction of left vs. right choices). Since this choice rate depends on stimulus strength, CP itself will depend on stimulus strength. They also show experimental evidence for this effect (in the classical Britten et al. dataset) that has been overlooked until now and present methods for refined analysis for linking the responses of sensory neurons to choices.

The reviewers raised a number of central concerns about the relationship between the model and the available empirical data. They agree that the relationship between pCR and CP must be firmly established as a general experimental fact, or reasons given why it is apparent in some conditions and not others. Also, the theoretical interpretation of the relationship between pCR and CP needs to deliver accurate predictions of the experimental results and also to demonstrate consistency with theoretical models of how other factors affect CP (interneuronal correlation and stimulus sensitivity, being the primary variables of concern here).

1. The relationship between pCR and CP appears more at the core of this rather than the related stimulus strength vs CP, but there is a quantitative mismatch between model prediction and empirically observed data, especially in Figure 3a and Figure 4b. Interpretation of Figure 3a. Our initial concern when reading the paper was that the predicted effect of choice rate on CP (dotted line in Figure 3a) is so small that there will not be enough statistical power to detect it in the available experimental data. Moreover, the analysis is complicated by the fact that the effect of h(p_CR_) is partly masked by the gain modulation effect (Figure 4b). However, for the n=48 neurons (Figure 4b) that show the predicted U-shape of CP vs. p_CR_, the effect is actually much larger than the theoretical prediction (e.g. or p_CR_ ~0.1 the CP is close to 0.65 when in theory it should be ~0.58). What could be the reason for this quantitative mismatch?

We would like to see in this regard:

a. A clear discussion of and situation of the results within the available empirical literature on CP, especially with regards to quantitative effect size in Britten et al. 1996 and Dodd et al. 2001.

We agree that in some previous papers CP is computed from z-scored spike counts across different stimulus conditions (and thus different choice rate values). However, there are several other papers (including Britten et al. 1996, Nienborg and Cumming 2009, Wimmer et al. 2015, Katz et al. 2016) that compute CP from "zero coherence trials" with choice rate ~ 0.5 (or the equivalent for binocular disparity in V5/MT: Dodd et al. 2001; Krug et al. 2004; Wasmuht et al. 2019). These papers should be mentioned in order to avoid the impression that all previous work suffers from a (slightly) incorrect way of computing CP.

b. Re-analysis of the main effect in different subsets of cells. We are concerned that modelling the group for which CP < 0.5 includes a lot of noise, because the members of this are almost always non-significant, as shown in Dodd et al. and in Britten et al. (See their Figure 5). If you look at Dodd et al., there is no sign of this relationship between pCR and CP. Moreover, in Dodd et al. there are very few 'wrong way' choice probabilities: so few in fact that they are within the statistical bounds of repeated measures. So the procedure of separating the CPs into >0.5 and <0.5 is not likely to alter the result at all from the published Figure in Dodd et al. It might be helpful to recalculate the results for Figure 3A but only including those neurons that show an individually significant CP > 0.5.

c. A simulation of the factors that affect expected effect size. The simulation should show what size of the effect is expected given the limited amount of noisy data (number of neurons, trials, etc.). This would allow you to determine whether the experimental result lies with the confidence bounds of the theoretical prediction.

There is apparently no such relationship between CP and p_CR_ in the Dodd et al. data (but it could perhaps show up when selecting and binning the neurons as described in the manuscript, line 288). In any case, a possible explanation for the discrepancy could be that the expected relationship is so small that it cannot reliably be detected from ~100 neurons. We also think also this issue can be clarified by simulating the size of the effect for limited data.

2. Clarifications are further needed in term of how the fluctuations in the stimulus-related gain of neuronal firing are responsible for the emergence of stronger CPs at higher performance levels. As we understand it, this is required both in terms of the formal implementation in the model and in terms of the proposed neurobiological implementation.

The authors present a detailed analysis of abstract decision-making models. They relate noisy neuronal responses ri, a hypothetical covert decision variable d and decision threshold θ, and an overt behavioural choice D. The authors assume a bivariate Gaussian association between ri and d, with a certain correlation coefficient, and from this minimal basis derive exact or approximate expressions for choice rate pCR, choice probability CPi, choice correlation CCi, between ri and d, and choice-triggered averages of ri, CTAi , and of d, CTAd. The treatment extends previous work in that it covers the entire range of choice rates pCR, not only the special case pCR = 0.5.

A key result is that choice probability CPi changes multiplicatively as a function of pCR, increasing as the decision grows more consistent in either direction, with the baseline level set by choice correlation CCi. An important implication is that the dependence of CPi on pCR, which is shared by all cells and expected to be U-shaped, can be averaged over cells with different choice correlation CCi, provided that cells with positive and negative choice correlations are distinguished.

To test these predictions, the authors re-analyze MT recordings from Britten et al. (1996) and were able to confirm a U-shaped dependence of average CP on pCR, which was statistically significant for cells with positive CCi. However, contrary to predictions, the U-shaped dependence was asymmetric and more pronounced when the more frequent choice (pCR > 0) is consistent with the preferred stimulus of the cell (positively correlated cells CCi>0). A cluster analysis of empirical individual cell dependencies of CPi on pCR revealed that, in addition to the predicted U-shaped dependencies, the presence of unexpected monotonically increasing dependencies.

To clarify the origin of these unexpected dependencies, the authors consider the effect of trial-to-trial response gain fluctuations (Goris et al., 2014) and, with the help of the model of these authors, confirm that gain fluctuations account for 62% of the observed trial-to-trial variance. The authors point out that gain fluctuations add a stimulus-dependent component to the noise covariance of neural responses, which is inherited by choice correlation CCi and by choice probability CPi. In Methods and Supplementary Text S4, the authors derive that this stimulus-dependent component can itself depend asymmetrically on pCR, to an extent that is specific to each cell (i.e., the specific coupling between response and gain). Unfortunately, the authors do not offer an intuitive argument about the origin of this asymmetry.

Please explain how gain fluctuations lead to rising dependencies on pCR in some cells. Which cells are these? Do they combine strong stimulus modulation with weak choice correlation? Without such an explanation, the entire cluster analysis appears incomplete and ultimately pointless.

3. Following on from this, there is also the question how could the stronger CP with increasing pCR be implemented in different ways in actual neural terms. Would for example the pool size of neurons contributing to the decision change? With regards to the model, how does the effect of pCR x CP intersect with the experimentally much stronger interaction effect of neurometric sensitivity and size of the CP?

The authors analyse this from the classical Britten et al. [1] data set, producing a framework for analysis in Figure 2 and the outcome of analysis in Figures3 and 4 of the paper. This type of analysis is not new. The specific form of the plots in Figures2,3,4 appears in Dodd et al. [2] in Figure 6, while Figure 3 in Britten et al. delivers almost the same information.

In regard to the clustering analysis, it seems that the big driver for the formation of clusters is the division between CP >0.5 and CP<0.5. For values of CP<0.5, there is not really a functional account of these, as they do not relate to the tuning of the cells for motion. The lack of functional meaning is highlighted by the fact that cluster 1 in Figure 4a (blue==CP <0.5) is statistically non-significant. Not unrelated to this lack of significance is the fact that Figure 13 of Britten et al. and Figure 6 of Parker et al. [3] show that CPs are stronger for neurons that are more sensitive to the visual task. The usual interpretations of this are either the intuitive claim that more sensitive neurons are more tightly involved in the task and therefore have higher CPs or, more subtly, that neurons with weaker sensitivity have lower degrees of interneuronal correlation.

The rest of the analysis in this paper advances the idea that fluctuations in the stimulus-related gain of neuronal firing are responsible for the emergence of stronger CPs at higher performance levels. The authors write on ll243-4 that "Briefly, the contribution of gain modulations to the covariance of the responses increases with neuronal firing rates, which in turn are stimulus-modulated as determined by tuning functions.". However, the lead author has already published a nice theoretical summary [4] showing that CP is related to the level of interneuronal correlations in the pool. Indeed, the analysis showed that under some conditions (large pool, correlated noise and at least one or two members of the pool contributing significantly to perceptual read-out) one might take CP as substitute indicator for the interneuronal correlation of the decision pool. In the light of the earlier analysis, the present paper does not address the very relevant question of changes in the membership of the neuronal pool with stimulus strength. In effect, if read-out weights change with stimulus strength, then CP will be expected to change. Equally, if pool membership changes then interneuronal correlation may be expected to change. We did not see anything in this analysis here that definitively ties down the change in CP to stimulus-related gain changes.

l 242-243: " We derived the specific form of CC𝑖(𝑝CR) predicted from gain modulations in the threshold decision model to explain additional CP stimulus dependencies beyond the symmetric modulation by h(𝑝CR ). Briefly, the contribution of gain modulations to the covariance of the responses increases with neuronal firing rates, which in turn are stimulus-modulated as determined by tuning functions. This leads to an asymmetric component of CP(𝑝CR), with higher CPs for 𝑝CR values associated with stimuli preferred by the cell. Furthermore, while stronger gain fluctuations increase this asymmetric stimulus dependence, they also decrease the magnitude of the cell-specific CP because they add variability to the responses unrelated to the choice "

As neuronal sensitivity has normally been measured, the change in response due to a change in the stimulus is assessed relative to the variability of neuronal firing. The description above implies that stimulus modulations in the Haefner models translate into response changes on top of which random gain modulations are applied. At first sight, there does not seem any room in the model for low firing rate, low variability neurons to contribute to CP, even though such neurons may have high neurometric sensitivity. One the other hand, it may well be that the new Haefner model all shakes down to give the established experimental result that CP is linked to neuronal sensitivity. If that's correct, then the paper is currently rather obscure on this point and it will be useful for the paper to lay this out clearly.

4. Editing for accessibility and readability

Unfortunately, our genuine enthusiasm for the manuscript is somewhat dampened by its length, by its opacity in places, and by the high degree of topic familiarity that it presupposes. For example, the discussion of grand CP on page 7 and in section S2 of the supplementary material, is difficult to follow even for someone in the field. Accordingly, if readability could be improved, the usefulness would be even greater.

We appreciate the theoretical advance of the paper. It is very useful to have equations that clarify the relationship between previously used measures (CP, CTA, CC) and the effect of informative stimuli. Overall, by the very nature of the topic, the paper is rather technical. Our impression was that the paper is not easy to read, in particular when we think about a broader readership that is not familiar with details of the theory of CP and the typical interpretation of CP measurements. We understand that this is not an easy job because the interpretation of CP is complicated (bottom-up vs. top-down contributions, relationship to spike count correlations, etc) but it would help to revise the Results section and add some more details and (where possible) intuitive interpretation of the theoretical and experimental results to guide the reader. As an example, it turns out that gain fluctuations are an important factor to explain CP vs p_CR_ but this is only treated very briefly in the results (gain fluctuation model is not explained, no figure is shown about how well the model explains the variability observed in the data, etc).

[Editors' note: further revisions were suggested prior to acceptance, as described below.]

Thank you very much for submitting your article "Stimulus-dependent relationships between behavioral choice and sensory neural responses" for consideration by *eLife*. Your article has been reviewed by 3 peer reviewers, and the evaluation has been overseen by a Reviewing Editor and Joshua Gold as the Senior Editor. The following individuals involved in review of your submission have agreed to reveal their identity: Jochen Braun (Reviewer #1); Klaus Wimmer (Reviewer #2).

The reviewers have discussed their reviews with one another, and the Reviewing Editor has drafted this to help you prepare a final revised submission.

All the reviewers agreed on the significant theoretical advance of your manuscript. However, there were some concerns about the generality of the proposed model given that the data supporting it have limitations. Therefore, we propose a small but essential revision that makes it clear to the reader that the empirical support for the new model requires further explorations. Additionally, we have added the reviewers' further suggestions for you as recommendations.

Essential Revision:

1. Overall, the empirical part looks a little bit like an attempt to drag meaning out of a weak relationship, at least as measured experimentally in the single, but classical data set of Britten et al. The more detailed explanation of the clustering analysis reveals that the main effect of interest is driven by one critical data point from the original Britten et al. data. This is the value of CP for pCR = 0.85, in the data set for which CP > 0.5. Much of the paper depends on just how confident we are about a difference in CP that rises from 0.57 for pCR=0.5 to 0.64 for pCR=0.85. The SEMs calculated for individual neurons in Figure 3d (lower right panel) are not encouraging in this regard.

We think that further empirical tests are needed to obtain clarity about significance about key elements of the proposed model but this is beyond the scope of the current manuscript. It should be flagged to the reader, though.

In order to clarify to the reader that the empirical part provides only an initial analysis – as was also explicitly mentioned by the authors in their response to the previous reviews – we ask the authors to insert something close to the following statement in the abstract and in the first paragraph of the results:

This paper provides preliminary empirical evidence for the promise of studying stimulus dependencies of choice-related signals, which requires further exploration in a wider data set.

2. The revisions are acceptable in that the presentation indeed streamlined in many respects (thank you!). However, one large reservation remains: the main findings are NOT explained in an intuitive manner.

Basically, when we distinguish neural responses associated with choice D=1 from those associated with choice D=0, we obtain two more or less distinct distributions, P(R|D=1) and P(R|D=0). When these distributions are well separated, then the choice probability CP (average probability that an R|D=1 is larger than an R|D=0) is somewhat larger than when these distributions are more overlapping.

However, both qualitative and quantitative aspects of the relation between choice probability CP and choice rate PCR are highly dependent on assumptions. For example, the symmetric U-shape depends on Gaussian variability of responses. When responses are assumed to be Poisson-variable, for example, the symmetric U-shape is replaced by a monotonic decline (Matlab code available upon request).

Gain modulation changes the picture in exactly this way. After gain modulation, response variability is no longer Gaussian and, additionally, the set of responses with the larger average is more affected than the set with the smaller average. In other words, gain modulation alters the shape of response distributions and in consequence also the shape of CP = f(PCR).

In spite of this reservation, I have to confess that this paper has been quite useful to me, because it forced me to think through these issues.

3. With regards to the dependence of the results on one critical data point in the Britten et al. data set (CP for pCR = 0.85) and the question of significance to other data sets, various suggestions for further data worth testing were made.

We wonder, for instance, whether Geoff Ghose's lab (https://pubmed.ncbi.nlm.nih.gov/30067123/ ; https://pubmed.ncbi.nlm.nih.gov/19109454/ ) is a better source of data to support this exercise. In particularly, his recordings could be searched for signs of noisy gain modulations, which is the mechanism that lies at the core of this analysis. Other suggestions were as mentioned before the Dodd et al. 2001/Wasmuht et al. 2019 data.

But we agree this is beyond the current scope of the manuscript.

4. Line 10: "activity-choice covariations are traditionally quantified with a single measure of choice probability (CP), without characterizing their changes across stimulus levels" I appreciate the need to demonstrate novelty in the paper, but this statement does a disservice to earlier researchers who recognized the possibility of a stimulus-choice interaction but did not find any evidence of such an interaction. The earlier papers are clear on this point. So the novelty here is not the failure of earlier researchers to think through their results carefully; the novelty here is an apparent improvement in sensitivity of the analysis methods. This may sound less exciting but is still important, if correct.

5. Line 65 "if the decision process uses a decision threshold"

6. Line 96 "choice probability, CP, defined as the probability that a random sample from all trials"; the reader only learns on line 99 what this is a random sample of. It would better read as "a random sample of neural activity r " and then explain in a separate sentence what r might be in any particular experimental situation.

7. Line 147: "threshold value 𝜃" the authors insist on referring to this parameter 𝜃 as a threshold value. This usage will inflame debate as to whether there is a "high threshold assumption" baked into this model, where "high threshold" here means a classical high threshold in visual detection models, as opposed to a signal detection model. As classical high threshold theory is now rejected for detection models, I think it would be better here and throughout to refer to 𝜃 as a criterion value, which is what it is and better aligns with the language of signal detection theory.

8. Lines 129-169: this development is still very difficult to follow and labors over what are some fairly basic points. It would better be rewritten with better structure at about half the length. For example, compare lines 111-112 and 134-135, which could be combined into a single point that is made once at the right stage in the argument.

9. Lines 170-209: the writing continues in a stilted manner with multiple cross-references to other material.

10. Lines 173-4 "We here will refer to CP stimulus dependencies and CP(𝑝CR ) patterns interchangeably". We rather fear that this is going to cause a lot of readers to trip over. I can see that these two are interchangeable from the theoretical perspective of these authors, but many will think that behavior pCR is dependent on a number of factors other than the stimulus. In the field of monkey neurophysiology, pCR will depend on reward, attention, arousal and so forth, in a way that the stimulus does not. What We think the authors actually mean is something like "Within the structure of our model, there is a fixed relationship between the dependence of CP on the stimulus strength and the dependence of CP on the choice rate pCR, for each threshold."

11. Lines 263-265 "we show how to extend Generalized Linear Models (GLMs), a popular model to characterize the factors modulating neural activity, to include a stimulus-choice interaction terms" As a statistical procedure, this is fairly routine stuff and could be abbreviated considerably.

12. Lines 376-377 "Specifying the existence of two clusters, we naturally recovered the distinction between cells with CP higher or lower than 0.5" We struggled to find a clear and unambiguous summary of the statistics associated with this. We can see that a consistent pattern emerges when the cluster number is increased from 2 to 3, but looking at the distributions in Figure 4c and Figure 4d does not appear to reveal clusters. The significance values in Figures4a and 4b relate to the significance of the modulation effect for each cluster, not the significance of the cluster separations. The methods section and supplementary analysis section cross-reference each other but neither seems to answer the simple question of whether 1 versus 2 clusters is statistically justified, let alone the step from 2 to 3.

13. Lines 612-638 This discussion suggests that there may be within-cell changes in CP as a function of pCR that may have been hidden by pooling across populations of cells. But in the end, this paper has problems in detecting real changes in this relationship at the level of recordings from single cells. The small SEMs that attach to the data in Figure 3a do indeed reflect pooling across a population sample; they do not relate to changes in individual neurons. The panel in Figure 3d shows the true picture for individual cells. So this discussion begs the question as to why the population analyses in Dodd et al. do not show the predicted relationships. The point made by the authors about within-cell changes does not appear to be material in this regard.

---

## [Author Response]

Essential revisions:This manuscript further develops previous theoretical findings concerning "choice probability", the relationship of the trial-to-trial variation of firing rates of a (sensory) neuron and the behavioral choice of a subject in perceptual decision making experiments. CPs and their interpretation have a long history, including the debate about their origin (do CPs reflect a causal effect of sensory neural variability on the decision or do they arise from top-down feedback signals?) Here, the authors identify an additional factor by showing that CP depends on the "choice rate" (the fraction of left vs. right choices). Since this choice rate depends on stimulus strength, CP itself will depend on stimulus strength. They also show experimental evidence for this effect (in the classical Britten et al. dataset) that has been overlooked until now and present methods for refined analysis for linking the responses of sensory neurons to choices.The reviewers raised a number of central concerns about the relationship between the model and the available empirical data. They agree that the relationship between pCR and CP must be firmly established as a general experimental fact, or reasons given why it is apparent in some conditions and not others. Also, the theoretical interpretation of the relationship between pCR and CP needs to deliver accurate predictions of the experimental results and also to demonstrate consistency with theoretical models of how other factors affect CP (interneuronal correlation and stimulus sensitivity, being the primary variables of concern here).1. The relationship between pCR and CP appears more at the core of this rather than the related stimulus strength vs CP, but there is a quantitative mismatch between model prediction and empirically observed data, especially in Figure 3a and Figure 4b. Interpretation of Figure 3a. Our initial concern when reading the paper was that the predicted effect of choice rate on CP (dotted line in Figure 3a) is so small that there will not be enough statistical power to detect it in the available experimental data. Moreover, the analysis is complicated by the fact that the effect of h(pCR) is partly masked by the gain modulation effect (Figure 4b). However, for the n=48 neurons (Figure 4b) that show the predicted U-shape of CP vs. pCR, the effect is actually much larger than the theoretical prediction (e.g. or pCR ~0.1 the CP is close to 0.65 when in theory it should be ~0.58). What could be the reason for this quantitative mismatch?

We now better clarify that the CP(p_CR_) dependency is the result of two factors: h(p_CR_) shared by the entire population, and CC(p_CR_) which is cell-specific. Our model predicts the former and allows us to infer the latter. These revisions are reported in lines 160-164 and 170-174.

We thank the Reviewers for the opportunity to clarify these important points. As we now discuss in lines 170-174, we refer to a CP(s) stimulus dependence or CP(p_CR_) dependence interchangeably, since most often -and in particular in our case- p_CR_ values change due to changes in stimulus levels, and hence the two quantities are intertwined and monotonically related. As discussed in lines 160-164, according to the model in Equation 7, a dependence of the CP on the p_CR_ -and by extension on the stimulus level- may appear in two ways. First, through the modulation factor h(p_CR_) which arises due to the thresholding operation when converting an internal, continuous estimate of the stimulus into a binary decision. Independent of that, it can appear in a second way, due to a stimulus-dependent choice correlation CC(s), for instance as the result of stimulus-dependent cross-neuronal correlations. Importantly, the U-shape due to the factor h(p_CR_) is common to all cells, while the CP stimulus dependencies induced by CC are cell-specific (as motivated in lines 164167 and lines 672-680). This makes both effects dissociable empirically – at least in principle.

We agree with the Reviewer that the modulation induced by h(p_CR_) is small, and therefore is challenging to detect it in existing experimental data. This is why we developed our refined methods, which allow examining within-cell CP modulations by the p_CR_. As we argue in lines 195-202, for each individual cell i the CP_i_(p_CR_) profile jointly reflects h(p_CR_) and any potential dependence CC_i_(p_CR_). The average across a population of neurons will therefore average out cell-specific CP_i_(p_CR_) dependencies associated with CC_i_(p_CR_). This will make the U-shape dependency due to the h(p_CR_), which is shared by the entire neural population and does not average away, more visible.

Empirically, we found that the average <CP(p_CR_)> in Figure 3 has an asymmetric dependency that goes beyond that predicted by the h(p_CR_) factor, which motivated us to use a cluster analysis to further separate h(p_CR_) from cell-specific contributions through CC(p_CR_) and to shed further light on general patterns of CC_i_(p_CR_) dependencies in the data.

Still, the cluster analysis cannot by itself separate h(p_CR_) from other CC(p_CR_) patterns that have the same shape (that is only possible by changes in the experimental design to dissociate stimulus strength from p_CR_). It only separates the most predominant profiles of CP(p_CR_) dependencies across cells. This means that the symmetric dependency of cluster 2 in Figure 4B may result not only from h(p_CR_), but also from other symmetric CC(p_CR_) patterns shared by the cells. Its symmetric shape is compatible with the symmetric shape of h(p_CR_) predicted by the model. However, as pointed out by the Reviewers, there is a quantitative mismatch in its magnitude. We now address this mismatch in the revised text (lines 424-431). First, the mismatch can be due to additional symmetric dependencies through CC(p_CR_) patterns that are also shared across the cells. Second, the quantitative mismatch can be due to the abstraction of the decision-making dynamics necessary to derive the analytical model. The analytical framework of the model, as developed by the work of Shadlen et al., 1996, Haefner et al. 2013, and others, constitutes an abstraction of any biological implementation of the decision process. The quantitative deviations could be caused by effects neglected by the model, such as a dynamic feedback contribution amplifying the U-shape dependence. In this regard, we consider that the qualitative prediction of the U-shape of h(p_CR_) is itself remarkable, given the degree of abstraction of the CP model.

Furthermore, our empirical characterization of existing patterns of CP-stimulus dependencies and our theoretical analyses of their contributing factors and predicted shapes are themselves original contributions, because these dependencies have mostly been unnoticed in previous analyses or have been ignored because no interpretation was available. Although motivated by the challenge of testing the prediction of the U-shape modulation, our new refined methods to examine within-cell CP(p_CR_) profiles are agnostic about their origin, and as we have shown provide a powerful tool to characterize this dimension of the cells choice-related neural responses that has not been thoroughly explored yet.

Importantly, despite focusing on the two main predominant patterns of the cluster analysis, we do not claim that those are the only patterns present. Conversely, as we discussed in lines 672-683, previous theoretical and experimental work would support the presence of a richer structure of CP(p_CR_) dependencies, which we would not be able to characterize because of the limitations of the data in terms of number of trials and number of cells. However, with rapidly improving recording technology allowing for chronic recordings (i.e. many more trials than currently) in addition to recording many neurons concurrently, we expect future studies to map this structure in much greater detail resulting in insights into the functional role and place in the microcircuit of the different clusters of neurons exposed by our analysis.

We would like to see in this regard:a. A clear discussion of and situation of the results within the available empirical literature on CP, especially with regards to quantitative effect size in Britten et al. 1996 and Dodd et al. 2001.We agree that in some previous papers CP is computed from z-scored spike counts across different stimulus conditions (and thus different choice rate values). However, there are several other papers (including Britten et al. 1996, Nienborg and Cumming 2009, Wimmer et al. 2015, Katz et al. 2016) that compute CP from "zero coherence trials" with choice rate ~ 0.5 (or the equivalent for binocular disparity in V5/MT: Dodd et al. 2001; Krug et al. 2004; Wasmuht et al. 2019). These papers should be mentioned in order to avoid the impression that all previous work suffers from a (slightly) incorrect way of computing CP.

We now discuss prior studies more accurately, by better delineating their respective contributions and by distinguishing between those reporting CP(s=0) and those computing a Grand CP. We clarify that our primary goal is to advance and promote the analysis of the CP dependence on p_CR_/stimulus level based on a deeper understanding of the nature of decision-related signals (both choice and detect probabilities) in sensory neurons. Both Grand CPs (computed in various ways) and CP(s=0) are scalar summaries of the richer and more informative underlying function that considers the CP changes across stimulus values. We clarified the importance of isolating within-cell modulations of the CP by p_CR_ from the across-cell heterogeneity of CP magnitudes, which is key for studying the within-cell patterns of CP as a function of p_CR_. These revisions are reported in lines 37-44 and 57-59.

Thanks to the reviewer for pointing out the necessity to further clarify how we propose to study CP(p_CR_) patterns building on previous studies. We now cite this previous work in lines 37-44, by better recognizing the importance of previous contributions and by better specifying that in many cases CPs have been calculated only from noninformative stimuli, or alternatively weighted averages or corrected z-scoring are used to combine data across stimulus levels. We emphasize the novelty with respect to these alternatives in our consideration of CP stimulus dependencies (lines 57-59). We also emphasize that our proposal to construct within-cell CP(p_CR_) profiles is not meant to serve only -not even primarily- as a way to improve the estimation of a single grand CP value per cell. The calculation of a single CP from trials with noninformative stimuli is meant to study choice driven neural responses isolated from stimulus-driven signals. Conversely, as discussed in lines 57-59, 683-690, our aim is to (a) theoretically motivate the need to study the shape of the CP(p_CR_) patterns, and (b) demonstrate our power to do so with refined methods using a classic and public dataset as an example.

We have now improved the description of how our refined analysis differs from previous analysis in Britten et al. 1996 (lines 616-621). We have also incorporated to this comparison the important work of Dodd et al. 2001 (lines 621-633). In the original submission we explicitly referred to the key differences with respect to the analysis in Figure 3 of Britten et al. only in Discussion. Now we expanded the description of the key steps of our analysis (lines 271-307) to more clearly indicate why it should improve the characterization of within cell CP dependencies on p_CR_. The key difference with respect to Figure 3 of Britten et al. 1996 and Figure 6 of Dodd et al. 2001 is that in our analysis we separate the within-cell modulations of the CP by p_CR_ from the across-cell heterogeneity of the CP magnitudes. Conversely, in Britten et al. 1996 an average of CP values across cells was carried out at each stimulus level without ensuring that the same cells were contributing to each of these averages. Similarly, in a scatter plot of the form presented in Figure 6a of Dodd et al. 2001, the cell-identity of each dot is lost, and hence it is not possible to trace the within-cell profile CP(p_CR_), making it harder to discriminate within-cell modulations from across cells CP heterogeneity. The construction of CP(p_CR_) profiles for each cell isolates within-cell modulations. The fact that the modulation h(p_CR_) is multiplicative, as we indicate in lines 189-190 and 275-277, makes the detection of modulations even more sensitive to the isolation of within-cell comparisons, since the magnitude of any modulation is relative to the magnitude of the CP value obtained with the noninformative stimulus. Furthermore, CP(p_CR_) profiles also have the advantage that they can subsequently be used -with cluster analysis as we do, but also with other pattern recognition methods- to characterize which patterns of CP stimulus dependencies are predominant across cells.

As we explain in lines 297-307, an important step when averaging across cells is that we need to ensure that our average CP profiles -as presented in Figure 3- really correspond to an average of the individual within-cell CP profiles of the cells included in the analysis, and are not the result of different subpopulation of neurons contributing to the average at each dot of the curve. That is, our average should correspond to an average of the within-cell CP modulations, and be isolated from heterogeneity in the CP magnitude across cells. This distinguishes our refined analysis from the type of previous analysis used in Britten et al. 1996 (Figure 3) or Dodd et al. 2001 (Figure 6), in which across-cell CP variability may hinder within-cell modulations. We have discussed these differences between these previous works and our new method in detail in lines 616-638. In addition, we also have improved the comparison of our analysis with the one of Britten et al. in the Introduction (lines 67-69). In particular, we corrected a key problem in the old text (old lines 71-74 in the previous submission) that misleadingly suggested that the key feature of our novel method was the separation of cells of opposite choice preferences. We now clarified that they key feature is, more broadly, the characterization of within-cell CP profiles.

b. Re-analysis of the main effect in different subsets of cells. We are concerned that modelling the group for which CP < 0.5 includes a lot of noise, because the members of this are almost always non-significant, as shown in Dodd et al. and in Britten et al. (See their Figure 5). If you look at Dodd et al., there is no sign of this relationship between pCR and CP. Moreover, in Dodd et al. there are very few 'wrong way' choice probabilities: so few in fact that they are within the statistical bounds of repeated measures. So the procedure of separating the CPs into >0.5 and <0.5 is not likely to alter the result at all from the published Figure in Dodd et al. It might be helpful to recalculate the results for Figure 3A but only including those neurons that show an individually significant CP > 0.5.

We completely agree with these suggestions. We note that cells with CP<0.5 were already excluded for our further cluster-based analysis, and this is better specified in revision. We also agree that is useful to better discuss the implications of Dodd’s work on our results and the suggestions that arise from our work on how to extend Dodd’s previous work. These issues have been clarified in revision. These revisions are reported in lines 330343, 297-307, and 616-633.

This is a good suggestion, and we agree that this issue needs clarifications. We agree with the Reviewers that "a separation of CPs into >0.5 and <0.5 is not likely to alter the result at all from the published Figure 6 in Dodd et al. 2001". However, as discussed above and now emphasized in lines 297-307 and 616-629, apart from the separation between cells with CPs >0.5 and <0.5 another important component of our refined analysis is to isolate within-cell CP(p_CR_) profiles from the across cells CP heterogeneity. In Figure 3A of Britten et al. 1996 the fact that the average of CPs at each coherence level is calculated without a separation of CPs into >0.5 and <0.5 affects the identification of the U-shape, but this lack of separation is not the only factor hiding the CP(p_CR_) patterns. In more detail, in the data of Britten et al. 1996 the set of coherence levels used varies across cells, which means that a different subpopulation of cells is contributing to each dot of their Figure 3A, so that the analysis of within-cell CP(p_CR_) profiles is lost. This is the reason why in our analysis we take special care that the same neurons are averaged at each p_CR_ level, even at the cost of excluding a portion of the cells from the analysis because a CP value cannot be calculated for all p_CR_ levels.

In the case of Dodd et al. 2001, all the results of the paper rely on the analysis of CPs from zero disparity trials, and hence their main conclusions stand and are not affected by our new advances. We make this point clear in our reference to Dodd et al. 2001 (lines 629-633). The point of their work that could be revisited at the light of our advances is that in the specific case of their Figure 6A the capacity to identify CP(p_CR_) dependencies may have been affected by the lack of isolation of within-cell CP(p_CR_) modulations. From our understanding of the experimental procedure of Dodd et al. 2001, the range of disparity levels changes from unit to unit, being usually five or seven. This means that different cells may be contributing more dots in a certain range of the y-axis (% Choice PREF). Moreover, it is not possible to identify the within-cell CP(p_CR_) profiles from their published scatter plot. Furthermore, in our understanding, Figure 6A includes a dot for each stimulus condition in which the animal made at least one incorrect response. As Dodd et al. indicates, this is reflected in the increasing spread of the values at the top and the bottom of the ordinate axis of Figure 6A. We attenuated this kind of problems in our refined method by constructing p_CR_ bins within which a weighted average of several CPs is calculated (lines 308-318). Because the CP(p_CR_) profile is a vector constructed using 5 bins, this means that the estimate for each bin is already more reliable than individual estimates for single stimulus levels.

Altogether, our method was specifically designed to better estimate within-cell CP(p_CR_) profiles, and this analysis may reveal some structure difficult to appreciate from the joint scatter plot of CPs from all cells. We now indicate this point also with regard to the discussion of Figure 6 of Dodd et al. 2001 (lines 621-633).

[Speculative note: We think it would be very interesting to collect more data in the experimental paradigm of Dodd et al., ideally with chronic recordings of many neurons at the same time. Comparing the results of our analysis of the CP(p_CR_) dependency for such a dataset with one collected in a more conventional discrimination task (e.g. motion direction discrimination) might provide deeper insights into the perceptual decision-making mechanisms underlying both, and how they might differ. For instance, one might find that the h(p_CR_) dependency is absent in the rotating cylinder task suggesting that the decision in that task – unlike in a motion direction discrimination task – is not mediated by a continuous internal estimate that is being thresholded.]

We also agree with the reviewers that "modelling the group for which CP < 0.5 includes a lot of noise". Indeed, in lines 330-339 we indicate that the fact that we do not find a modulation consistent with the factor h(p_CR_) significant for the cells with CP<0.5 can be explained by two factors. One of the factors is that fewer cells are included in this group, so that the estimated average CP profile is noisier. The other factor is related to the point of the Reviewers, namely the fact that the h(p_CR_) modulation is multiplicative to CP-0.5, and hence if few cells in the group of CP<0.5 have a CP magnitude significantly different from 0.5, then CP-0.5 = 0+noise, and a consistent effect of the multiplicative factor h(p_CR_) cannot be isolated when averaging across cells. We now further highlighted (lines 339-343) this fact that as pointed out by the Reviewers "CP < 0.5 includes a lot of noise" and this explains why the prediction of the U-shape modulation could only be corroborated to be significant for the cells with CP>0.5. As a result, the lack of a significant inverted U-shape modulation for the cells with CP<0.5 does not constitute evidence against our model. Furthermore, note that the cells with CP <0.5 do not affect the subsequent discrimination between a symmetric and an asymmetric pattern in Figure 4b (cluster 1 is the same in Figure 4a and 4b), since they are already excluded for that analysis, as we now highlight (lines 386-387).

c. A simulation of the factors that affect expected effect size. The simulation should show what size of the effect is expected given the limited amount of noisy data (number of neurons, trials, etc.). This would allow you to determine whether the experimental result lies with the confidence bounds of the theoretical prediction.There is apparently no such relationship between CP and p_CR_ in the Dodd et al. data (but it could perhaps show up when selecting and binning the neurons as described in the manuscript, line 288). In any case, a possible explanation for the discrepancy could be that the expected relationship is so small that it cannot reliably be detected from ~100 neurons. We also think also this issue can be clarified by simulating the size of the effect for limited data.

We now include a power analysis based on simulations that provides a clearer intuition about the expectations for effect size as a function the number of trials, the number of neurons, and the magnitude of the CPs. These simulations are reported in lines 1238-1276.

We thank the Reviewers for the suggestion to perform simulations that provide additional intuition of the expected statistical power for the detection of the CP modulation h(p_CR_). We now implemented simulations of how the detection of a significant modulation h(p_CR_) depends on the number of trials, the number of neurons, and the magnitude of the CPs (see new section S1.2 in the Supplementary Material). As expected, these simulations show that increasing the number of cells and of trials used to estimate an average CP(p_CR_) profile increases the statistical power for the detection of a CP modulation. This indicates the utility of our refined analysis of within-cell CP(p_CR_) dependencies, which can substantially improve the statistical power of the analysis by efficiently combining CP values, both across stimulus levels and across cells. Most importantly, in our method CP values are combined while preserving the characterization of within-cell CP(p_CR_) profiles.

These simulations also indicate that the p-values obtained for the experimental data, in particular p = 0.0008 for the red curve of Figures 4B, are smaller than predicted from the model. This new analysis complements the comparison of the experimental effect size (red curve) and predicted effect size (dashed black curve) in Figure 3A, and in general the comparison of the experimental results (Figures 3-4) with the analytical effect sizes displayed in Figure 2, which already pointed to the higher magnitude of the experimental effect size. In the revised paper we now address this quantitative mismatch between the predictions of the h(pcr) modulation and the CP(pcr) pattern obtained for the symmetric cluster (red curve) in Figure 4b (lines 424-431).

As already discussed above in a previous reply, the quantitative mismatch can be explained either by the presence of further symmetrical sources of CP stimulus-dependence through CC(p_CR_), and/or dynamic feedback amplifying the threshold-induced U-shape dependence. Given the level of abstraction of our CP model, we think it is remarkable that qualitative evidence compatible with h(p_CR_) is found. Furthermore, independently of the quantitative fit to the prediction of the h(p_CR_) modulation, it is in itself an achievement of our method that the characterization of within-cell CP(p_CR_) allows identifying any existing pattern of CP stimulus dependence, since their existence had been previously unnoticed. This identification is a first step towards inferring the underlying mechanisms producing a characteristic structure of CP(p_CR_) patterns, as we discussed in lines 672-683.

Finally, we agree with the Reviewers that our analysis applied to the Dodd et al. data may show the h(pCR) dependence (but see the speculative note above).

2. Clarifications are further needed in term of how the fluctuations in the stimulus-related gain of neuronal firing are responsible for the emergence of stronger CPs at higher performance levels. As we understand it, this is required both in terms of the formal implementation in the model and in terms of the proposed neurobiological implementation.The authors present a detailed analysis of abstract decision-making models. They relate noisy neuronal responses ri, a hypothetical covert decision variable d and decision threshold θ, and an overt behavioural choice D. The authors assume a bivariate Gaussian association between ri and d, with a certain correlation coefficient, and from this minimal basis derive exact or approximate expressions for choice rate pCR, choice probability CPi, choice correlation CCi, between ri and d, and choice-triggered averages of ri, CTAi , and of d, CTAd. The treatment extends previous work in that it covers the entire range of choice rates pCR, not only the special case pCR = 0.5.A key result is that choice probability CPi changes multiplicatively as a function of pCR, increasing as the decision grows more consistent in either direction, with the baseline level set by choice correlation CCi. An important implication is that the dependence of CPi on pCR, which is shared by all cells and expected to be U-shaped, can be averaged over cells with different choice correlation CCi, provided that cells with positive and negative choice correlations are distinguished.To test these predictions, the authors re-analyze MT recordings from Britten et al. (1996) and were able to confirm a U-shaped dependence of average CP on pCR, which was statistically significant for cells with positive CCi. However, contrary to predictions, the U-shaped dependence was asymmetric and more pronounced when the more frequent choice (pCR > 0) is consistent with the preferred stimulus of the cell (positively correlated cells CCi>0). A cluster analysis of empirical individual cell dependencies of CPi on pCR revealed that, in addition to the predicted U-shaped dependencies, the presence of unexpected monotonically increasing dependencies.To clarify the origin of these unexpected dependencies, the authors consider the effect of trial-to-trial response gain fluctuations (Goris et al., 2014) and, with the help of the model of these authors, confirm that gain fluctuations account for 62% of the observed trial-to-trial variance. The authors point out that gain fluctuations add a stimulus-dependent component to the noise covariance of neural responses, which is inherited by choice correlation CCi and by choice probability CPi. In Methods and Supplementary Text S4, the authors derive that this stimulus-dependent component can itself depend asymmetrically on pCR, to an extent that is specific to each cell (i.e., the specific coupling between response and gain). Unfortunately, the authors do not offer an intuitive argument about the origin of this asymmetry.Please explain how gain fluctuations lead to rising dependencies on pCR in some cells. Which cells are these? Do they combine strong stimulus modulation with weak choice correlation? Without such an explanation, the entire cluster analysis appears incomplete and ultimately pointless.

We now provide a more complete description of the gain modulation model and how it relates to the results of the cluster analysis in the Results section. Some of the final questions raised by the Reviewers are excellent ones but will require data to characterize the cells and their responses (e.g. structure of cross-neuronal correlations? what layer? Excitatory/inhibitory? Target of feedback projections? Projecting to other cells within the same column/the same area/other cortical areas/thalamus/…?) beyond those available in Britten et al. data. We performed the cluster analysis to demonstrate that there is structure in the CP(pCR) dependencies, and that beyond the modulation h(p_CR_) common to all cells there is additional structure that is cell-specific and can be expected to be related to the cell properties mentioned above. Now we’re handing this back to the field to figure out how these structures correlate with other properties or structures since we believe that this will provide insights into the neural circuits underlying perceptual decision-making. Together with more fine-grained improvements reported below we specifically improved the presentation of the gain model in Results (lines 205-256), Methods (lines 882-957) and Suppl. Information S4.

Thanks to the Reviewers for the summary of our findings. We now better explain that the finding of an asymmetric component of the CP(p_CR_) dependence is not "contrary to predictions" of our model, as presented in Eq 7. We indicate in lines 160-164 that the CP(p_CR_) dependence can be caused by the threshold-induced factor h(p_CR_) but also by choice correlations CC(p_CR_). We indicate in lines 197-200 that averaging across neurons is expected to help to isolate the h(p_CR_) pattern because it is common to all cells, while cell-specific patterns are expected to be introduced by CC_i_(p_CR_). The identification of the U-shape of h(p_CR_) relies on the assumption that CC(p_CR_) is stimulus independent or that the CP(p_CR_) dependencies induced by CC(p_CR_) are sufficiently heterogeneous across cells so that when averaging across cells they average out and the predominant modulation observable in the average profile <CP(p_CR_)> is h(p_CR_). The fact that we find the asymmetric component of CP(p_CR_) is against this assumption, suggesting that there is a shared pattern of CC(p_CR_) among part of the cells (lines 349-353), but not against the predictions of our model. Indeed, as we discuss in lines 164-167 and 672-680, there is evidence from previous theoretical and experimental work suggesting that choice correlations CCs have cell-specific stimulus dependencies, and hence given Eq 7 it should be expected that also the CPs have cell-specific stimulus-dependent patterns. As we discuss in lines 683-692, our refined method to characterize within-cell CP(p_CR_) should help to examine this dimension of the data that has not been thoroughly explored yet, and which can provide insights into the nature of the underlying decision-making mechanisms beyond what is captured by a single CP value.

As mentioned above, the finding of the asymmetric pattern is evidence of a stimulus dependence of the CC. This raises the question of which can be the origin of this component of the CC stimulus dependence that is shared by the neurons. As we mention in lines 164167 and 672-680 it can be expected that CC stimulus dependencies have a rich structure reflecting the structure, for example, of feedback connections. However, as indicated in lines 458-460, to examine the structure of CP(p_CR_) patterns associated with the computational decision-making mechanisms would require a characterization the cross-neuronal correlation structure of the responses which is beyond the single cell recordings of Britten et al. This is why we chose to examine if the asymmetric component of the CP(p_CR_) could be explained by a non-specific mechanism for which prior empirical support exists (Goris et al. 2014), such as the presence of trial-to-trial gain fluctuations.

We have now substantially improved the description of the gain model (lines 205-256) and add new analyses quantifying how well the gain model can predict the experimentally observed asymmetric CP pattern extracted from the cluster analysis (lines 443-460). In particular, we now report also that the theoretical coefficients estimated from the gain model significantly correlate with the experimentally estimated coefficients for the asymmetric cluster, although overestimating their magnitude.

The separation in a new Section for the gain model (lines 205-256) helps to clearly differentiate between the generic model introduced before, and the specific model that considers gain fluctuations as the source of a CC(p_CR_) contribution. In this section we describe the main components of the model and introduce the resulting analytical CP expression (Eq 11), which in the previous version was only provided in Methods. For this model of gain fluctuations, we specifically adopted a purely feedforward encoding/decoding model, as previously studied in Shadlen et al. (1996) and Haefner et al. (2013). Equation 9 is a generalization of the CP expression of Haefner et al. (2013) to all stimulus levels. We now further explain this extension in connection with the previous results of Haefner et al., which characterized the dependence between CPs and properties such as the neurometric sensitivity and the cross-neuronal correlation structure (lines 218-227).

In more detail, we consider that the neural decoder that determines the choice is tuned to be optimal at the decision boundary, that is, that the read-out weights are tuned to the structure of covariability in the population responses for the non-informative stimuli. For this optimal read-out, the CP is proportional to the neurometric sensitivity (Haefner et al. 2013, Pitkow 2015). However, in the presence of gain-fluctuations, this structure of covariability changes for different stimuli -due to the additional gain-related component in Eq 10. We now explain in more detail how the CC stimulus dependence appears as a consequence of the stimulus dependence of the gain-related component of the cross-neuronal correlations in Eq 10 (lines 234-243).

We have also improved the description of the formal implementation of the model in Methods (lines 882-957) and in section S4 of the Suppl. Material. In Eq 23 and lines 924-934 we explain how the changes in the cross-neuronal correlation structure of Eq 22 create additional response variability that attenuates the CC but also create a new component of covariation between the decision variable and the single-cell responses due to the shared gain fluctuations common to the neural population. A more detailed derivation is now extended in section S4.

Moreover, to further understand which properties of the single-cell responses and of the decoder determine the form of the asymmetric gain-induced CP(p_CR_) component, in S4 we further generalize the model (lines 1468-1484). This generalization is valid not only for an optimal decoder but for any unbiased decoder. We describe how the stimulus dependencies of CC induced by the gain are determined In Equation S12. The shape of these dependencies depends on cell-specific properties, namely on the neurometric sensitivity of the cells and on the relative contribution of the gain to their variability, as well as on population properties, such as the behavioral sensitivity of the decoder to changes in the stimulus, and on the relative contribution of the gain to the variability of the decision variable. In the particular case of the model with an optimal decoder in which we focus in the main article, as explained in lines 244-245 for each cell the strength of the asymmetric component is determined by the CC magnitude for p_CR_ = 0.5 as well as by the relative contribution of gain to their variability (λ_i_).

Additionally, to further connect our model to previous computational (Shadlen et al. 1996) and analytical (Haefner et al. 2013) studies of the relation between CPs and the structure of cross-neuronal correlations, we now analyzed in detail the effect of gain fluctuations when the decoder is formed by two pools of neurons/antineurons with opposite choice preference (lines 1427-1452).

While the gain model explains main features of the CP(p_CR_) patterns observed with the cluster analysis we would like to highlight that the value of the cluster analysis stands independently of modeling the observed patterns with the gain model. This is now clearer since the results of the cluster analysis and the gain model are in separate subsections. As we now discuss in lines 349-353 and 381-382, the cluster analysis identifies the existence of a statistically significant asymmetric CP(p_CR_) pattern together with the symmetric pattern that the threshold-induced modulation h(p_CR_) predicts. The presence of this asymmetric pattern is a signature of the existence of stimulus dependent choice correlations CC(p_CR_), something that had not been identified in previous studies of CPs, but that is consistent with both theoretical and experimental work (lines 672-680) which studies the structure of stimulus dependent decision-related feedback signals, and more broadly stimulus-dependent cross-neuronal correlations. In this sense, our cluster analysis resolves the apparent contradiction between expectations from this bulk of previous work and the lack of evidence of CC stimulus dependencies.

A full characterization of the CP(p_CR_) patterns in relation to cell types would require a joint characterization of properties such as the connectivity structure and cross-neuronal correlation structure, which are not available from the single unit recordings of Britten et al. 1996 (lines 458-460). Despite this limitation, the cluster analysis is useful because it reveals a significant structure of CP(p_CR_) previously unnoticed, and validates the use of within-cell CP(p_CR_) profiles to study the interaction between stimulus-driven and choice-driven signals in neural responses. Similarly, the gain model we present shows that an asymmetric component of CP stimulus dependence can be caused by a generic mechanism such as trial-to-trial gain fluctuations, and is informative about the relative strength of the asymmetric CP stimulus dependencies for the cluster in which the asymmetric dependence is predominant (lines 443-460), although overestimating its magnitude. We expect that the analysis of within-cell CP(p_CR_) patterns will in future work be a useful tool to identify a finer structure of CP stimulus dependencies across cell, distinctive of specific mechanisms of the decision-making process, such as the structure of stimulus-dependent decision-related feedback.

3. Following on from this, there is also the question how could the stronger CP with increasing pCR be implemented in different ways in actual neural terms. Would for example the pool size of neurons contributing to the decision change? With regards to the model, how does the effect of pCR x CP intersect with the experimentally much stronger interaction effect of neurometric sensitivity and size of the CP?

The h(pCR) modulation that we derive does not require any change in

“implementation”, pool size or membership of the decision pool. This modulation is already embedded (undescribed) in the Shadlen et al. 1996 model and in the Haefner et al. 2013 model when considering a p_CR_ different than 0.5. Our results extend the results in Haefner et al. to pCR0.5 and thereby open the door to a principled study of the CP(pCR) relationship. In fact, without our results – both on h(pCR) and the impact of the known gain variability – a measurement of a U-shaped CP(pCR) or monotonic CP(pCR) relationship might lead to suggestions that pool size and/or pool membership must be changing with pCR/stimulus (which appears rather unlikely given that this would imply a change of the decoder driven precisely by the property to be decoded). These revisions are reported in lines 210-231 and 1405-1415.

The authors analyse this from the classical Britten et al. [1] data set, producing a framework for analysis in Figure 2 and the outcome of analysis in Figures3 and 4 of the paper. This type of analysis is not new. The specific form of the plots in Figures2,3,4 appears in Dodd et al. [2] in Figure 6, while Figure 3 in Britten et al. delivers almost the same information.

We respectfully disagree with this summary. All panels of Figure 2 represent novel mathematical results and insights that have not previously appeared anywhere. While Britten et al. and Dodd et al. present plots similar to those in Figure 3, our method introduces a key new refinement separating within-cell CP patterns of stimulus dependence from the across cell heterogeneity of CP magnitudes. Please find above in reply to point (1a) of the Reviewers a more detailed discussion of how we now improved the comparison with these previous studies. A comparison with these Figures from Dodd et al. and Britten et al. is now discussed in the Discussion, lines 612-638.

In regard to the clustering analysis, it seems that the big driver for the formation of clusters is the division between CP >0.5 and CP<0.5. For values of CP<0.5, there is not really a functional account of these, as they do not relate to the tuning of the cells for motion. The lack of functional meaning is highlighted by the fact that cluster 1 in Figure 4a (blue==CP <0.5) is statistically non-significant. Not unrelated to this lack of significance is the fact that Figure 13 of Britten et al. and Figure 6 of Parker et al. [3] show that CPs are stronger for neurons that are more sensitive to the visual task. The usual interpretations of this are either the intuitive claim that more sensitive neurons are more tightly involved in the task and therefore have higher CPs or, more subtly, that neurons with weaker sensitivity have lower degrees of interneuronal correlation.

Please see our previous answers regarding CP<0.5 in reply to point (1b) of the Reviewers. The reviewer’s points about the relationship across cells between the magnitude of the CP and the neurometric threshold (d’) is orthogonal to the analysis of the within-cell dependence of CP as a function of p_CR_ that we focus on here. Put differently, a single neuron’s CP magnitude may be a function this single neuron’s d’ and additionally is modulated by the p_CR_, which is shared by all neurons by virtue of being a behavioral quantity. Furthermore, as we now explained in our improved description of the gain model, the neurometric sensitivity also affects the strength of the CP stimulus dependence induced by the gain.

The rest of the analysis in this paper advances the idea that fluctuations in the stimulus-related gain of neuronal firing are responsible for the emergence of stronger CPs at higher performance levels. The authors write on ll243-4 that "Briefly, the contribution of gain modulations to the covariance of the responses increases with neuronal firing rates, which in turn are stimulus-modulated as determined by tuning functions.". However, the lead author has already published a nice theoretical summary [4] showing that CP is related to the level of interneuronal correlations in the pool. Indeed, the analysis showed that under some conditions (large pool, correlated noise and at least one or two members of the pool contributing significantly to perceptual read-out) one might take CP as substitute indicator for the interneuronal correlation of the decision pool. In the light of the earlier analysis, the present paper does not address the very relevant question of changes in the membership of the neuronal pool with stimulus strength. In effect, if read-out weights change with stimulus strength, then CP will be expected to change. Equally, if pool membership changes then interneuronal correlation may be expected to change. We did not see anything in this analysis here that definitively ties down the change in CP to stimulus-related gain changes.l 242-243: " We derived the specific form of CC𝑖(𝑝CR) predicted from gain modulations in the threshold decision model to explain additional CP stimulus dependencies beyond the symmetric modulation by h(𝑝CR ). Briefly, the contribution of gain modulations to the covariance of the responses increases with neuronal firing rates, which in turn are stimulus-modulated as determined by tuning functions. This leads to an asymmetric component of CP(𝑝CR), with higher CPs for 𝑝CR values associated with stimuli preferred by the cell. Furthermore, while stronger gain fluctuations increase this asymmetric stimulus dependence, they also decrease the magnitude of the cell-specific CP because they add variability to the responses unrelated to the choice "

The gain model examines the effect on CP(p_CR_) of a particular structure of stimulus-dependent cross-neuronal correlations, namely produced by shared gain fluctuations across neurons. We clarified the advances with respect to Haefner et al. 2013. We assume that only the cross-neuronal correlations are stimulus-dependent, while the read-out weights are independent of the stimulus. We believe this assumption is reasonable since the cross-neuronal correlations depend on the dynamics of the network, while the weights are expected to be hardwired in the network structure and the decoder is not expected to depend on the very sensory stimulus it has to decode. We did not claim (nor could we given the available data, as explained in the ‘In Brief’ summary in replied to point 2 above) that gain variability is the only explanation for a monotonic increase of CP with stimulus strength. What we show is that the gain model explains main features of the patterns of CP stimulus dependence observed with the cluster analysis. We have reinforced these conclusions with additional analysis of the predictive power of the gain model (lines 443-460). The paper has been revised to make the gain model clearer (lines 205-256, 882957).

As neuronal sensitivity has normally been measured, the change in response due to a change in the stimulus is assessed relative to the variability of neuronal firing. The description above implies that stimulus modulations in the Haefner models translate into response changes on top of which random gain modulations are applied. At first sight, there does not seem any room in the model for low firing rate, low variability neurons to contribute to CP, even though such neurons may have high neurometric sensitivity. One the other hand, it may well be that the new Haefner model all shakes down to give the established experimental result that CP is linked to neuronal sensitivity. If that's correct, then the paper is currently rather obscure on this point and it will be useful for the paper to lay this out clearly.

As mentioned above, our model does not amend the findings in Haefner et al. 2013 that support with their analytical CP model the experimental finding that CP is proportional to the neurometric sensitivity. Haefner et al. derived that relation for p_CR_ = 0.5 and it equally holds in our extended model, hence supporting that neurons with low firing rate and low variability contribute to the internal decoder. Using this suggestion and other suggestions (see reply to point 2) we have improved the description of the gain model to better explain the role of the neurometric sensitivity both determining the CP magnitude at p_CR_ = 0.5 and the strength of the modulation induced by the gain. The connection with the neurometric sensitivity can be found in lines 220-227 and 1405-1415.

We now provide an expanded discussion of all the points raised by the Reviewers in point 3.

As detailed in reply to point (2), we have extended the analysis of the CP stimulus dependency patterns extracted from the cluster analysis jointly with the predictions of the gain model (lines 443-460) and substantially rewritten the presentation of the CP gain induced model in the main text (lines 205-256), Methods (lines 882-957), and Suppl. Material (section S4). We now better explain the role of single cell properties, namely the neurometric sensitivity of the cells and the relative contribution of the gain to their variability, as well as population properties, such as the behavioral sensitivity of the decoder to changes in the stimulus, and the relative contribution of the gain to the variability of the decision variable.

As indicated in lines 59-61 and 125-128, our general model is agnostic with respect to the feedforward or feedback origins of activity-choice covariations. The only assumption is that "the link between sensory responses and choices is mediated by a continuous decision variable and a thresholding mechanism" (lines 529-530). This general model (Eq 7) identifies two sources of CP stimulus dependencies (lines 160-164). The stereotypical modulation h(p_CR_), common to all cells and threshold-induced, and potentially stimulus dependencies inherited from the choice correlation CC(p_CR_). To study the form of CC(p_CR_) associated with a specific source of cross-neuronal correlations we focused on the model of gain induced correlations of Goris et al. 2014 (Eq 10). When studying these gain-induced CP stimulus dependencies we follow Haefner et al. 2013 and adopt a traditional purely feedforward encoding/decoding model. Eq 9 is equal to the expression derived by Haefner et al. 2013, except for the factor h(p_CR_) and considering that the cross-neuronal correlation structure can be potentially stimulus dependent.

We now highlight in lines 230-231 that we model the effect of these gain-induced stimulus dependent cross-neuronal correlations while assuming that the read-out weights -and hence the population size- are stimulus independent. We believe it is reasonable to assume that the read-out weights are independent of the stimulus value, since otherwise the form of the decoder would depend precisely on what the decoder estimates. On the other hand, we admit that the traditional linear decoder (Eq 8) used in the analytical model -following Shadlen et al. 1996 and Haefner et al. 2013, among others- is an approximation of the internal neural decoder, and that stimulus-dependent weights may reflect dynamic aspects of the decision-making process neglected when using the linear decoder. Either way, the assumption of stimulus independent weights can also be taken as a modeling choice to determine which CP stimulus dependencies can be derived purely from the stimulus dependent cross-neuronal correlations induced by gain fluctuations.

In reply to point (2) of the Reviewers above we already described our improvements connecting the extended model to the previous analysis of Haefner et al. 2013, which examined the interpretability of CPs in relation to properties such as the neurometric sensitivity and cross-neuronal correlations for the case of uninformative stimuli. We here address the more specific points of the Reviewers to complement that description. As we now describe in lines 210-229, the CP model including gain fluctuations is a generalization to all stimulus levels of the model of Haefner et al. 2013, and hence their characterization of the CP in relation to the neurometric sensitivity and cross-neuronal correlations for uninformative stimuli still holds and is consistent with our analysis. Indeed, all the analysis of Haefner et al. 2013 focus on the properties of the term CC(p_CR_ = 0.5) in Eq 11, and not on the rest of this expression. The key difference and novelty with respect to this previous work is that in our work we do not focus on the characterization of which neural properties determine the magnitude of CPs across neurons, but within-cell changes of the CP across stimulus levels.

In lines 218-227 we refer to the previous results from Haefner et al. 2013, indicating that the relation between the CP and the neurometric sensitivity for p_CR_ = 0.5 also holds in our model. This is described in more detail in lines 1016-1018 and Equation 21 in Methods. This means that, equally to the previous model of Haefner et al. 2013, neurons with low firing rate and low variability can have a high CP at p_CR_ = 0.5, if they have high neurometric sensitivity. Also the strength of the asymmetric CP(p_CR_) pattern is related to the neurometric sensitivity since in Equation 11 the slope depends on the factor [1-CC_i_(p_CR_=0.5)] and CC_i_(p_CR_=0.5) is proportional to the neurometric sensitivity. Apart from the case of an optimal decoder studied in the main text, we now in the Suppl. Material S4 derive a more general gain model valid for any unbiased decoder, which explains more generally how the strength of the gain-induced asymmetric stimulus-dependence depends on the neurometric sensitivity of each cell (denoted as \eta_i_ in Eqs S12 and S13).

Also the previous conclusions of the model of Haefner et al. 2013 regarding the connection between CPs and the cross-neuronal correlation structure hold in our extended model, since they describe properties of the CP for p_CR_ = 0.5, not properties of the dependence CP(p_CR_) across p_CR_ values (or equivalently across stimulus levels). We believe this is clearer in this revised version since we now describe the gain model in a separate section (lines 205-256) and we explicitly show the extended feedforward model of Eq 9, which shows the dependence of the CP on the cross-neuronal correlation structure. In lines 218234 we explain how this model extends the one of Haefner et al. 2013. To further facilitate the connection of our model with previous work studying the relation between CPs and cross-neuronal correlations, now in the Suppl Material S4 we additionally derived the form of CP(p_CR_) dependencies for the paradigmatic model of a decoder formed by two pools of neurons/ anti neurons (lines 1427-1452). For this model, Haefner et al. 2013 found that, as indicated by the Reviewers, the CP for the uninformative stimulus is determined by the cross-neuronal correlations, in particular by the difference between within-pool and between pools correlations (see now lines 1434-1437). We show that also for this particular two-pool based decoder gain fluctuations are expected to produce an asymmetric CP stimulus dependence, as modeled in Equation S14.

Regarding the novelty of our methods of analysis with respect to Figure 3 of Britten et al. 1996 and Figure 6 of Dodd et al. 2001, please see our detailed reply above to point (1) raised by the Reviewers. Very briefly, the key difference is that our method isolates within-cell CP(p_CR_) profiles from the heterogeneity of the CP magnitude across cells. This is now repeatedly emphasized in the revised paper, and especially described in lines 297-307 and 612-638.

We agree with the description of the Reviewers of the reason why no significant modulation of the CP with p_CR_ is found for the cells with CP<0.5. In lines 330-339 we provide two explanations for this lack of significance. Apart from a smaller power due to the smaller size of the group of cells with CP<0.5, we also indicated that these cells have a smaller CP magnitude. Because the factor h(p_CR_) is multiplicative to CP-0.5, a smaller CP-0.5 results in a weaker CP modulation by p_CR_. We now elaborated on this second point following the argument of the Reviewers that in few cases CPs <0.5 are significant (lines 339-343). We indicate that therefore the fact that we do not observe an inverted U-shape dependence as predicted by our model for the cells with CP<0.5, is not strong evidence against the presence of the existence of a threshold-induced CP stimulus dependence.

We agree with the Reviewers that the separation in two clusters reflects the division between cells with CP>0.5 and CP<0.5. However, beyond this separation, the cluster analysis naturally subdivides the group of cells with CP>0.5 into cells with a predominantly symmetric or asymmetric CP(p_CR_) pattern, and the identification of these patterns is not driven by the sign of the CP>0.5 or <0.5. We have now highlighted that the separation of clusters 2 and 3 in Figure 4b is a subdivision of the cells with CP>0.5 (lines 386-387), although in fact these results are robust and analogous results are found when identifying three clusters without a priori excluding the cells with CP<0.5 (lines 387-389).

4. Editing for accessibility and readabilityUnfortunately, our genuine enthusiasm for the manuscript is somewhat dampened by its length, by its opacity in places, and by the high degree of topic familiarity that it presupposes. For example, the discussion of grand CP on page 7 and in section S2 of the supplementary material, is difficult to follow even for someone in the field. Accordingly, if readability could be improved, the usefulness would be even greater.We appreciate the theoretical advance of the paper. It is very useful to have equations that clarify the relationship between previously used measures (CP, CTA, CC) and the effect of informative stimuli. Overall, by the very nature of the topic, the paper is rather technical. Our impression was that the paper is not easy to read, in particular when we think about a broader readership that is not familiar with details of the theory of CP and the typical interpretation of CP measurements. We understand that this is not an easy job because the interpretation of CP is complicated (bottom-up vs. top-down contributions, relationship to spike count correlations, etc) but it would help to revise the Results section and add some more details and (where possible) intuitive interpretation of the theoretical and experimental results to guide the reader. As an example, it turns out that gain fluctuations are an important factor to explain CP vs pCR but this is only treated very briefly in the results (gain fluctuation model is not explained, no figure is shown about how well the model explains the variability observed in the data, etc).

We did our very best to improve readability while at the same time addressing all concerns and suggestions of the Reviewers. We sincerely appreciate the effort of the Reviewer’s in helping us to improve every aspect of the paper.

We thank the Reviewers for indicating the necessity to make the paper more accessible. Throughout the paper we have worked in improving its readability. We simplified the description of the grand CP in the main text and moved it to the Discussion (old lines 245266, now lines 593-603) to make clear the effect that CP stimulus dependencies may have on the interpretation of grand CPs and the additional information that can provide CP(p_CR_) profiles. We simplified section S2 focusing on the connection between the corrected z-score of Kang and Maunsell and a weighted CP average, which is the result relevant for our reasoning (lines 584-588). The additional description of how for the standard z-score the bias in the estimated grand CP (first pointed out by Kang and Maunsell) can be understood as a consequence of an unnormalized weighted average has been removed, since it is not necessary for our reasoning. See the reply to point (11) below for more details on our improvement of section S2.

We appreciate that in particular our explanation of the gain model in the main text was insufficient and it was unclear how to situate our contributions in relation to the previous work of Haefner et al. 2013. We now better explain the model in the Results in a separate subsection (lines 205-256) and we also have rewritten the description in Methods (lines 882957) and Supl. Material S4. As described in detail in reply to previous points of the Reviewers, we now much better describe the neuronal properties, such as the neurometric sensitivity, that determine the strength of an asymmetric CP(p_CR_) pattern induced by gain fluctuations.

Furthermore, we have expanded the analysis of the symmetric and asymmetric clusters extracted from the cluster analysis using the gain model (lines 443-453). We now also indicate that in cluster 3, when the asymmetric pattern is predominant, the coefficients estimated with the gain model are significantly correlated with the experimental estimates, although overestimating their magnitude. We also better explain the theoretical and experimental limitations in the analysis of the gain model (lines 454-460) despite the merits of this simple model explaining main features of the additional asymmetric CP pattern that could not be explained by the threshold-induced stimulus dependency.

Finally, we acknowledge that we have struggled with the trade-off between length and understandability. Invariably, any point that we cut from the text will have to be filled in by an interested reader. And we worry that the effort to do so for an interested reader, would exceed the annoyance from having to skim over seemingly irrelevant text for a reader interested in other aspects of our work. However, we welcome of course any specific suggestions of where to expand and what to cut beyond the changes in the current revision.

[Editors' note: further revisions were suggested prior to acceptance, as described below.]

Essential Revision:1. Overall, the empirical part looks a little bit like an attempt to drag meaning out of a weak relationship, at least as measured experimentally in the single, but classical data set of Britten et al. The more detailed explanation of the clustering analysis reveals that the main effect of interest is driven by one critical data point from the original Britten et al. data. This is the value of CP for pCR = 0.85, in the data set for which CP > 0.5. Much of the paper depends on just how confident we are about a difference in CP that rises from 0.57 for pCR=0.5 to 0.64 for pCR=0.85. The SEMs calculated for individual neurons in Figure 3d (lower right panel) are not encouraging in this regard.We think that further empirical tests are needed to obtain clarity about significance about key elements of the proposed model but this is beyond the scope of the current manuscript. It should be flagged to the reader, though.In order to clarify to the reader that the empirical part provides only an initial analysis – as was also explicitly mentioned by the authors in their response to the previous reviews – we ask the authors to insert something close to the following statement in the abstract and in the first paragraph of the results:This paper provides preliminary empirical evidence for the promise of studying stimulus dependencies of choice-related signals, which requires further exploration in a wider data set.

We added the following sentence in the Abstract:

“Our analysis provides preliminary empirical evidence for the promise of studying stimulus dependencies of choice-related signals, encouraging further assessment in wider data sets.” (lines 20-22)

At the beginning of the Results section we also added these sentences (lines 85-86):

“This analysis provides preliminary empirical evidence in support of using these new methods for studying stimulus dependencies of activity-choice covariations.”

2. The revisions are acceptable in that the presentation indeed streamlined in many respects (thank you!). However, one large reservation remains: the main findings are NOT explained in an intuitive manner.Basically, when we distinguish neural responses associated with choice D=1 from those associated with choice D=0, we obtain two more or less distinct distributions, P(R|D=1) and P(R|D=0). When these distributions are well separated, then the choice probability CP (average probability that an R|D=1 is larger than an R|D=0) is somewhat larger than when these distributions are more overlapping.However, both qualitative and quantitative aspects of the relation between choice probability CP and choice rate PCR are highly dependent on assumptions. For example, the symmetric U-shape depends on Gaussian variability of responses. When responses are assumed to be Poisson-variable, for example, the symmetric U-shape is replaced by a monotonic decline (Matlab code available upon request).

We really appreciate the Reviewer taking time to probe the robustness of our analytical results. Below, we include Matlab code for a simple simulation based on the neuron-antineuron model (Britten et al. 1992) showing that our results also hold for Poisson neurons.

As explained in lines 759-771, the shape of the factor h(p_CR_) in the choice-triggered average CTA_i_ (Equation 17) is determined based only on the assumption of the distribution p(d) being Gaussian. This approximation is likely excellent due to the Central Limit Theorem since d is the combination of many sensory neurons. We now further justify this in lines 153-155. On the other hand, the relationship between the CTA_i_ and CP_i_ depends also on the Gaussian assumption for the neural response distribution. While this assumption is less accurate (e.g. when Poisson), the CTA-CP relationship derived by Haefner et al. 2013 is very robust to deviations, down to only a few spikes per trial (see Figure S2 in Haefner et al. 2013). As part of an earlier preliminary version of our study (Chicharro et al. bioRxiv cited below), we actually already performed numerical simulations with Poisson responses to verify the robustness of our analytical results and confirmed the above insights. In our simulations, we found divergencies that were minor and with a different shape than the additional asymmetries that the gain model could explain. See Figure 3C-D in:

Decision-related signals in the presence of nonzero signal stimuli, internal bias, and feedback Daniel Chicharro, Stefano Panzeri, Ralf M. Haefner bioRxiv 118398; doi: https://doi.org/10.1101/118398

Based on the insights in the manuscript and numerical checks in the above two references, we can only speculate that in the Reviewer’s simulation the choice correlation might not have been constant as a function of the stimulus, in which case the shape of the threshold-induced modulation h(p_CR_) cannot be isolated from CC(p_CR_). The threshold-induced U-shape, albeit slightly distorted, persists down to populations of only 2 Poisson neurons, firing only 5 spikes per trial on average (the more of either, the better our approximations). We also include in Author response image 1 a figure showing multiple simulations with only 30 neurons. Equivalent results are obtained by increasing both neuron number and within-pool noise correlations.

**Author response image 1. respfig1:** 5 runs simulating 30 Poisson neurons, spiking an average of 5 spikes/trial. For larger populations, and for higher spike counts, our approximations will be even better.

stim=(-0.6:0.2:0.6);%change in the firing rate with informative stimuli

for i=1:length(stim)

r1=poissrnd(5+stim(i),15,1e4); % 15 neurons & 1e4 trials

r2=poissrnd(5-stim(i),15,1e4); % 15 anti-neurons & 1e4 trials

choice=sign(mean(r1)-mean(r2)); % difference of average population responses

choice(choice==0)=2*binornd(1,0.5,1,sum(choice==0))-1;

pcr(i)=sum(choice==1)/length(choice); % choice ratio

for j=1:size(r1,1)

cp2(j,i)=ChoiceProbability(r1(j,choice==-1),r1(j,choice==1));

end

end

plot(pcr,mean(cp2),'b.-');

(ChoiceProbability(x,y) is a function returning the choice probability value for the two sets of responses x, y)

Gain modulation changes the picture in exactly this way. After gain modulation, response variability is no longer Gaussian and, additionally, the set of responses with the larger average is more affected than the set with the smaller average. In other words, gain modulation alters the shape of response distributions and in consequence also the shape of CP = f(PCR).

Equation 7 shows that the dependence of CP on p_CR_ can be decomposed into two parts. The first part is the modulation h(p_CR_), shared by all neurons. As discussed above, this modulation appears in the CTA_i_ under the assumption of p(d) being Gaussian, it is inherited by the CP_i_ (Equation 7) and is robust for distributions p(r) that depart from Gaussianity. The second part is a modulation specific to each neuron that is due to a stimulus-dependence of the choice correlation CC_i_. To be precise: it is not the non-Gaussianity of p(r) that determines this dependence of CP on p_CR_, it is the stimulus-dependence of CC_i_ (Equation 7).

These insights were the motivation for our gain model: the idea that the stimulus dependent covariance of Equation 10 would likely result in a stimulus-dependent CC(s) that then is reflected in CP(p_CR_). It is through this additional contribution CC(p_CR_) and not modifying the U-shape factor h(p_CR_) induced by the threshold that the gain fluctuations contribute to the CP stimulus dependencies.

In spite of this reservation, I have to confess that this paper has been quite useful to me, because it forced me to think through these issues.

Thank you very much. Making these issues explicit, and shedding light on the deeper relationships underlying CP, has been a major motivation for our paper.

3. With regards to the dependence of the results on one critical data point in the Britten et al. data set (CP for pCR = 0.85) and the question of significance to other data sets, various suggestions for further data worth testing were made.We wonder, for instance, whether Geoff Ghose's lab (https://pubmed.ncbi.nlm.nih.gov/30067123/ ; https://pubmed.ncbi.nlm.nih.gov/19109454/ ) is a better source of data to support this exercise. In particularly, his recordings could be searched for signs of noisy gain modulations, which is the mechanism that lies at the core of this analysis. Other suggestions were as mentioned before the Dodd et al. 2001/Wasmuht et al. 2019 data.But we agree this is beyond the current scope of the manuscript.

Thank you very much for these suggestions. We agree with the Reviewers that further data sets will need to be examined in the future to confirm the existence of CP stimulus dependencies. In fact, in particular the shape of any dependence associated with the Choice Correlation CC(p_CR_), can be expected to depend on the particular role of the cells and on the task, so it would be very interesting to examine different data sets and the CP stimulus dependencies therein and to understand how any difference of their shape across data sets can be explained in terms of the particularities of the tasks or the properties of the cells. We also have high hopes for new datasets that come with cell type and layer information, as well as from chronic recordings where the same neurons are held across multiple days allowing for more accurate CP estimates.

4. Line 10: "activity-choice covariations are traditionally quantified with a single measure of choice probability (CP), without characterizing their changes across stimulus levels" I appreciate the need to demonstrate novelty in the paper, but this statement does a disservice to earlier researchers who recognized the possibility of a stimulus-choice interaction but did not find any evidence of such an interaction. The earlier papers are clear on this point. So the novelty here is not the failure of earlier researchers to think through their results carefully; the novelty here is an apparent improvement in sensitivity of the analysis methods. This may sound less exciting but is still important, if correct.

We agree with the Reviewers. Indeed, in the original submission we were giving historical context to the focus on a single CP based on early works such as the one of Britten et al. 1996, which examined the possibility of CP stimulus dependencies but did not find a significant dependence. This was lost in the simplifications of the Introduction. We now have recovered this important explanation in lines 36-38. We have also substituted ‘traditionally’ by ‘commonly’ in the piece of text cited by the Reviewers.

5. Line 65 "if the decision process uses a decision threshold"

Please see our explanation under (7) below. We expanded this sentence to include

‘if the decision-making process relies on a threshold mechanism (or threshold criterion) to convert a continues decision variable into a binary choice’ (lines 66-67)

6. Line 96 "choice probability, CP, defined as the probability that a random sample from all trials"; the reader only learns on line 99 what this is a random sample of. It would better read as "a random sample of neural activity r " and then explain in a separate sentence what r might be in any particular experimental situation.

Thank you, we now indicate that it is a sample of neural activity earlier in the explanation (line 99).

7. Line 147: "threshold value 𝜃 " the authors insist on referring to this parameter 𝜃 as a threshold value. This usage will inflame debate as to whether there is a "high threshold assumption" baked into this model, where "high threshold" here means a classical high threshold in visual detection models, as opposed to a signal detection model. As classical high threshold theory is now rejected for detection models, I think it would be better here and throughout to refer to 𝜃 as a criterion value, which is what it is and better aligns with the language of signal detection theory.

Thanks for pointing out this potential misunderstanding. We use ‘threshold’ as it is used for example in the literature of decision-making models that consider the accumulation of evidence until a threshold is reached, triggering a decision, and also often in the literature examining decisions in LIP. For example, the idea of a decision threshold is used in this way in the Review paper by Gold and Shadlen 2007. The model used in Shadlen et al. 1996, and in Haefner et al. 2013 can be viewed as a simplified analytically tractable version of this type of models.

To minimize misunderstandings, we added further explanation in the Introduction (see our answer to point (5)) that allows understanding the meaning of this decision threshold. We believe that this initial clarification, together with the detailed explanation of the meaning of the decision threshold early in the Results section and in Figure 1 should avoid confusion. The revised text reads as follows (lines 64-67):

‘We show that they can also appear for all neurons because of the transformation of the neural representation of the stimulus into a binary choice, if the decision-making process relies on a threshold mechanism (or threshold criterion) to convert a continues decision variable into a binary choice.’

8. Lines 129-169: this development is still very difficult to follow and labors over what are some fairly basic points. It would better be rewritten with better structure at about half the length. For example, compare lines 111-112 and 134-135, which could be combined into a single point that is made once at the right stage in the argument.

We simplified this explanation removing the more technical details of the threshold model not yet required at this point (lines 122-123). However, we believe these are important conceptual points: carefully relating the neuron-specific quantities to the choice-specific one, and we worry that shortening would increase the risk of a first-time reader missing important information. Also, some level of redundancy between lines 111-112 (now 116-117) and 134-135 (now 136-137) is justified by the discursive structure of the text: lines 111-112 serve to contrast with the subsequent explanation, that indicates that despite the lack of additional assumptions in the definition of the CP and CTA, their interpretation has been mostly led by a feedforward model. On the other hand, in lines 134-135 we are already describing the assumptions of a threshold mechanism, introducing the continuous decision variable d.

9. Lines 170-209: the writing continues in a stilted manner with multiple cross-references to other material.

To improve readability, we now connected the explanations of lines 174-179 with the previous text. Nonetheless, we believe all of these are important points, and that the cross-references are important, too. We don’t know how to simplify without omissions, or risking comprehension by a first-time reader. However, we would gladly implement any specific suggestion.

This section accomplishes 3 things:

- Explaining the possible sources of variation in p_CR_.

- Justifying the use of the linear relationship for all our explanations and pointing to the exact formulas.

- Laying out the empirical predictions of our theoretical results.

10. Lines 173-4 "We here will refer to CP stimulus dependencies and CP(𝑝CR ) patterns interchangeably". We rather fear that this is going to cause a lot of readers to trip over. I can see that these two are interchangeable from the theoretical perspective of these authors, but many will think that behavior pCR is dependent on a number of factors other than the stimulus. In the field of monkey neurophysiology, pCR will depend on reward, attention, arousal and so forth, in a way that the stimulus does not. What We think the authors actually mean is something like "Within the structure of our model, there is a fixed relationship between the dependence of CP on the stimulus strength and the dependence of CP on the choice rate pCR, for each threshold."

We have now clarified this point. We now write in lines 174-179: “Note that we do not distinguish between CC stimulus dependencies and a dependence of the CC on p_CR_. We do not make this distinction here because most generally a change in the stimulus level results in a change of p_CR_, and the two cannot be disentangled. However, the p_CR_ more generally depends on other factors such as the reward value, attention level, or arousal state, and in Equation 7 the separate dependencies on the stimulus and p_CR_ can be explicitly indicated as CC_i_(p_CR_, s) when the experimental paradigm allows to separate these two influences.”

So the model is flexible to accommodate different dependencies of CC on the stimulus s and on p_CR_, because the details of this dependencies are irrelevant for the model, that is, the threshold model does not characterize the form of CC(p_CR_), or CC(p_CR_, s). The reason why we do not distinguish between the dependence on p_CR_ and on s is because in most experimental settings the two are intertwined. This is the case for example in the Britten et al. data we reanalyze, in which a relation p_CR_(s) is estimated experimentally by the psychometric function. We hope that explicitly mentioning that Equation 7 can be rewritten with CC_i_(p_CR_, s) when these two effects are separable will avoid the misunderstanding that our model is limited to assuming a one to one relation between p_CR_ and stimulus levels.

11. Lines 263-265 "we show how to extend Generalized Linear Models (GLMs), a popular model to characterize the factors modulating neural activity, to include a stimulus-choice interaction terms" As a statistical procedure, this is fairly routine stuff and could be abbreviated considerably.

Thanks, we have shortened this description, now in lines 270 and 478.

12. Lines 376-377 "Specifying the existence of two clusters, we naturally recovered the distinction between cells with CP higher or lower than 0.5" We struggled to find a clear and unambiguous summary of the statistics associated with this. We can see that a consistent pattern emerges when the cluster number is increased from 2 to 3, but looking at the distributions in Figure 4c and Figure 4d does not appear to reveal clusters. The significance values in Figures4a and 4b relate to the significance of the modulation effect for each cluster, not the significance of the cluster separations. The methods section and supplementary analysis section cross-reference each other but neither seems to answer the simple question of whether 1 versus 2 clusters is statistically justified, let alone the step from 2 to 3.

As correctly indicated by the Reviewers, the significance values relate to the modulation effects, and not to the cluster separations. The analysis of the Supplementary figure S2c indicates that the distinction between the symmetric and asymmetric cluster is robust, in the sense that they still contain a substantial number of cells even when the number of clusters is allowed to be 6. However, our objective here was not to conclude that there is a concrete number of separable clusters. Indeed, we argue in the discussion that from considerations about the structure of stimulus-dependent feedback signals, it could be expected to exist a richer structure of CP stimulus dependencies (lines 683-694). This richer uncharacterized structure may explain the lack of separability of the clusters in the distributions of Figure 4c and 4d, although it may be due as well to noise in the estimated CP values. We limited the analysis to three clusters because of the limitations of statistical power of the data set. We considered a third cluster because qualitatively it was instructive to detect a CP dependence that was not symmetric as predicted by the threshold-effect h(p_CR_), since the asymmetric CP dependence implies (according to Equation 7) that it has to be originated through a choice correlation CC(p_CR_) modulation.

We now clarify in lines 425-431 that we do not claim that these clusters are the only existing ones, or even that a finite number of CP(p_CR_) patterns exists, leading to a finite number of clusters. For example, if the CP(p_CR_) profiles were associated with the structure of stimulus-dependent feedback across cells with different tuning functions, a continuum of CP(p_CR_) dependencies would be expected, in agreement with the continuum of tuning functions.

13. Lines 612-638 This discussion suggests that there may be within-cell changes in CP as a function of pCR that may have been hidden by pooling across populations of cells. But in the end, this paper has problems in detecting real changes in this relationship at the level of recordings from single cells. The small SEMs that attach to the data in Figure 3a do indeed reflect pooling across a population sample; they do not relate to changes in individual neurons. The panel in Figure 3d shows the true picture for individual cells. So this discussion begs the question as to why the population analyses in Dodd et al. do not show the predicted relationships. The point made by the authors about within-cell changes does not appear to be material in this regard.

The difference may be in the fact that, as indicated in lines (302-303), our average corresponds to an average -across cells- of within-cell CP(p_CR_) profiles. The importance of this is discussed in lines 304-309. It is further motivated in lines (569-576). When we average across cells we are averaging their individual within-cell CP(p_CR_) profiles. Therefore, for each particular p_CR_ value all cells contribute in the same way.

As explained in lines 629-634, from Figure 6 of Dodd et al. 2001 we cannot directly read the within-cell CP(p_CR_) profiles. Given that, in our understanding, the set of stimulus levels presented to each cell were different, it is not possible to think of an average of all the dots in Figure 6a with CP>0.5 as analogous to an average of the within-cell CP(p_CR_) profiles of the cells with CP>0.5. We do not know though how strong was the variability in the set of stimulus levels presented to each cell and whether using our method a clear CP(p_CR_) dependence would be found. We do not discard that in that data set there is really no significant CP stimulus modulation, but we believe that our argument about how to isolate within-cell CP(p_CR_) profiles –even if averaged– from across cell heterogeneities, is still applicable in this case.

We do appreciate that a higher number of trials per neuron would be required to definitively answer the question about the nature of the CP(p_CR_) relationship, and that this manuscript provides theory and insights that we will only be fully exploitable in the future. With regard to Dodd et al., it is entirely possible that CPs in MT during the rotating cylinder task do not depend on p_CR_, while they do depend on p_CR_ during a classic motion discrimination task. This is exactly what would be expected if the CPs during the rotating cylinder task were mostly due to feedback from a binary variable – as entirely plausible – rather than linked to the decision through a continuous decision-variable. Only in the latter case does our theory predict the symmetric h(p_CR_) relationship.